# Uncertainty in the projected Antarctic contribution to sea level due to internal climate variability

Justine Caillet[1], Nicolas C. Jourdain[1], Pierre Mathiot[1], Fabien Gillet-Chaulet[1], Benoit Urruty[1], Clara Burgard[1,2], Charles Amory[1,3], Mondher Chekki[1], and Christoph Kittel[1,4,5]

[1]Univ. Grenoble Alpes, CNRS, INRAE, IRD, Grenoble INP, Institut des Géosciences de l'Environnement, Grenoble, France
[2]Laboratoire d'Océanographie et du Climat, LOCEAN-IPSL, Sorbonne Université, CNRS-IRD-MNHN, Paris, France
[3]Laboratoire de Sciences du Climat et de l'Environnement, LSCE-IPSL, CEA-CNRS-UVSQ-UMR8212, Université Paris Saclay, Gif-sur-Yvette, France
[4]Department of Geography, Laboratory of Climatology, SPHERES, University of Liège, Liège, Belgium
[5]Physical geography research group, Department geography, Vrije Universiteit Brussel, Brussels, Belgium

**Correspondence:** Justine Caillet (justine.caillet@univ-grenoble-alpes.fr)

**Abstract.** Identifying and quantifying irreducible and reducible uncertainties in the Antarctic Ice Sheet response to future climate change is essential for guiding mitigation and adaptation policy decision. However, the impact of the irreducible internal climate variability, resulting from processes intrinsic to the climate system, remains poorly understood and quantified. Here, we characterise both the atmospheric and oceanic internal climate variability in a selection of three CMIP6 models (UKESM1-0-LL, IPSL-CM6A-LR and MPI-ESM1.2-HR) and estimate their impact on the Antarctic contribution to sea-level change over the 21$^{st}$ century under the SSP2-4.5 scenario. To achieve this, we use a standalone ice-sheet model driven by the ocean through parameterised basal melting and by the atmosphere through emulated surface mass balance estimates. The atmospheric component of internal climate variability in Antarctica has a similar amplitude in the three CMIP6 models. In contrast, the amplitude of the oceanic component strongly depends on the climate model and its representation of convective mixing in the ocean. A low bias in sea-ice production and an overly stratified ocean lead to a lack of deep convective mixing which results in weak ocean variability near the entrance of ice-shelf cavities. Internal climate variability affects the Antarctic contribution to sea-level change until 2100 by 45% to 93% depending on the CMIP6 model. This may be a low estimate as the internal climate variability in the CMIP models is likely underestimated. The effect of atmospheric internal climate variability on the surface mass balance overwhelms the effect of oceanic internal climate variability on the dynamical ice-sheet mass loss by a factor of 2 to 5, except in Dronning Maud area and Amundsen, Getz and Aurora basins where both contributions may be similar depending on the CMIP model. Based on these results, we recommend that ice-sheet model projections consider ($i$) several climate models and several members of a single climate model to account for the impact of internal climate variability and ($ii$) longer temporal period when correcting historical climate forcing to match present-day observations.

## 1 Introduction

The Antarctic Ice Sheet (AIS) is losing mass at an increasing rate (Rignot et al., 2019; Shepherd et al., 2019), particularly in the Amundsen and Totten/Moscow sectors, where ocean-induced melting under floating ice shelves is relatively high (Jenkins et al., 2018; Hirano et al., 2023). The AIS response to future climate change, including its potential instability (Garbe et al., 2020; Armstrong McKay et al., 2022), is one of the main sources of uncertainty in projections of global sea-level rise (Fox-Kemper et al., 2021), with an estimated contribution over 2015–2100 ranging from -5 to 43 cm under a high-end anthropogenic emission scenario (ISMIP6, Edwards et al., 2021).

Estimates of the AIS contribution to future sea-level rise are currently mostly based on standalone ice-sheet models, driven by atmospheric and oceanic data from the Coupled Model Intercomparison Project (CMIP, Eyring et al., 2016). The diversity of climate conditions across the CMIP models explains an important part of the uncertainty in some of the drainage basins (Seroussi et al., 2023), despite the use of an anomaly method to reduce known biases in the CMIP models (Jourdain et al., 2020; Purich and England, 2021). Internal climate variability is usually not accounted for in the uncertainty of AIS projections. A single study, so far, has estimated that this uncertainty could be 18-21% higher due to internal climate variability (Tsai et al., 2020). This was estimated using a single ice-sheet model and two versions of the same climate model.

Climate variability is the combination of two components, on the one hand the variability resulting from external forcing of both natural (e.g., volcanoes or solar activity) and anthropogenic (e.g., $CO_2$ emissions) sources, and internal variability on the other hand. The latter results from processes intrinsic to the climate system, due to the chaotic nature of fluid dynamics and to non-linearities in the coupled interactions between the ocean, atmosphere, land and cryosphere (e.g., Kravtsov et al., 2007; Penduff et al., 2018; Gwyther et al., 2018; Hogg et al., 2022). For a given climate model, the impact of internal climate variability can be isolated by considering several forced simulations with identical external forcing but slightly different initial conditions. For this reason, an increasing number of CMIP models include several members which differ only in their initial state.

A part of the internal climate variability can be characterised as modes such as the El Niño/Southern Oscillation and Inter-decadal Pacific Oscillation, which have remote connections with the Amundsen Sea Low (ASL, Holland et al., 2022; Dalaiden et al., 2023). The ASL is a low-pressure system located over the South Pacific sector of the Southern Ocean, which generates decadal wind anomalies that affect the oceanic undercurrent along the continental slope, thereby modulating the amount of warm water flowing towards the ice shelves of the Amundsen Sea Embayment (Silvano et al., 2022). The regional influence of these modes makes internal climate variability particularly strong in the Amundsen sector: internal climate variability is thought to be responsible for the retreat of Pine Island's grounding line in the 1940s (Holland et al., 2022), and mid-depth ocean warming trends over the 21[st] century can vary by a factor of two depending on the phasing of internal climate variability (Naughten et al., 2023).

In this paper, we first investigate atmospheric and oceanic internal climate variability of several CMIP6 models. Then, using a standalone ice-sheet model forced by CMIP6 model outputs, we quantify the impact of internal climate variability on Antarctic

Sea Level Contribution (SLC) over the 21$^{st}$ century under the medium SSP2-4.5 scenario for both the whole ice sheet and the main basins, especially the Amundsen basin which is expected to be particularly affected by internal climate variability.

## 2 Methods

### 2.1 CMIP6 models

We choose to analyse three CMIP6 models to get a more general picture of the internal climate variability than we would get using a single model.

The selected models are UKESM1-0-LL (19 members, Sellar et al., 2020), MPI-ESM1.2-HR (10 members, Müller et al., 2018) and IPSL-CM6A-LR model (33 members, Boucher et al., 2020). This choice was made based on ($i$) the size of their ensemble (at least 10 members), ($ii$) the availability of 6-hourly outputs that were needed to run regional climate projections, and ($iii$) their representation of the present-day oceanic and atmospheric properties.

For the third point, the three selected models are in the best half of the CMIP6 ensemble according to Agosta (2024) who ranked 45 models based on several atmospheric variables relevant for precipitation over Antarctica. These three models also have a high fidelity in the representation of the mean ocean properties, as detailed in Appendix A.

Although their oceanic and atmospheric mean state are some of the closest to observations, the three selected models have distinct characteristics of their internal climate variabilities. As shown in Appendix B, the atmospheric variability at the scale of Antarctica is close to the multi-model median in IPSL-CM6A-LR and MPI-ESM1-2-HR, while it is much higher in UKESM1-0-LL. The oceanic variability is among the lowest in MPI-ESM1-2-HR, close to the multi-model median in UKESM1-0-LL, and much higher in IPSL-CM6A-LR.

It is interesting to note that both UKESM1-0-LL and IPSL-CM6A-LR have prescribed ice-shelf melting that is vertically distributed to mimic the presence of unresolved ice-shelf cavities (Mathiot et al., 2017), which is known to be important for coastal ocean properties around Antarctica (Mathiot et al., 2017; Donat-Magnin et al., 2021). Most CMIP models prescribe meltwater fluxes at the surface, which tends to increase the ocean stratification (Mathiot et al., 2017) and reduce exchanges between the surface and deeper waters, thereby limiting variability at depth.

### 2.2 Ice-sheet model

We use the version v9.0 of the Elmer/Ice finite element model (Gagliardini et al., 2013), in a configuration of the entire Antarctic Ice Sheet adapted from Hill et al. (2023). The ice dynamics is computed by solving the Shallow Shelf Approximation (SSA) of the Stokes equations (MacAyeal, 1989), assuming an isotropic rheology following Glen's flow law (Glen, 1955) and a linear friction law (i.e., $\tau_b = Cu_b$ where $\tau_b$ is the basal shear stress, $C$ the friction coefficient and $u_b$ the basal velocity). The location of the grounding line is determined using a flotation criterion and a sub-grid scheme is applied for the friction in partially floating elements (SEP3 in Seroussi et al., 2014).

The mesh is refined both close to the grounding line and in areas where observed surface velocities and thickness show high curvatures (i.e., high second derivative of the modelled field, Gillet-Chaulet et al., 2012). The mesh has a maximum size of 50 km in the very interior of the ice sheet and a minimum size of 1 km in the refined areas. The model domain does not change over time, but the ice thickness is subject to a lower limit of 1 m and elements that reach this limit are considered deglaciated in the post-processing. For stability reasons, the domain boundary is slightly smoothed and isolated icebergs (ice-covered area disconnected from the ice sheet) with less than 7 elements, are removed if they appear during the simulation (i.e, their thickness is set to the critical thickness of 1 m). Apart from these corrections, we assume a steady calving front.

Inverse methods (Gillet-Chaulet et al., 2012; Brondex et al., 2019) provide viscosity and friction parameters by minimising the misfit between modelled and observed velocities from Mouginot et al. (2019) using the ice thickness from BedMachine-Antarctica-v2 (Morlighem et al., 2020). Details of the inversion are available in Hill et al. (2023). Our model configuration does not represent a prognostic evolution of ice temperature and damage that may affect viscosity in transient simulations. From the inversion step, we run a 20-year "relaxation" under the present-day forcing described hereafter. This attenuates the artificial high surface elevation rate of change that occurs when we switch from a diagnostic to a prognostic simulation (Gillet-Chaulet et al., 2012).

The PICO box model (Reese et al., 2018) is used to parameterise ice-shelf basal melting, with a distinct calibration from Hill et al. (2023). Here, the parameters are those detailed in Reese et al. (2023), i.e., $C = 2 \, \mathrm{Sv} \, \mathrm{m}^3 \, \mathrm{kg}^{-1}$ and $\gamma_T = 5.5 \times 10^{-5} \, \mathrm{m} \, \mathrm{s}^{-1}$, which are based on the observed or ocean-modelled sensitivity of melt rates to ocean temperature changes. The present-day sea floor temperature and salinity for each of the 19 regions defined in Reese et al. (2018) are extracted from the ISMIP6 ocean climatology (Jourdain et al., 2020) and averaged within 50 km of the ice-shelf front as described in Burgard et al. (2022). A correction of temperature, ranging from -1.8°C to 0.6°C with respect to the ocean climatology, is added to match the 1994-2018 observational melt estimates from Adusumilli et al. (2020) for each of the 19 regions. This regional correction resulted in improved estimates of local ice-shelf melting, except for Totten and Thwaites ice shelves (see Fig. 1). This correction differs from Reese et al. (2023) as the current ice-sheet geometry and the oceanic climatology used in this study are different from the one considered in Reese et al. (2023).

The present-day Surface Mass Balance (SMB) is based on the 1995-2014 climatology (a period of relatively stable SMB) of the RACMO-2.3.p2 regional climate model (Van Wessem et al., 2018). In contrast to Hill et al. (2023), we do not correct the surface mass balance to maintain a steady state, but we lower the inverted friction coefficients to reduce the model drift. For this, we minimise the model bias in West Antarctic grounded ice mass loss with respect to the 1995-2014 observational estimate of The IMBIE Team (2018). West Antarctica is chosen to tune the basal friction coefficients as the ice dynamics is known to strongly explain mass loss in this sector. We then apply the resulting 10% correction to the friction coefficients of the entire ice sheet. The resulting model configuration overestimates the mass loss trend in the West Antarctica by only 6% but still largely overestimates mass gain in East Antarctica and in the Peninsula (Tab. 1). As a consequence, the simulated Antarctic Ice Sheet is currently gaining a little mass (+36 $\mathrm{Gt}\,\mathrm{yr}^{-1}$, Tab. 1), instead of losing mass as observed (-106±55 $\mathrm{Gt}\,\mathrm{yr}^{-1}$, Tab. 1). This growing bias is quite common in ice-sheet models (Seroussi et al., 2020; Aschwanden et al., 2021). However, this bias

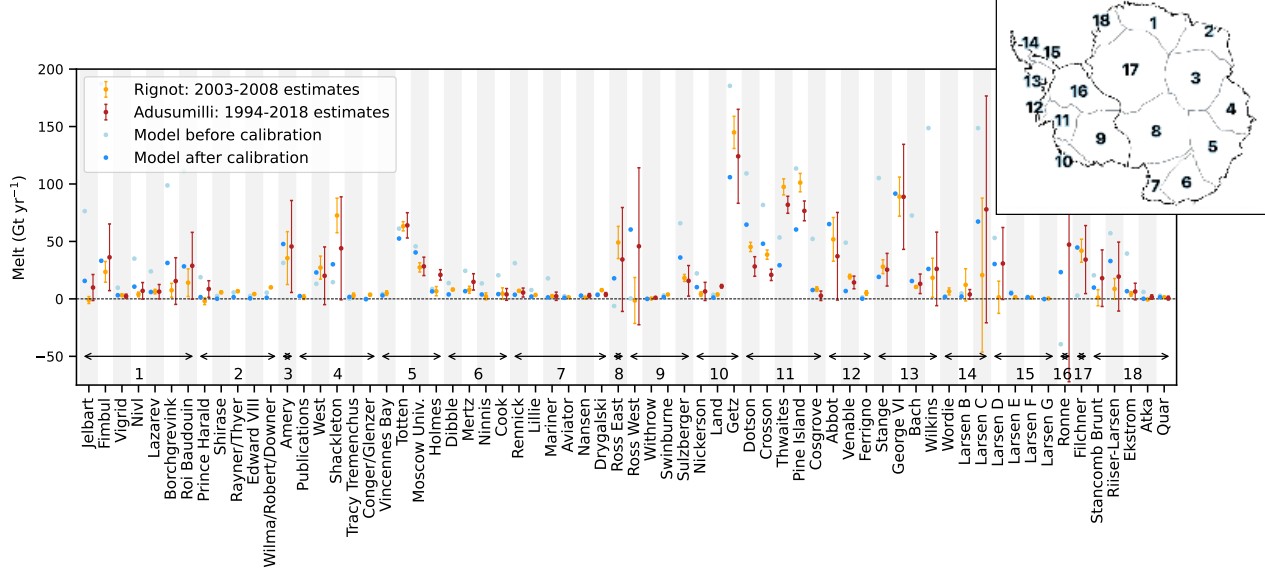

**Figure 1.** Basal melt rate of main ice shelves over the period 1994-2014 before (lightblue) and after calibration (blue) compared to the melting estimates over the period 1994-2018 from Adusumilli et al. (2020, in red). Observed data uncertainties correspond to one standard deviation. Note that the data from Adusumilli et al. (2020) only cover the area northward of 81.5°S, which excludes part of the Filchner-Ronne and Ross ice shelves. Melting estimates over the period 2003-2008 from Rignot et al. (2013, in orange) are shown for comparison. Numbers from 1 to 18 indicate the basin where ice shelves are located, as shown in the top right-hand corner.

**Table 1.** Rates of ice-sheet mass change $(\mathrm{Gt\,yr^{-1}})$ for the entire Antarctic Ice Sheet (AIS) and its three major basins: East Antarctica (EAIS), West Antarctica (WAIS) and Peninsula (APIS). The IMBIE data are from The IMBIE Team (2018).

|      | IMBIE estimates (1995-2014) | Elmer/Ice (1995-2014) |
|------|------------------------------|------------------------|
| AIS  | -106±55                      | +36                    |
| EAIS | +8±45                        | +107                   |
| WAIS | -93±27                       | -127                   |
| APIS | -21±15                       | +35                    |

should not impact most of the analyses presented here, as the projections in response to the CMIP6 climate models are analysed relatively to each other.

## 2.3 Ice-sheet projections to 2100

The future mass imbalance of Antarctica results from combined effects of changes in surface mass balance (SMB) and ice dynamics. In standalone ice-sheet simulations, variations in surface mass balance can be attributed to the atmosphere and dynamical mass loss can be attributed to the ocean as SMB changes have little impact on the Antarctic dynamical contribution to sea level over a century (Seroussi et al., 2014, 2023). Thus, the effect of changes in surface mass balance and ice dynamics can be analysed separately and then summed to reconstruct the total Antarctic sea-level contribution (Bindschadler et al., 2013).

In this study, variations in SMB are evaluated through the emulation of a regional climate model driven by the atmosphere of the selected CMIP6 models, while the dynamical mass losses are calculated using the ice-sheet model Elmer/Ice driven by the ocean of the selected CMIP6 models.

We use the medium SSP2-4.5 scenario, which corresponds to a global warming of 1.4 to 3.0°C from 1995-2014 to 2081-2100 (90% confidence interval, Lee et al., 2021) and considered plausible in view of current efforts to tackle climate change

(Hausfather and Peters, 2020). As the choice of greenhouse gas emission scenario has only a limited impact on the projected Antarctic contribution to sea-level rise until 2100 (Seroussi et al., 2020), we have not repeated our calculations for other scenarios.

The SMB contribution to sea level, for each available member of the three selected CMIP6 models (see Appendix C), is directly deduced from the emulation of the behaviour of a regional climate model driven by the first member of the three

selected CMIP6 models. Regional climate projections were not used to calculate the future SMB of ISMIP6-Antarctica (Nowicki et al., 2020; Seroussi et al., 2020), mostly because they were not available early enough in the intercomparison process. Since then, this kind of simulations have been used to refine SMB projections (Kittel et al., 2021, 2022). Using a dedicated regional climate model is particularly important for the IPSL-CM6A-LR model given that its snow physics over ice sheets is too simple to simulate firn saturation and runoff in a warmer climate. However, running the regional climate model driven by

many members of the CMIP ensemble would be computationally too expensive and practically not feasible due to the non availability of 6-hourly output for most members, which are needed to drive the regional model. In this paper, we therefore use the approach developed by Jourdain et al. (2024) to emulate the behaviour of the *Modèle Atmosphérique Régional* (MAR, Kittel et al., 2021). This method uses exponential fits of accumulation and surface melting perturbations due to changes in surface air temperature, as well as simple physical relationships to derive runoff and SMB. This method is thoroughly evaluated in

Jourdain et al. (2024) for the emulation of other CMIP models and scenarios based on a few existing regional simulations, and here we apply it to emulate other members based on existing regional simulations of the first member of each CMIP model. As previously done in ISMIP6 (Nowicki et al., 2020), we calculate annual anomalies (with respect to 1995-2014 mean SMB) and add them to the present-day SMB. Based on available CMIP6 outputs for SSP2-4.5 scenario, we calculate SMB for 11 members of IPSL-CM6A-LR model, 17 members of UKESM1-0-LL model and 2 members of MPI-ESM1.2-HR model.

For each selected CMIP6 model, the contribution of ice dynamics to sea level is estimated through Elmer/Ice simulations driven by the SMB of the first member (as SMB changes have little impact on the Antarctic dynamical contribution to sea level over a century, the choice of SMB member does not matter) and the ocean of several members. We then remove the

SMB contribution of the first member to isolate the dynamical contribution of each member. All the Elmer/Ice simulations start from the same state, corresponding to 2014. Because of the numerical cost of our simulations, we select a limited number of members for ocean. In addition to the first member, the selection is made over the current period (1995-2014 means) to cover the widest range of values for the ocean temperature on the continental shelf in the Amundsen Sea. We focus on this region as ($i$) the largest mass loss is observed there and has been attributed to the ocean, and ($ii$) the amplitude of the standard deviation of the 1995-2014 mean potential temperature across all members is particularly high in this region (see section 3.1). The annual ocean potential temperature and practical salinity from CMIP model outputs were interpolated to a stereographic (8 km $\times$ 8 km $\times$ 60 m) grid, then extrapolated to fill unrepresented areas as in Jourdain et al. (2020). The corresponding ocean anomalies were then added to the present-day temperature and salinity to feed the ice-shelf basal melt parameterisation. In total, we run 11 simulations (see Appendix C), five with the IPSL-CM6A-LR model, four with the UKESM1-0-LL model, and only two with the MPI-ESM1.2-HR model given that its oceanic variability is very low (see section 3.1) and the number of available members for the SSP2-4.5 scenario very limited.

## 3  Results

We first characterize internal climate variability of the oceanic (subsect. 3.1) and atmospheric (subsect. 3.2) components in the selected CMIP6 models. For this, we use all available members and we describe the effect of internal climate variability on the present-day mean state, i.e., 1995-2014, which is used as a reference for the calculation of anomalies in ISMIP6 and in our Elmer/Ice simulations. Then, we estimate the importance of internal climate variability for sea-level projections by examining transient Elmer/Ice simulations from 2015 to 2100 (subsec. 3.3), driven by the subset of the CMIP6 ensemble used in the subsections 3.1 and 3.2.

### 3.1  Oceanic internal climate variability

Oceanic internal climate variability is investigated through salinity and temperature variability. Oceanic internal climate variability at mid-depth is much weaker in MPI-ESM1.2-HR than in IPSL-CM6A-LR and UKESM1-0-LL (Fig. 2). MPI-ESM1.2-HR shows a relatively low and homogeneous internal climate variability on the continental shelf, with standard deviations of 0.02 g kg$^{-1}$ and 0.06°C across the members (Fig. 2a,d). The mean salinity of this model is too low over the whole continental shelf (34.2 g kg$^{-1}$) compared to the World Ocean Atlas dataset (WOA 2018, Boyer et al., 2018), particularly in front of the Ross and Filchner ice shelves (Fig. 3a,b). This suggests that the weak internal climate variability is related to an underestimation of dense water formation (Fig. 3b).

For IPSL-CM6A-LR and UKESM1-0-LL, salinity exhibits higher variability over the whole continental shelf (around 0.03-0.04 g kg$^{-1}$ in Fig. 2b,c) but this does not systematically lead to a high variability in temperature (Fig. 2e,f). A region that undergoes large variability in mid-depth temperatures in both IPSL-CM6A-LR and UKESM1-0-LL is the region extending westward from the Bellingshausen Sea to the western Ross Sea. There are nonetheless noticeable differences between the two models.

For IPSL-CM6A-LR, the largest variability in mid-depth salinity is found in the western part of the Ross Sea, where High Salinity Shelf Water (HSSW) is formed (Fig. 2b). The deepest part of the Ross Sea is occupied by the densest water mass, so that there is a competition between intrusions of relatively warm and salty Circumpolar Deep Water (CDW) advected from offshore and the production of cold dense water (HSSW) through sea-ice formation and associated convection (Siahaan et al., 2022; Mathiot and Jourdain, 2023). The variation between the occupation of these two water masses may explain the high mid-depth temperature variability in the Ross Sea (Fig. 2e). In contrast, the variability in salinity at the HSSW formation site of the eastern Weddell Sea (Fig. 2b) is probably too weak to be associated with any CDW intrusion (Fig. 2e).

For UKESM1-0-LL, the highest variability in mid-depth salinity is located around Prydz Bay in East Antarctica (Fig. 2c), which is an area of important dense shelf water formation (Williams et al., 2016). It nonetheless does not induce a strong temperature variability near the ice shelves (Fig. 2f). An interesting feature of UKESM1-0-LL (and IPSL-CM6A-LR to a lower extent) is the high salinity variability beyond the continental shelf, northward of the Amundsen Sea (Fig. 2c), which coincides with a region of high variability in sea-ice concentration (not shown) and air temperature (Fig. 4f).

We now focus on the Amundsen Sea, as the region is currently experiencing the largest mass loss in Antarctica. In MPI-ESM1.2-HR, the first 100 m are much fresher than observed and than in the two other models (Fig. 2g), and the entire water column is too warm with an overly strong and shallow thermocline (Fig. 2j). Sea-ice concentration is considerably lower than for the other two models and observations (Fig. 3i-l), which results in a lack of deep convection on the continental shelf. The low oceanic internal climate variability in MPI-ESM1.2-HR may result from this lack of convection, which prevents atmospheric internal climate variability from propagating into the deep ocean.

The weaker stratification in IPSL-CM6A-LR and UKESM1-0-LL than in MPI-ESM1.2-HR indicates the presence of more convective mixing, as convection mixes cold and salty water produced by sea-ice formation with warmer water at depth. Consequently, both IPSL-CM6A-LR and UKESM1-0-LL exhibit more realistic temperature profiles than MPI-ESM1.2-HR in the Amundsen Sea. All the IPSL-CM6A-LR members are nonetheless cold biased at depth (weakest bias of -0.75°C at 900 m depth in Fig. 2k), while all the UKESM1-0-LL members are warm biased (weakest bias of +0.54°C at 900 m depth in Fig. 2l). The spread across the ensemble is large for both models, with 0.79°C (IPSL-CM6A-LR) and 0.39°C (UKESM1-0-LL) difference in the 1995-2014 mean temperature at 900 m between the extreme members.

These conclusions remain valid for 60-year averages as well as 20-year averages, albeit with attenuated internal climate variability. For example, there is still 0.43°C (IPSL-CM6A-LR) and 0.34°C (UKESM1-0-LL) difference in the 1955-2014 mean temperature at 900 m between the extreme members (Fig. D1). This finding is consistent with a strong internal climate variability at multi-decadal time scales in the Amundsen Sea, as previously pointed out by Purich and England (2021) who identified typical periodicity of approximately 30 years for MPI-ESM1.2-HR, 70 years for IPSL-CM6A-LR and 120 years for UKESM1-0-LL (their Fig. S6). In comparison, paleoclimate reconstructions indicate a ∼50-year period for the wind variability at the Amundsen Sea shelf break (Holland et al., 2022).

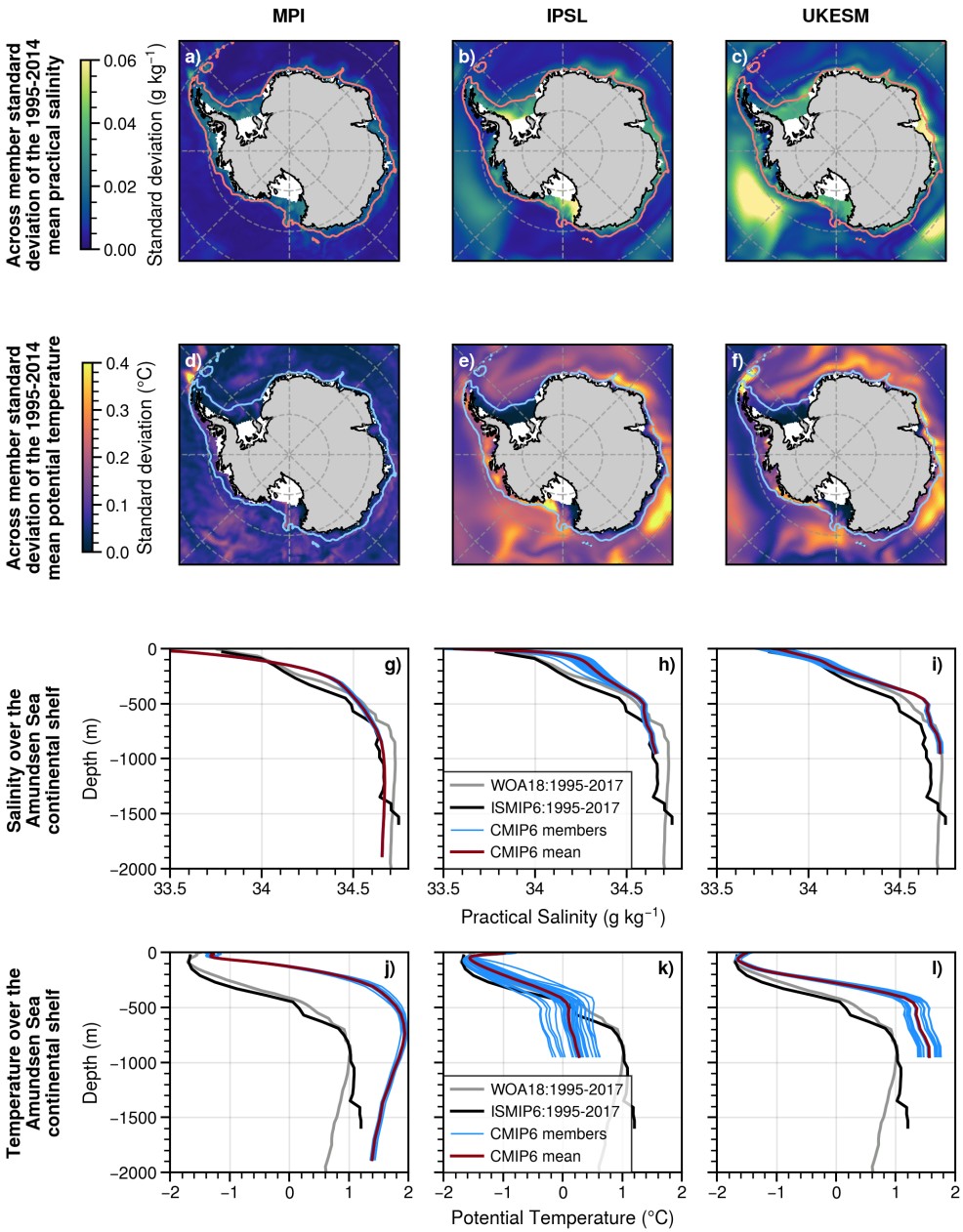

**Figure 2.** Comparison of the saline and thermal properties of the CMIP6 models MPI-ESM1.2-HR (left), IPSL-CM6A-LR (middle) and UKESM1-0-LL (right). (a-c) standard deviation of the 1995-2014 mean practical salinity across the ensemble relative to the multi-member mean, considering the salinity averaged from 200 m to 700 m depth. The 1500 m isobath (pink) delimits the continental shelf. (d-f) same as (a-c) but for potential temperature and with the 1500 m isobath in blue. (g-i) mean vertical profiles of practical salinity on the Amundsen Sea continental shelf (as defined in Caillet et al., 2023). For each model, the blue curves represent the individual members (1995-2014 mean), and the red line the multi-member mean. The grey curve corresponds to the 2018 World Ocean Atlas data (WOA 2018, Boyer et al., 2018) over the period 1995-2017 and the black curve to observational climatology based on the WOA, EN4 and MEOP datasets and built for ISMIP6 (Jourdain et al., 2020). (j-l) same as (g-i) but for potential temperature.

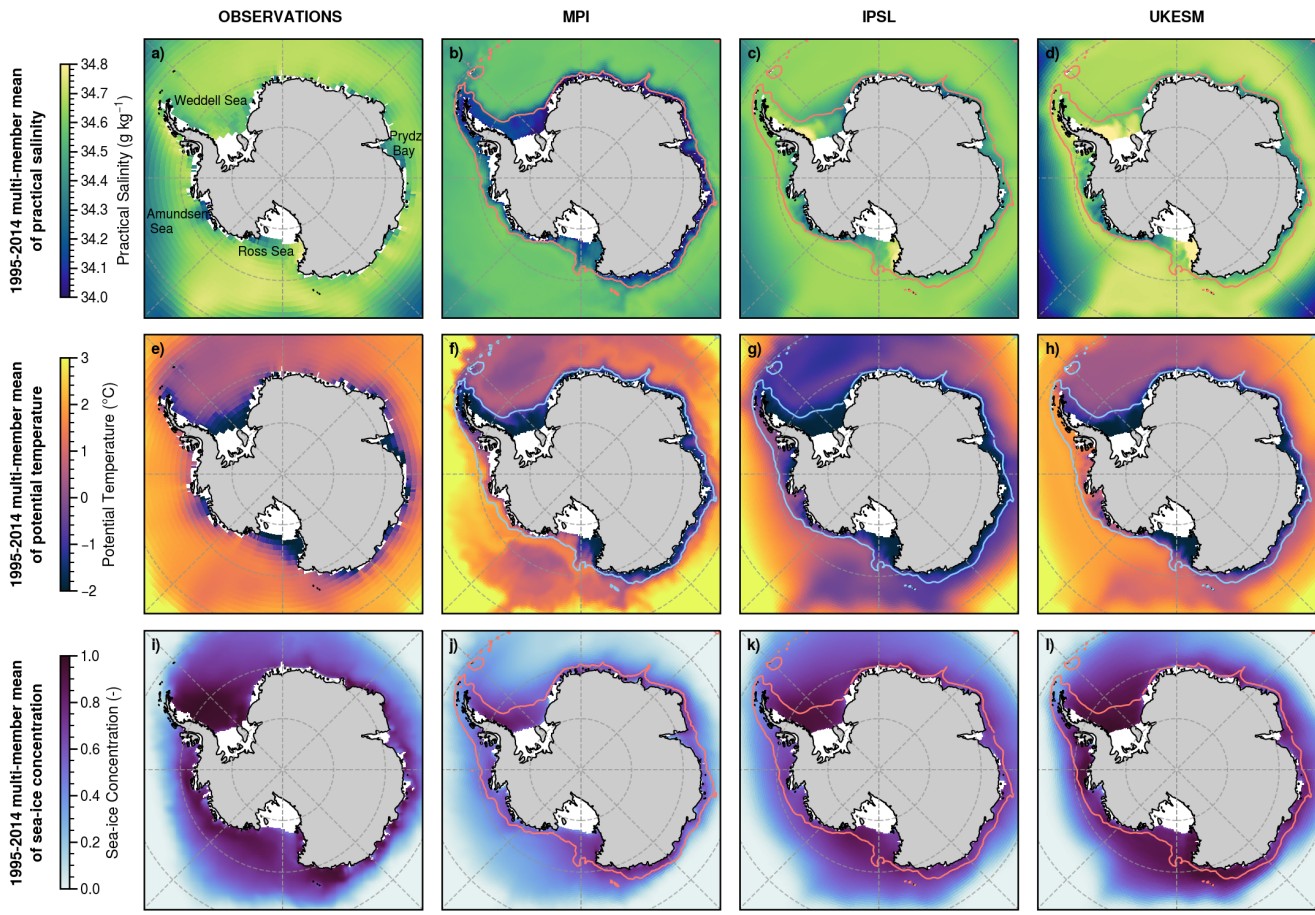

**Figure 3.** Comparison of the saline, thermal and sea-ice properties of observations (left) and CMIP6 models MPI-ESM1.2-HR (middle left), IPSL-CM6A-LR (middle right) and UKESM1-0-LL (right). (a) 1995-2017 mean practical salinity from the 2018 World Ocean Atlas datasets(WOA 2018, Boyer et al., 2018), considering the salinity averaged from 200 m to 700 m depth. (b-d) 1995-2014 mean practical salinity across the climate model ensemble, considering the salinity averaged from 200 m to 700 m depth. The 1500 m isobath (pink) delimits the continental shelf. (e) same as (a) but for potential temperature. (f-h) same as (b-d) but for potential temperature and with the 1500 m isobath in blue. (f) 1995-2014 mean sea-ice concentration from NSIDC dataset (version 4.0) (Comiso, 2023). (j-l) same as (b-d) but for sea-ice concentration.

## 3.2 Atmospheric internal climate variability

The SMB is defined as the difference between precipitation (liquid and solid, positive contribution) and evaporation, sublimation and runoff (negative contribution). The present-day Antarctic SMB mostly consists of snowfall, a small part of which (<10%) is sublimated at the surface and in blowing snow (Van Wessem et al., 2018; Agosta et al., 2019; Mottram et al., 2021). Runoff is currently negligible as most of the meltwater refreezes due to cold temperatures. By 2100 and for the SSP2-4.5 medium scenario, runoff is expected to remain limited (Kittel et al., 2021), so the SMB is projected to increase largely due to the increased water vapour saturation in warmer air, resulting in more precipitation (e.g. Krinner et al., 2008; Agosta et al., 2013). We therefore focus on variability in emulated SMB and its main components such as precipitation and air temperature.

In contrast to the ocean, the atmospheric internal climate variability is relatively similar in the three selected CMIP6 models (Fig. 4). This is partly due to similar emulated present-day SMB: the integrated value over the whole ice sheet ranges between 2641 and 2892 $\mathrm{Gt\,yr^{-1}}$ for all members of the three models. The present-day SMB internal climate variability is stronger in coastal regions (Fig. 4a-c) where the average SMB is higher (Fig. E1), consistent with the precipitation variability in the CMIP simulations (Fig. 4g-i).

The largest SMB variability is simulated along the coast of the Amundsen and Bellingshausen seas, which results from the high internal climate variability of atmospheric circulation (e.g., Amundsen Sea Low position) and air temperature in these regions (Fig. 4d-i). UKESM1-0-LL also exhibits significant variability in the Dronning Maud region. As previously reported by Marshall and Thompson (2016), the internal climate variability of sea-level pressure and air temperature have the typical characteristics of the two Pacific-South American modes (usually referred to as PSA1 and PSA2), which are associated with wave trains originating in the tropical Pacific and possibly modulated by feedbacks with clouds and sea ice (Wang et al., 2022).

## 3.3 Impact of internal climate variability on the Antarctic contribution to sea level

In our ice-sheet projections, Antarctica gains mass over the century for all selected members of the three CMIP models, with an estimated SLC in 2100 ranging from -1.34 to -8.46 cm (Fig. 5a). This contribution results from a compensation between (i) increased ice mass flux through the grounding line driven by the ocean (Fig. 5b), mainly occurring in West Antarctica (Fig. 5h) and (ii) increasing SMB (Fig. 5c), occurring in all regions for almost all members (Fig. 5f,i,l).

Regions behave in different ways. While the East Antarctica and the Peninsula gain mass (SLC in 2100 ranging respectively from -3.80 to -6.32 cm in Fig. 5d and from -0.96 to -2.24 cm in Fig. 5j), West Antarctica looses mass (SLC in 2100 ranging from +0.11 to +3.78 cm in Fig. 5g). The West Antarctic positive SLC is mostly explained by the dynamical response of Pine Island and Thwaites ice shelves (∼3 cm in Fig. 6c, basin 11) as well as Getz ice shelf (∼1 cm, basin 10). The absolute trends in East Antarctica and the Peninsula regions are largely influenced by the unforced drift in Elmer/Ice (see section 2.2), but the simulations can still inform on the sensitivity to internal climate variability.

Internal climate variability affects the estimated SLC of Antarctica in 2100 by more than 45%, 79% and 93% for the IPSL-CM6A-LR, UKESM1-0-LL and MPI-ESM1.2-HR models respectively (considering the difference between the lowest and highest member divided by the multi-member mean). Thus, the estimated SLC can vary by 1.64 cm, 4.35 cm and 2.33 cm,

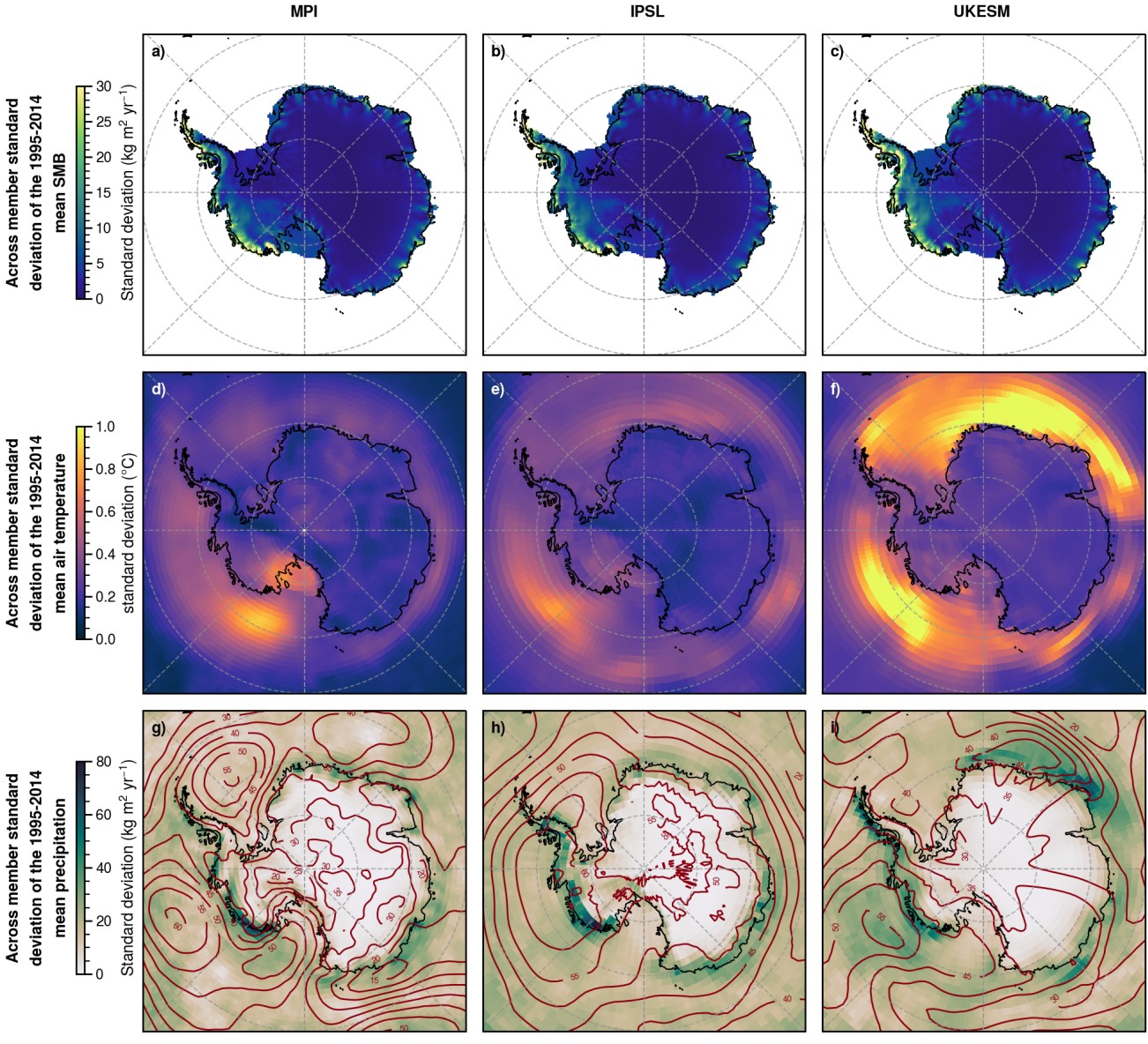

**Figure 4.** Comparison of the internal climate variability in surface mass balance, air temperature, precipitation and sea-level pressure in MPI-ESM1.2-HR (left), IPSL-CM6A-LR (middle) and UKESM1-0-LL (right). (a-c) standard deviation of the 1995-2014 mean SMB across the ensemble relative to the multi-member mean. The SMB shown is not a direct CMIP6 output but is derived from emulated behaviour of the regional climate model MAR driven by selected CMIP6 models. (d-f) same as (a-c) but for air temperature at 2 m (directly from the CMIP6 outputs). (g-i) same as (a-c) but for total precipitation (shaded) and sea-level pressure (contours every 5 hPa) from the CMIP6 outputs.

respectively (Fig. 5a) (considering the difference between the lowest and highest member). This uncertainty is comparable to that associated with the selection of the CMIP6 model (3 cm, Fig. 5a).

For the three climate models and in most Antarctic regions, the effects of atmospheric internal climate variability overwhelm the effects of oceanic internal climate variability (Figs. 5-6). On average, by the end of the century, the amplitude of SLC variability related to the atmosphere (Fig. 5c) is 3.4 times higher than that related to the ocean (Fig. 5b). However, there are significant spatial variations across the individual basins and CMIP models.

The West Ross, Getz and Amundsen basins (No. 9,10,11 in Fig. 6) show the most significant atmospheric and oceanic variability in the WAIS region. For the IPSL-CM6A-LR model, internal oceanic variability even exceeds atmospheric variability in these basins (Fig. 6b-c). As described in the previous paragraphs, this variability results from competition of CDW intrusions and convective mixing on the continental shelf (subsect. 3.1), and from the atmospheric circulation, especially the varying position of the Amundsen Sea Low depending on the members (subsect. 3.2). It should be noted that the MPI-ESM1.2-HR model does not show any internal oceanic variability, as expected from the analyses carried out in subsect. 3.1.

In East Antarctica, the Totten basin, which is currently experiencing the highest melt rates in East Antarctica (Rignot et al., 2019), and the Dronning Maud basin (No. 5 and 1 in Fig. 6) show strong internal oceanic variability reaching or exceeding the internal atmospheric variability for the three CMIP6 models. The other basins, like those of the Peninsula, show low basal melting and are largely dominated by internal atmospheric variability, induced primarily by interconnections with the tropical Pacific (see subsect. 3.2).

## 4 Discussion

### 4.1 Robustness of internal climate variability in climate models

Since all the diagnoses we have done are based on CMIP models, the realism of their internal climate variability needs to be addressed.

Parsons et al. (2020) compared the distribution of standard deviation of global mean surface air temperature of CMIP piControl simulations to paleoclimate proxies representative of the 1450-1849 period (PAGES2k, 2019). While some of the CMIP6 models had a high-biased temperature variability, the three models used in this study are within the observational plausible range [0.03;0.15], with standard deviation (for variability beyond 25-year timescales) of 0.12°C in IPSL-CM6A-LR, 0.09°C in UKESM1-0-LL and 0.08°C in MPI-ESM1.2-HR.

However, based on ice core reconstructions of temperatures at the surface of Antarctica over the past 1,000 years, Casado et al. (2023) estimated that the internal climate variability was underestimated over Antarctica in the CMIP5 and CMIP6 models, although the three models used here were not part of the assessment. Previdi and Polvani (2016) suggested that the SMB interannual internal climate variability is well captured by the CMIP5 models, but this is only based on the reanalysis period and is therefore more relevant for the interannual variability than for the multi-decadal variability that is emphasized in our work. The higher fidelity of the internal climate variability in CMIP models at the interannual frequency than at multi-decadal frequencies was indeed reported by Cheung et al. (2017) for the main modes of variability in the Atlantic and Pacific

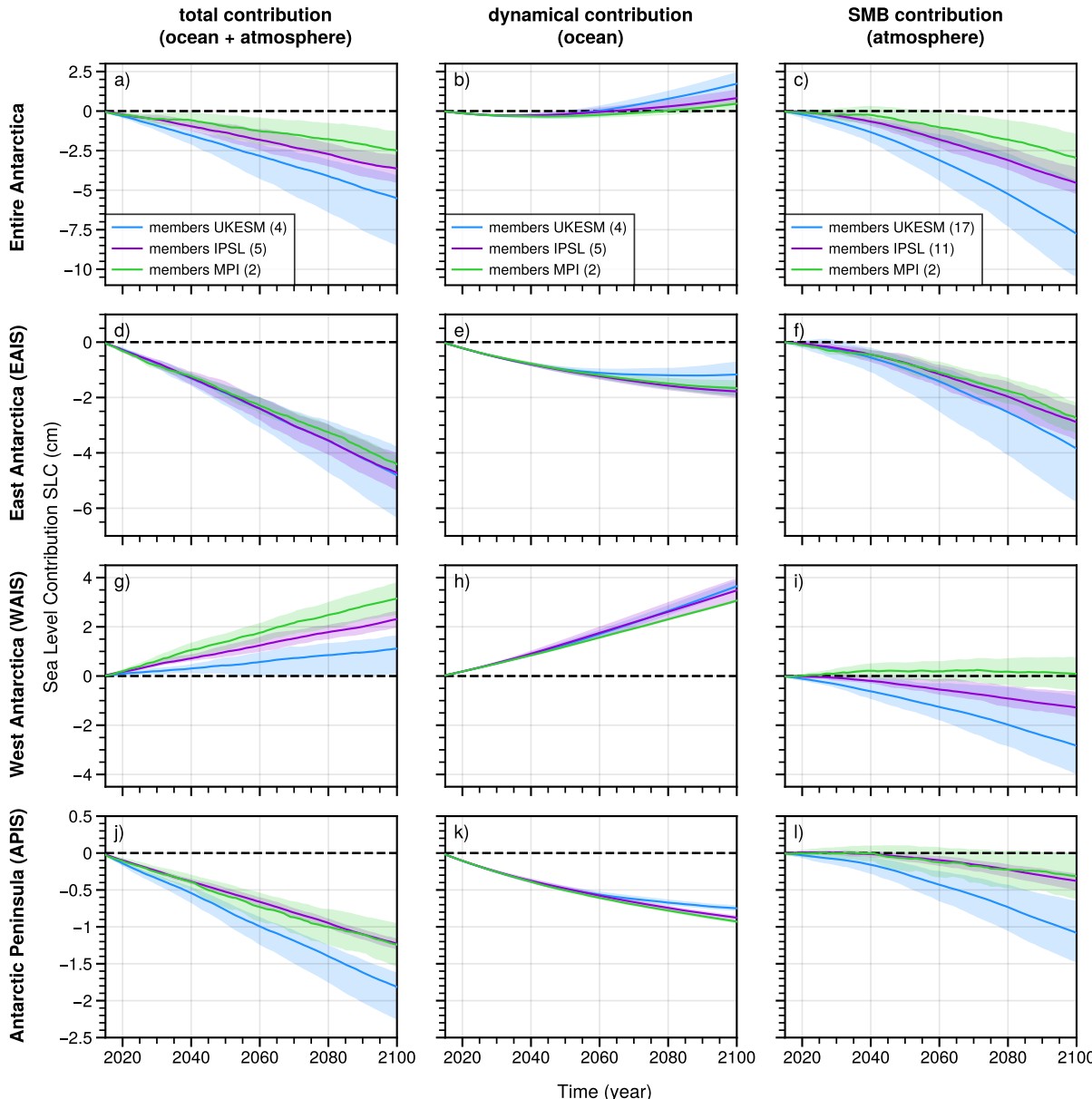

**Figure 5.** Antarctic Sea Level Contribution (SLC) over the $21^{st}$ century relative to year 2015 under the SSP2-4.5 scenario, for MPI-ESM1.2-HR (green), IPSL-CM6A-LR (purple) and UKESM1-0-LL (blue). Results are displayed for the whole ice sheet (upper row) and for the main sub-regions (as defined in The IMBIE Team, 2018). The left rows show the combination of the dynamical ice-sheet contribution (modulated by the oceanic internal climate variability, middle row) and the surface mass balance contribution (modulated by the atmospheric internal climate variability, right row). The dynamical contribution is calculated from the change in volume above flotation minus the accumulated SMB changes, using the method described in Goelzer et al. (2020) to convert to sea-level variations. The SMB contribution is calculated over the grounded ice area of BedMachine-Antarctica-v2, which is very close to Elmer/Ice's initial state (difference of less than 0.1% in grounded area). The use of the grounded ice area from BedMachine-Antarctica-v2 instead of the one from Elmer/ice (which takes into account the grounding line retreat), impacts the SLC due to the SMB by less than 1 mm. The solid line represents the multi-member mean, while the shaded area represents the range of values covered by the ensemble members. The number in bracket refers to the number of selected members for each CMIP6 model.

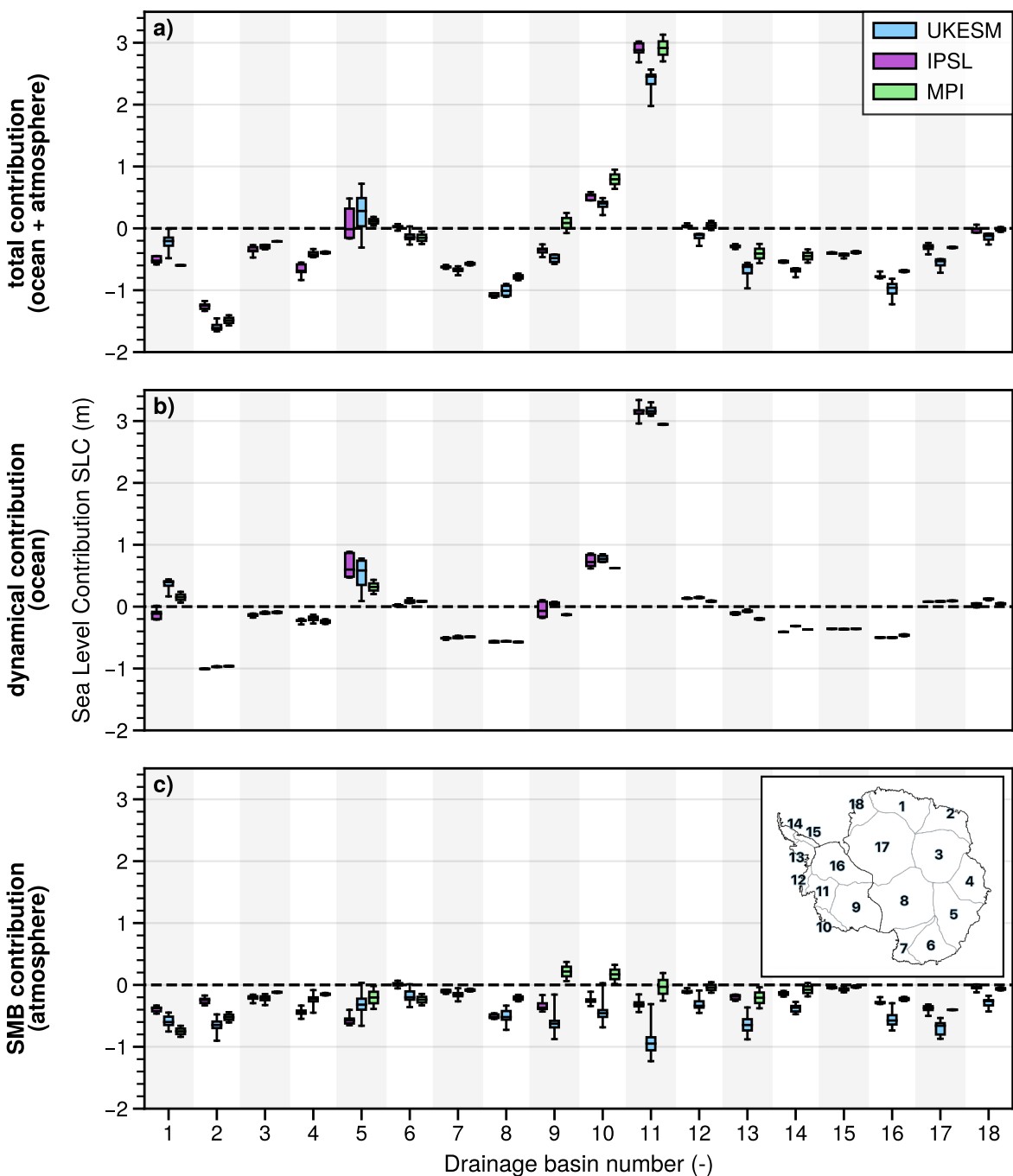

**Figure 6.** Regional Antarctic Sea Level Contribution (SLC) in 2100 relative to year 2015 for MPI-ESM1.2-HR (green), IPSL-CM6A-LR (purple) and UKESM1-0-LL (blue), integrated over the IMBIE drainage basins shown in (c) The IMBIE Team (2018). a) all contributions, b) dynamical ice-sheet contribution modulated by the ocean internal climate variability and d) the SMB contribution modulated by the atmospheric internal climate variability (see methods and definitions in the caption of Fig. 5). The box plots correspond to the ensemble median (line), interquartile range (box) and total range (whiskers) of each model.

oceans. Both IPSL-CM6A and MPI-ESM1.2-HR have an internal variability of their 20-year mean surface air temperature close to the CMIP6 multi-model median (Appendix B), so their atmospheric multi-decadal variability is possibly underestimated given the results of Casado et al. (2023). Nevertheless, this variability is significantly stronger in UKESM1-0-LL, which suggests that our study may cover realistic atmosphere internal variability.

Our results also show that the amplitude of oceanic internal climate variability around Antarctica strongly depends on the climate model. When compared with 12 other CMIP6 models (Appendix B), the three selected models cover the whole range of oceanic multi-decadal variability in the CMIP6 ensemble, with one of the lowest values (MPI-ESM1.2-HR), one close to the multi-model median (UKESM1-0-LL) and one of the highest values (IPSL-CM6A-LR). The low variability of the MPI-ESM1.2-HR model is inconsistent with the temperature and salinity profiles observed in the Amundsen Sea (Dutrieux et al., 2014; Jenkins et al., 2018), which likely results from model biases (see subsect. 3.1). We therefore consider that the plausible range of oceanic variability is covered by IPSL-CM6A-LR and UKESM1-0-LL. It is nonetheless important to keep in mind that these CMIP models do not resolve eddies, which have been suggested to generate substantial low-frequency oceanic internal variability in the Southern Ocean (Sérazin et al., 2017).

Furthermore, climate models do not explicitly include the ice sheet, even though the non-linearities due to ice-sheet–ocean and ice-sheet–atmosphere interaction have the potential to generate internal climate variability (Kravtsov et al., 2007; Gwyther et al., 2018). To capture the full uncertainty due to internal climate variability, ice-sheet models would ideally be fully coupled to climate models and be run for multiple members. Although still challenging (Smith et al., 2021), this would enable a consistent representation of internal climate variability, including the effects of ice-sheet–ocean and ice-sheet–atmosphere feedbacks.

Therefore, the low-frequency internal climate variability that affects the ice-sheet mass through oceanic and atmospheric pathways is probably underestimated in current climate models and its impact on the Antarctic SLC as well.

### 4.2 Internal climate variability as a source of uncertainty in sea-level projections

The comparison of the amplitude of SLC in 2100 due to internal variability (shaded area in Fig. 5a) with the one due to the choice of climate model (difference between extreme thick lines, Fig. 5a) shows that the choice of climate model and internal climate variability both have a similar impact on Antarctic SLC. The relative importance of internal climate variability in our simulations (45-93%) is higher than the 18-21% reported by Tsai et al. (2020). However, its absolute importance is lower in our simulations, with a 2015-2100 SLC modulated by 1.6 to 4.4 cm, versus 8 cm in Tsai et al. (2020). This is likely due to the fact that the SLC projections of Tsai et al. (30 to 48 cm) are at the very high end of the ensemble of other ice-sheet projections (Seroussi et al., 2020; Payne et al., 2021; Coulon et al., 2023), which is partly due to the parameterised ice-shelf hydrofracturing and ice cliff failure in Tsai et al. (2020) as opposed to the aforementioned other models. In contrast, our simulations are at the very low end of the ensemble of other ice-sheet projections (-8.5 to -1.3 cm, Fig. 5a). This is partly due to the present-day drift in East Antarctica and Peninsula that we did not remove from our projected trends as opposed to the aforementioned other models.

The anomaly method used to build the ocean and atmospheric forcing in both our experiments and in ISMIP6 (Nowicki et al., 2020) was designed to correct biases in SMB and ocean-induced melting over the 1995-2014 period. However, given the wide

confidence interval on a 20-year means ([0.06°C;0.24°C] for air temperature and [0.02°C;0.12°C] for oceanic temperature, see Fig. B1), correcting a random phase of the historical CMIP simulations towards the actual 1995-2014 period may significantly shift the projections. For example, members with colder forcing over the present-day period become warmer throughout the 21[st] century due to the correction. Casado et al. (2023) recommend averaging over 50-years to be long enough to weaken internal climate variability and short enough not to dilute forced trends. This corresponds to the typical period of internal climate variability in the paleoclimate reconstructions (Holland et al., 2022). As discussed in subsect. 3.1, some models like UKESM1-0-LL nonetheless have internal climate variability over longer periods, so that 50-year averages do not attenuate internal climate variability to a significant extent. Another issue with extending the period over which the correction is applied is that not so many observations were available 50 years ago in Antarctica.

Given the difficulties of correcting biases, it is tempting to select the members that are most in phase with observations and not to apply any bias correction, which is investigated in the next subsection. It is nonetheless important to consider that the anomaly method is only responsible for a part of the uncertainty associated with internal climate variability. Indeed, Tsai et al. (2020) highlighted important internal climate variability despite correcting the 1920–2012 period. As ice-sheet modelers sometimes run large ensemble simulations to select or weight the members that best fit observational records (e.g., Coulon et al., 2023), it seems important that they either consider multiple climate model members or select the more realistic ones.

### 4.3 Identifying the best member

For greater confidence in the ice-sheet projections, the models have to be initialised and calibrated to match historical observations. Given the importance of internal variability, selecting the CMIP member that is most in phase with the observational record might be useful to achieve it. Such a member could also be primarily used for projections when running multiple members is too computationally expensive. Here, we investigate this possibility with the example of the IPSL-CM6A-LR and UKESM1-0-LL models.

The first challenge is to define metrics that can be used to quantify the phasing of individual members. Among the observations that are available over several decades, it is somehow an expert judgement to decide which metrics are most relevant for the Antarctic mass variations. Here, we choose several metrics to ensure:

– a good representation of the mean atmospheric and oceanic states. We selected variables directly used to drive the ice-sheet model, such as SMB for the atmosphere and temperature for the ocean. For the ocean, we focused our analyses on the Amundsen sector as the region experiences the current main mass loss and CTD profile data are available for a relatively long period from 1994 to 2018.

– a good representation of the amplitude of oceanic variability using the same observational data described in the previous paragraph. We did not evaluate the variability of SMB since it has been relatively stable in recent years.

– a good representation of important modes of variability known to affect the ocean and atmosphere in/around Antarctica. We focus our analyses on the indices representative of the Southern Annular Model and the Interdecadal Pacific Oscillation.

– a good phasing of internal variability with observations, which could be important for future detection/attribution studies and for projected Antarctic contribution to sea-level rise. We chose two variables, sea-ice concentration and the presence of warm periods on the continental shelf of the Amundsen Sea to provide insights on the phasing of internal variability.

The metrics definition and the rank of all members are presented in Appendix F. Overall, ranks are not very consistent across the chosen metrics, and no member is best for all metrics. Although the perfect member does not exist, some members

nevertheless seem more in phase with the observed climate variability than the other members. For the IPSL-CM6A-LR model, member 26 seems to be the most consistent with the observed variability despite a lack of variability in front of Pine Island and a sea-ice trend that is mostly negative as opposed to the positive observed trend. For the UKESM1-0-LL model, member 4 seems to be the most consistent with the observed variability despite an overestimated SAM trend and SMB but also a negative sea-ice trend.

However, the member selection appears very sensitive to the list of chosen metrics, and the phasing of the best member is only marginally better than for the other members. Given the number of degrees of freedom of climate models, it would probably be unrealistic to expect finding a member perfectly in phase with the observed variability among ensembles of a few tens of members, even if models were not biased. For these reasons, it appears judicious to consider several climate model members in ice-sheet projections, to account for the substantial uncertainty related to internal climate variability.

For the same reason, the initialisation of ice-sheet models should account for internal climate variability, either by starting from various members and/or by including internal climate variability in the long initialisation of some ice-sheet models, as previously suggested by Robel et al. (2023).

## 5   Conclusions

In this study, we show that internal climate variability affects the Antarctic contribution to changes in sea level until 2100, for

medium-range scenario, by 45%-93%, i.e., a variation between 1.6 and 4.4 cm under the SSP2-4.5 scenario. This may be a low estimate as the internal climate variability of the CMIP models is likely underestimated. In our case, the uncertainty in Antarctic contribution to sea level due to internal climate variability is of comparable magnitude as the uncertainty related to the choice of the climate model. The internal climate variability has a strong multi-decadal component, so that ($i$) it is not completely diluted over a century, and ($ii$) it strongly affects the 20-year averages used to build the forcing anomaly.

By the end of the century, the effect of atmospheric internal climate variability on the surface mass balance overwhelms the effect of oceanic internal climate variability on the dynamical ice-sheet mass loss by a factor of 2 to 5, except in Amundsen, Getz and Aurora basins where both contributions may be similar depending on the CMIP model.

The atmospheric internal climate variability over Antarctica has similar amplitudes in the three CMIP6 models analysed in this study. Conversely, the amplitude of oceanic internal climate variability around Antarctica strongly depends on the

380 climate model. The oceanic internal climate variability in the MPI-ESM1.2-HR model is very low, which may be explained by underestimated ocean convective mixing on the continental shelf, due to either biases in the sea-ice behaviour or in the ocean stratification.

From these results, we recommend using the following practices for future ice-sheet projections:

- A CMIP model selection based on the assessment of the model ability to produce a plausible multi-decadal variability in both the atmospheric and oceanic drivers of ice-sheet changes (in addition to the usual mean state assessment, which should ideally be done for multiple members).

- The consideration of several members for each climate model forcing given the difficulty or impossibility to identify a perfect member (2 or 3 members can already be very informative if running more is too computationally expensive).

- The use of longer reference period for the calculation of anomalies than that usually used (e.g., 20 years in ISMIP Nowicki et al., 2020) as climate models show important modes of variability longer than 20 years. Casado et al. (2023) recommend averaging over 50 years to be long enough to weaken internal climate variability and short enough not to dilute forced trends. Few observations were available 50 years ago in Antarctica, so the observational climatologies will likely remain representative of 20-30 years. This nonetheless likely remains a preferable approach than using the last 20 years.

## Appendix A: Assessment of mean oceanic properties in the CMIP6 models

The present-day oceanic properties of multiple CMIP6 models are assessed through a review of three studies which evaluate water masses properties in the Southern Ocean and Antarctic seas (Purich and England, 2021), oceanic and atmospheric metrics relevant for the Southern Ocean dynamics (Beadling et al., 2020), and bottom properties in the Southern Ocean (Heuzé, 2021). Each of these studies provides the bias of several CMIP model variables. We normalise the bias of individual variables by the multi-model standard deviation, and we rank the models based on the increasing RMSE calculated over the variables of a given study (Fig. A1). Tab. A1 details the variables and observations used to estimate the model biases. The analysis is done here for the first available member of each model.

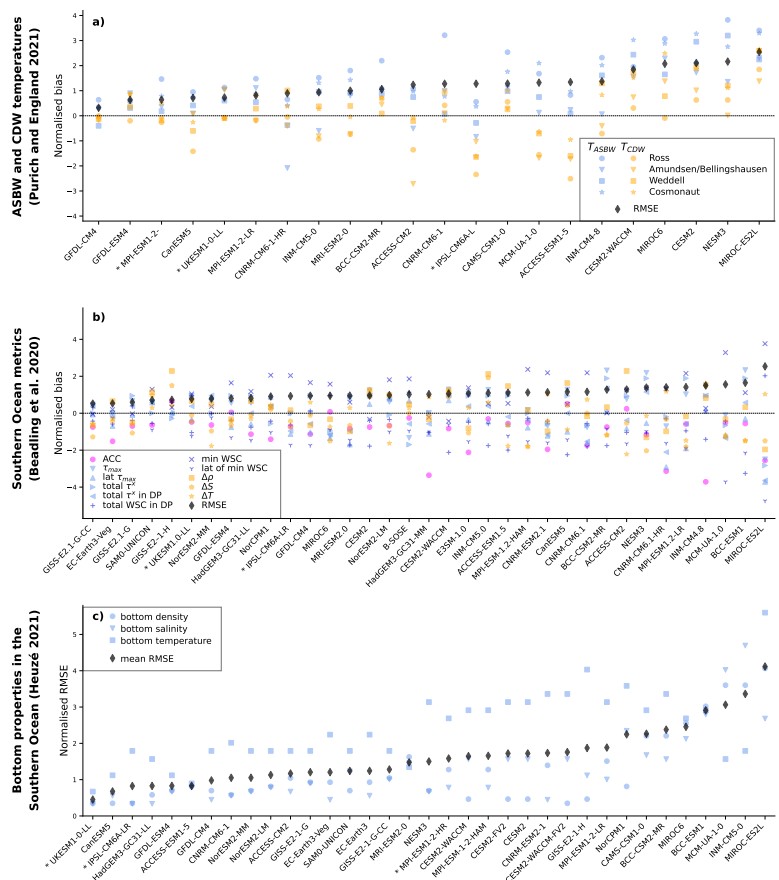

**Figure A1.** Assessment of Southern Ocean and Antarctic seas properties in the CMIP6 models. In each panel, CMIP models are ranked by increasing RMSE. (a) Antarctic Shelf Bottom Water (ASBW) and Circumpolar Deep Water (CDW) temperature biases compared to temperatures from Schmidtko et al. (2014) in the Ross, Amundsen/Bellingshausen, Weddell and Cosmonauts seas (Purich and England, 2021, their Fig. S16). (b) Biases in the Southern Ocean metrics defined in Tab. 1 of Beadling et al. (2020) with respect to the observational estimates describing several characteristics of the Antarctic Circumpolar Current, zonal wind stress strength, location and curl, as well as meridional gradients of water mass properties. (c) RMSE of bottom ocean properties (density, salinity and temperature) in the Southern Ocean (Heuzé, 2021, their Figs. 1, A1, A2). For each panel, the metric is normalised by the CMIP6 multi-model standard deviation. Selected models are labelled with a star.

**Table A1.** List of metrics used for ranking CMIP6 models. Evaluating period and observational dataset to which CMIP6 models are compared are indicated in the two last columns.

| Metric | Name | Unit | Period | Observations |
|---|---|---|---|---|
| **Purich and England (2021) - 22 CMIP6 models** | | | | |
| $T_{ASBW}$ | Temperature of Antarctic Shelf Bottom Water (ASBW) | °C | 1975-2012 | Schmidtko et al. (2014) |
| $T_{CDW}$ | Temperature of Circumpolar Deep Water (CDW) | °C | 1975-2012 | Schmidtko et al. (2014) |
| **Beadling et al. (2020) - 34 CMIP6 models** | | | | |
| $ACC$ | Volume transport of the Antarctic Circumpolar Current (ACC) through the Drake Passage | Sv | 1986-2005 | Donohue et al. (2016) |
| $\tau_{max}$ | Maximum zonally averaged zonal wind stress | $N.m^{-2}$ | 1986-2005 | ERA5 |
| lat $\tau_{max}$ | Position of the peak wind stress | $°S$ | 1986-2005 | ERA5 |
| total $\tau^{(x)}$ | Zonally averaged westerly wind stress | $10^{12}N$ | 1986-2005 | ERA5 |
| total $\tau^{(x)}$ in DP | Zonally averaged westerly wind stress over the Drake Passage latitudes (55°S-64°S) | $10^{12}N$ | 1986-2005 | ERA5 |
| total $WSC$ in DP | Integrated Wind Stress Curl (WSC) over the Drake Passage latitudes (55°S-64°S) | $10^{12}N.m^{-1}$ | 1986-2005 | ERA5 |
| min $WSC$ | Minimum zonally integrated Wind Stress Curl | $Nm^{-2}$ | 1986-2005 | ERA5 |
| lat of min $WSC$ | Latitude of minimum zonally integrated Wind Stress Curl | $°S$ | 1986-2005 | ERA5 |
| $\Delta\rho$ | Zonally and full-depth-averaged potential density difference between 65°S and 45°S | $kg.m^{-3}$ | 1986-2005 | WOA18 |
| $\Delta S$ | Zonally and full-depth-averaged salinity difference between 65°S and 45°S | (-) | 1986-2005 | WOA18 |
| $\Delta T$ | Zonally and full-depth-averaged potential temperature difference between 65°S and 45°S | $°C$ | 1986-2005 | WOA18 |
| **Heuzé (2021) - 35 CMIP6 models** | | | | |
| bottom density | bottom density | $kg.m^{-3}$ | 1985-2014 | WOA18 |
| bottom salinity | bottom practical salinity | (-) | 1985-2014 | WOA18 |
| bottom temperature | bottom potential temperature | $°C$ | 1985-2014 | WOA18 |

## Appendix B: Atmospheric and oceanic components of the internal climate variability in multiple CMIP6 models

In Fig. B1 we briefly show where the three selected CMIP6 models sit in terms of internal climate variability. Based on the analysis of 15 CMIP6 models with more than 10 members, the multi-member standard deviation of 2 m air temperature over the whole Antarctica varies between 0.06°C and 0.24°C. Both IPSL-CM6A-LR and MPI-ESM1.2-HR have a variability close to the median (0.12 and 0.13°C), while the UKESM1-0-LL model is among the models with the highest variability (0.20°C). For the ocean, the multi-member standard deviation of the ocean temperature averaged between 200 and 700 m over the continental shelf varies between 0.02°C and 0.12°C. The MPI-ESM1.2-HR model is one of the models with the lowest ocean variability (0.02°C), UKESM1-0-LL is close to the median (0.04°C) and IPSL-CM6A-LR is one of the models with the highest variability (0.07°C).

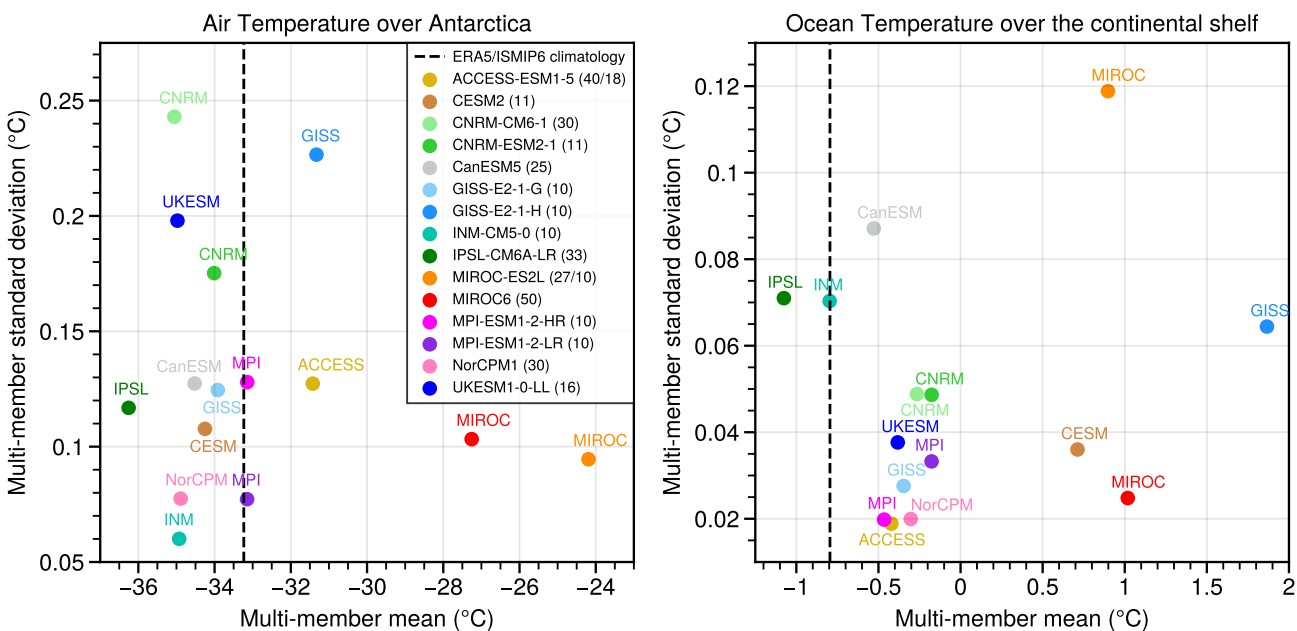

**Figure B1.** Assessment of 1995-2014 multi-member mean and standard deviation of Antarctic air temperature at 2 m (left) and circum-Antarctic ocean temperature between 200 and 700 m depth on the continental shelf (right) in 15 CMIP6 models. The number of members for each model is in brackets. When two numbers are indicated, they correspond to the available members for the atmosphere and ocean, respectively. The black dashed lines represent the observational means, from the ERA5 atmospheric reanalysis (Hersbach et al., 2020) and from the ISMIP6 observational ocean climatology (Jourdain et al., 2020).

## Appendix C:  Variables and number of members included in the various analyses performed in this study

The following table details (column 1) the variables studied for each of the analyses carried out in this paper, i.e., the studies of oceanic and atmospheric internal climate variability and the study of the impact of internal climate variability on projections of Antarctic contribution to sea level by 2100 under the SSP2-4.5 scenario, (column 2) the origin of the variables, (column 3) the scenario considered for the CMIP6 outputs, (columns 4-6) the number of members considered for each analysis for each of the three selected CMIP6 models.

For projections of the dynamical contribution of the ice sheet to sea level, we have not used all the members available for the SSP2-4.5 scenario due to the numerical cost of simulations. In addition to the first member, the selection is made over the current period (1995-2014 means) to cover the widest range of values for the ocean temperature on the continental shelf in the Amundsen Sea. In total, we run 11 simulations, five with the IPSL-CM6A-LR model (r1i1p1f1, r3i1p1f1, r6i1p1f1, r11i1p1f1, r25i1p1f1, see the CMIP6 naming convention in https://goo.gl/v1drZl), four with the UKESM1-0-LL model (r1i1p1f2, r2i1p1f2, r4i1p1f2, r8i1p1f2), and only two with the MPI-ESM1.2-HR model (r1i1p1f2, r2i1p1f2) given that its oceanic variability is very low (see section 3.1) and the number of available members for the SSP2-4.5 scenario very limited.

**Table C1.** Data used for the three analyses carried out: studies of (i) oceanic and (ii) atmospheric internal climate variability and (iii) study of the impact of internal climate variability on projections of Antarctic contribution to sea level by 2100 under the SSP2-4.5 scenario. The second column describes the origin of analysed variables, the third column describes the CMIP6 output scenario, the last columns describe the number of members used for each of the three selected CMIP6 models. (A) indicates that all available members have been used while (NA) indicates that a limited number of available members have been used for computational cost reasons.

| Variable | Origin | scenario | MPI-ESM1.2-HR members | IPSL-CM6A-LR members | UKESM1-0-LL members |
|---|---|---|---|---|---|
| **Oceanic internal climate variability** | | | | | |
| temperature | CMIP6 outputs | historical | 10 (A) | 33 (A) | 16 (A) |
| salinity | CMIP6 outputs | historical | 10 (A) | 33 (A) | 16 (A) |
| **Atmospheric internal climate variability** | | | | | |
| surface mass balance | emulation of regional climate model behaviour driven by CMIP6 outputs | historical | 10 (A) | 33 (A) | 16 (A) |
| air temperature | CMIP6 outputs | historical | 10 (A) | 33 (A) | 16 (A) |
| sea-level pressure | CMIP6 outputs | historical | 10 (A) | 33 (A) | 16 (A) |
| precipitation | CMIP6 outputs | historical | 10 (A) | 33 (A) | 16 (A) |
| **Projection until 2100 under SSP2-4.5 scenario** | | | | | |
| SMB contribution | emulation of regional climate model behaviour driven by CMIP6 outputs | SSP2-4.5 | 2 (A) | 11 (A) | 17 (A) |
| dynamical contribution | Elmer/Ice simulation driven by oceanic CMIP6 outputs | SSP2-4.5 | 2 (A) | 5 (NA) | 4 (NA) |
| total contribution | sum of the two previous contributions for a given member | SSP2-4.5 | 2 (A) | 5 (NA) | 4 (NA) |

 **Appendix D: Assessment of oceanic internal climate variability for 60-year period**

In Fig. D1, we repeat the analysis that was presented in the main text and illustrated in Fig. 2, but analysing the multi-model variability of 60-year means (1955-2014) instead of 20-year means (1995-2014). This emphasises that substantial internal climate variability is still present in 60-year averages.

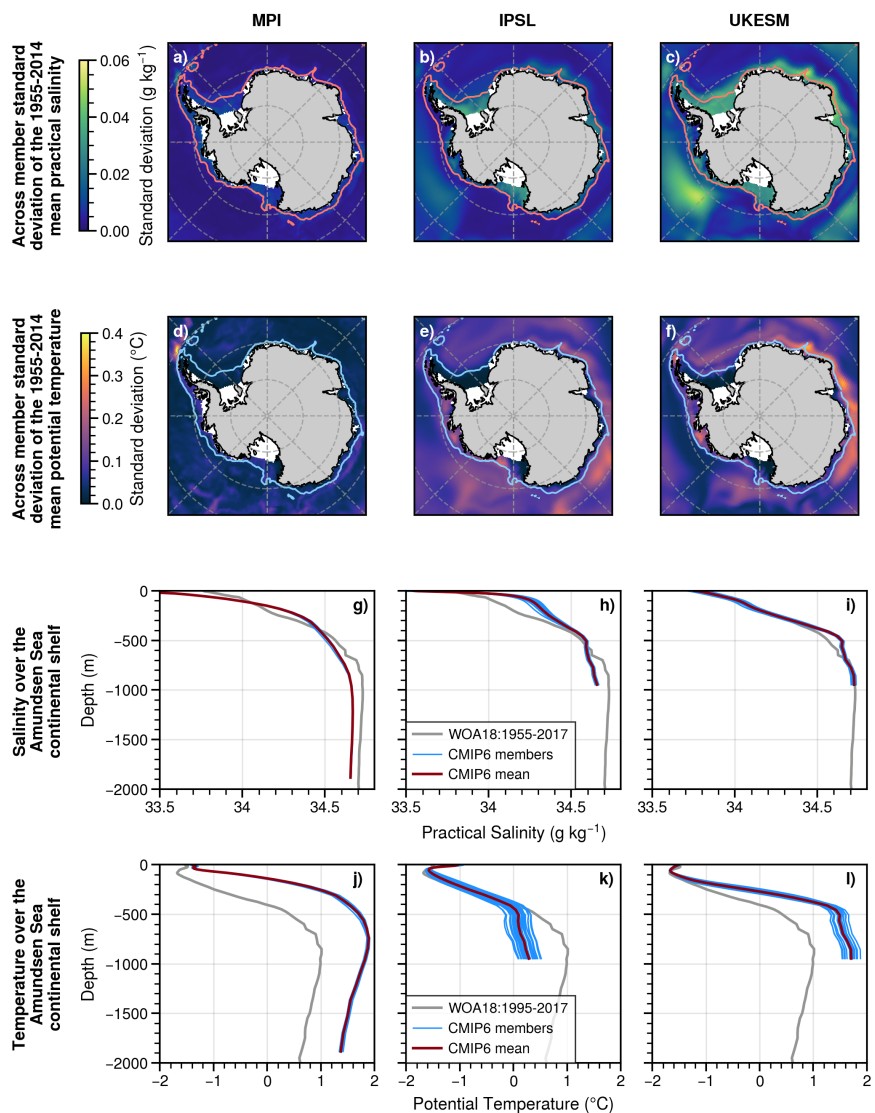

**Figure D1.** Comparison of the saline and thermal properties of the CMIP6 models MPI-ESM1.2-HR (left), IPSL-CM6A-LR (middle) and UKESM1-0-LL (right). (a-c) standard deviation of the 1955-2014 mean practical salinity across the ensemble relative to the multi-member mean, considering the salinity averaged from 200 m to 700 m depth. The 1500 m isobath (pink) delimits the continental shelf. (d-f) same as (a-c) but for potential temperature and with the 1500 m isobath in blue. (g-i) mean vertical profiles of practical salinity on the Amundsen Sea continental shelf (as defined in Caillet et al., 2023). For each model, the blue curves represent the individual members (1955-2014 mean), and the red line the multi-member mean. The grey curve corresponds to the 2018 World Ocean Atlas data (WOA 2018, Boyer et al., 2018) over the period 1955-2017. (j-l) same as (g-i) but for potential temperature.

## Appendix E: Atmospheric mean properties

Here we evaluate the mean atmospheric state of the three selected CMIP6 models in comparison to the ERA5 atmospheric reanalysis (Hersbach et al., 2020).

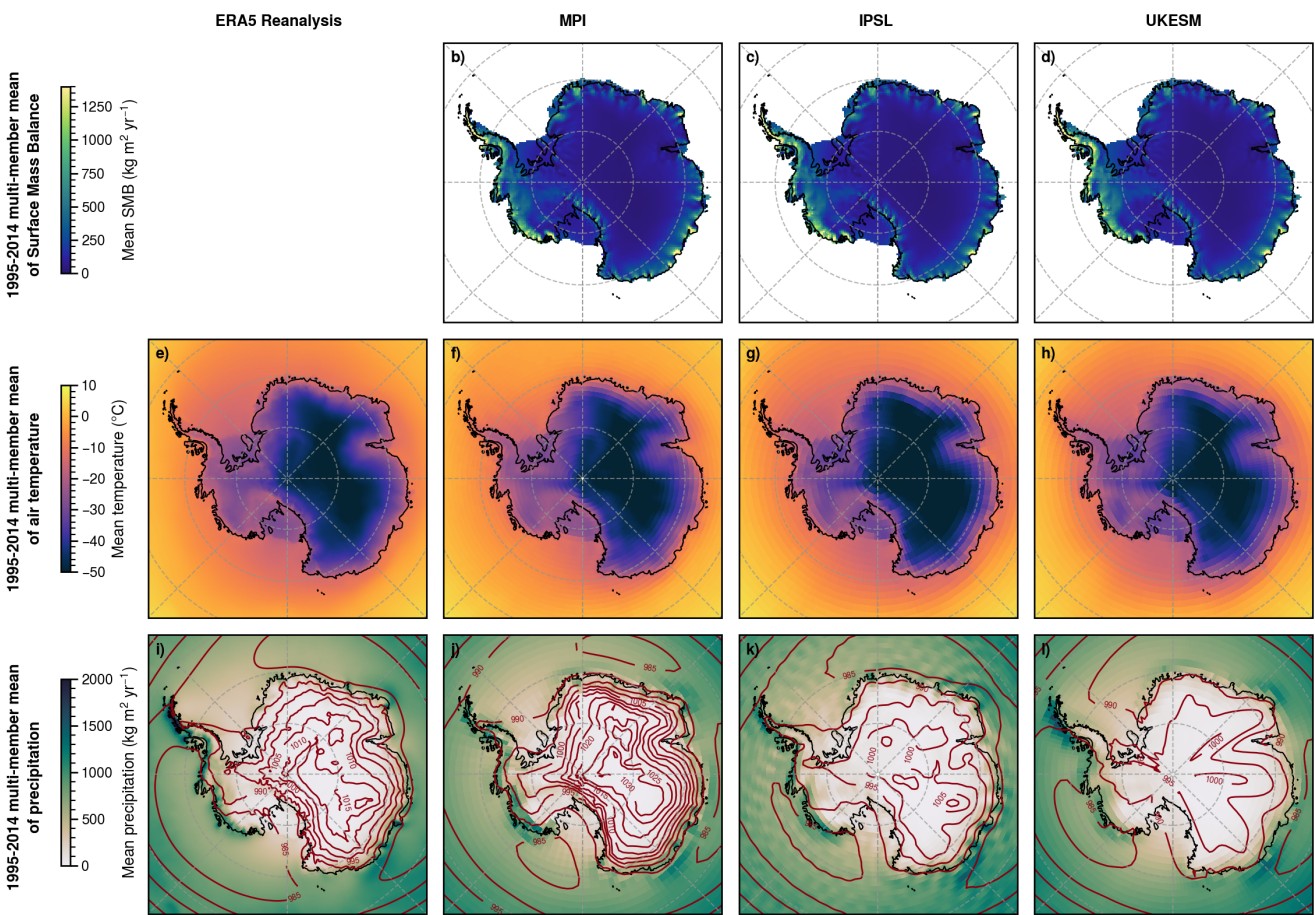

**Figure E1.** Comparison of mean surface mass balance, air temperature, precipitation and sea-level pressure from the ERA5 reanalysis (left) and the three CMIP6 models: MPI-ESM1.2-HR (middle left), IPSL-CM6A-LR (middle right) and UKESM1-0-LL (right). (e) 1995-2014 mean air temperature at 2 m from ERA5 reanalysis dataset (Hersbach et al., 2020). (i) same as (e) but for total precipitation (shaded) and sea-level pressure (contours every 5 hPa). (b-d) 1995-2014 mean emulated surface mass balance (SMB) across the climate model ensemble. (f-h) same as (b-d) but for air temperature at 2 m. (j-l) same as (b-d) but for total precipitation (shaded) and sea-level pressure (contours every 5 hPa).

## Appendix F: Best member ranking

Given the importance of internal climate variability, it is tempting to select the member that is most in phase with the observational record. Here we calculate the following metrics for individual members:

- Root mean squared difference between the multi-year mean ocean temperature profiles measured and modelled in front of Pine Island (years 1994, 2000, 2007, 2009, 2010, 2012, as given in Dutrieux et al., 2014) and Dotson (years 2000, 2006, 2007, 2009, 2011, 2012, 2014, as given in Jenkins et al., 2018) ice shelves. This is a proxy for the phase of multi-decadal variability in the region where the ocean has triggered the largest ice-sheet mass loss.

- Root mean squared difference between the standard deviation of multi-year ocean temperature profiles measured and modelled in front of Pine Island and Dotson ice shelves. This is a proxy for the amplitude of multi-decadal variability in the region where the ocean has triggered the largest ice-sheet mass loss.

- Difference between observed and modelled trend in the Southern Annular Mode (SAM), estimated over 1965-2014 based on the index defined by Marshall (2003). SAM affects both the Antarctic SMB (Medley and Thomas, 2019) and ice-shelf basal melting (Verfaillie et al., 2022). Here we evaluate the phase of multi-decadal variability in individual members by quantifying the modulation of the SAM 60-year trend by internal climate variability.

- Pearson correlation coefficient, root mean squared difference between and standard deviation of the observed and modelled Southern Annular Mode index (SAM, Marshall, 2003) with a 5-year running window on detrend data over 1965-2014. SAM index is based on the zonal pressure difference between the latitudes of 40°S and 65°S and is a proxy for the phase and amplitude of inter-annual variability over all Antarctica as SAM has a large impact on atmospheric and oceanic circulations. Taylor diagram combines these 3 metrics to quantify the degree of correspondence between modelled and observed SAM.

- Pearson correlation coefficient, root mean squared difference between and standard deviation for the observed and modelled Tripole Index for the Interdecadal Pacific Oscillation (TPI, Henley et al., 2015) with a 5-year running window over 1854-2014. The TPI is based on the difference between the Sea Surface Temperature Anomalies (SSTA) averaged over the central equatorial Pacific and the average of the SSTA in the Northwest and Southwest Pacific. The TPI describes decadal to interdecadal changes in the strength of the El Niño–Southern Oscillation (ENSO) and its teleconnections. ENSO affects the West Antarctic SMB (Genthon and Cosme, 2003; Scott et al., 2019) and ice-shelf basal melting in the Amundsen Sea (Steig et al., 2012; Holland et al., 2019) through the south-eastward propagation of atmospheric Rossby waves from the inter-tropical Pacific. Taylor diagram combines these 3 metrics to quantify the degree of correspondence between modelled and observed TPI.

- Comparison of the mean ocean temperature at 750 m depth on the continental shelf in the Amundsen Sea between identified warm periods and the preceding cold periods. Three warm periods have been identified in observations: 1945±12 (1933-1957), 1970±4 (1966-1974) (based on sediment records, Smith et al., 2017) and 2006-2012 (based on Dotson and

Pine Island melt rates estimates, Dutrieux et al., 2014; Jenkins et al., 2018). These periods are compared respectively to 1850-1932, 1958-1965 and 1975-2005, supposed to be colder periods. For each member, we assume that the warm period exists if a 5-year mean, at least, within years that define the warm period is higher than the mean of the preceding cold period. This is a proxy for the phase of multi-decadal variability in the region where the ocean has triggered the largest ice-sheet mass loss.

– Root mean squared difference between observed and modelled sea-ice concentration trend around Antarctica over 1979-2014 (version 4, Meier et al., 2022). Here, we evaluate the phase of multi-decadal variability in individual members by quantifying the modulation of sea-ice concentration trend by internal climate variability (Zhang et al., 2019).

– Root mean squared difference between the 1995-2014 average SMB output of the MAR simulations forced by ERA5 described in Kittel et al. (2021) and the reconstructed SMB of each member of the IPSL-CM6A-LR model on the same period as described in Jourdain et al. (2024).

We then rank the performance of all the members by assigning them a rank, with lower rank for the member that best matches the observations:

– for root mean squared difference (RMSE) metric, the member with the lowest (respectively highest) RMSE value is assigned rank 1 (respectively rank 33).

– for metrics relative to Taylor Diagram (Pearson correlation coefficient and standard deviation), we first calculate a rank for each individual metric by assigning the lowest rank value to the lowest Pearson correlation coefficient and to the lowest difference between observed and modelled standard deviation. We then average all calculated ranks and finally assign the best (worst) final rank to the lowest (highest) average.

– for metric relative to warm and cold period alternation, we assume that the warm period exists if a 5-year mean, at least, within years that define the warm period is higher than the mean of the preceding cold period. If the condition is met, the member is assigned the value 1, and 0 otherwise. This process is applied to each of the three warm periods and the values are then summed. Members with a value of 3 (of 0) are assigned the best (the worst) rankings.

The ranks of all members for all metrics are presented in Fig. F1 for IPSL-CM6A-LR and UKESM1-0-LL models.

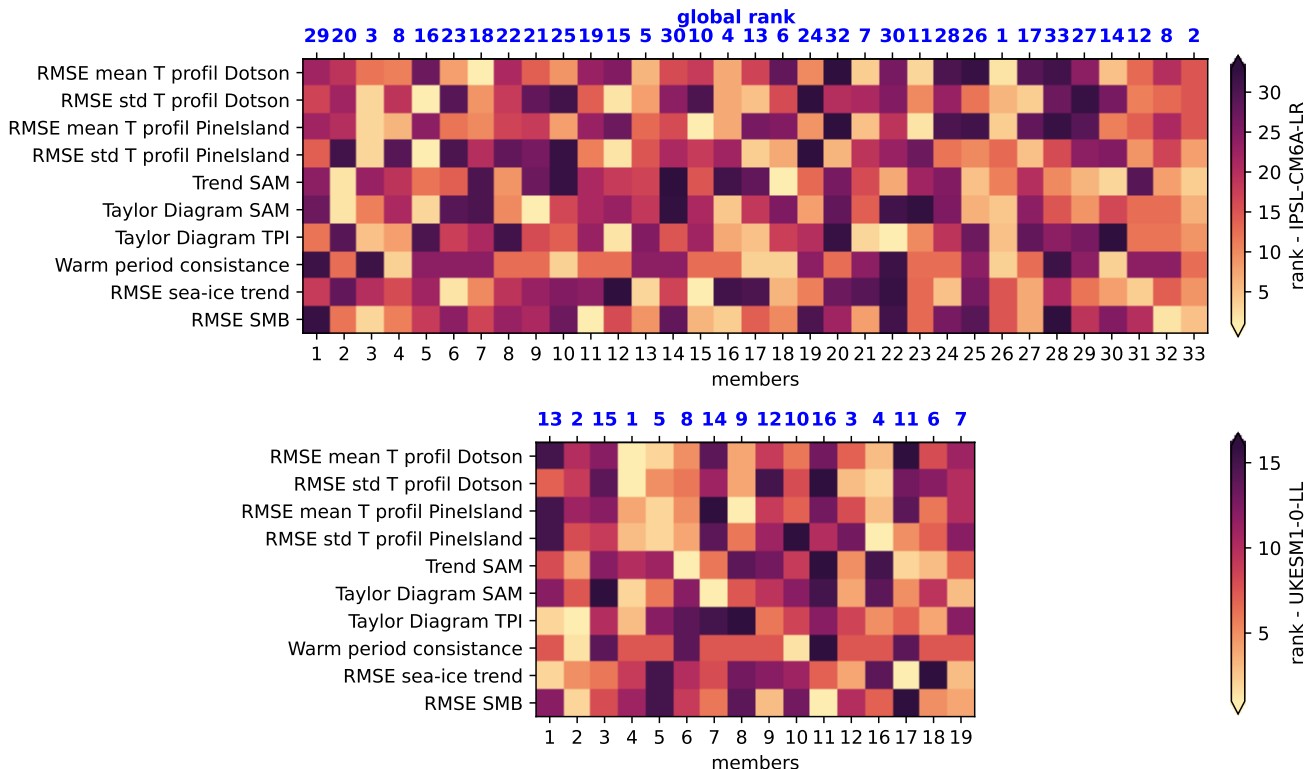

**Figure F1.** Combination of relevant metrics for evaluating members of the IPSL-CM6A-LR model (top) and UKESM1-0-LL model (bottom) compared to various observations/index data. The y-axis represents the selected metrics while the x-axis represents individual members. A rank of 1 (respectively 33 or 16) refers to the member with the value closest to (respectively furthest from) the assessed observational value. The blue numbers at the top indicate the member's overall ranking, with equal weight for all metrics.

*Code and data availability.* The ice-sheet model version, set of parameters used to run our experiments and sea-level contribution outputs are provided in https://zenodo.org/records/14417535 and https://doi.org/10.5281/zenodo.14393484. Simulations are driven by (i) oceanic CMIP6 outputs interpolated and extrapolated to the ISMIP6 stereographic grid through the tool developed by Nicolas Jourdain, available in https://doi.org/10.5281/zenodo.12755909 and (ii) SMB resulting from the emulation of the behaviour of a regional climate model driven by atmospheric CMIP6 outputs. The tool used to emulate the ensemble of regional climate simulations is available in https://doi.org/10.5281/zenodo.13756239. The python scripts used to build the figures are provided in https://doi.org/10.5281/zenodo.7436688 and are mainly based on the Xarray (Hoyer and Hamman, 2017), Numpy (Harris et al., 2020) and Matplotlib (Hunter, 2007) packages.

*Author contributions.* JC and NCJ designed the overall study. JC wrote the initial draft and NCJ did the first review. JC ran the Elmer/Ice experiments with valuable inputs from PM, FGC, BU and MC. NCJ compiled the intercomparison of CMIP6 ocean models. NCJ, CA and CK calculated the multi-member surface mass balance. JC and CB worked on the basal melting parameterisation. All authors contributed to the final manuscript.

*Competing interests.* The authors declare no competing interests.

*Acknowledgements.* This study was funded by the French National Research Agency (ANR) under grants ANR-19-CE01-0015 (EIS) and ANR-22-CE01-0014 (AIAI). The work also benefited from the support of the French Government through the France 2030 program managed by ANR (ISClim, ANR-22-EXTR-0010). Some of the authors received funding from the European Union's Horizon 2020 research and innovation programme under Grant agreements 820575 (TiPACCs), 101003536 (ESM2025) and 101003826 (CRiceS).

This work was granted access to the high-performance computing (HPC) resources of TGCC under allocations A0140106066 and A0140106035 attributed by GENCI. The CMIP6 data were analysed on the ESPRI-MOD platform thanks to the CLIMERI-France infrastructure.

The authors would like to acknowledge David Ferreira for interesting discussion on sea-ice, Jérome Servonnat for his valuable help on ESPRI-MOD as well as Hélène Seroussi, Gaël Durand, Jérémie Mouginot and Hartmut Hellmer for their constructive feedbacks.

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
