# Peer review of "Uncertainty in the projected Antarctic contribution to sea level due to internal climate variability"

_EGUsphere, 2024_

## Referee Comment (RC1)

**Review of manuscript egusphere-2024-128 "Uncertainty in the projected Antarctic contribution to sea level due to internal climate variability" by J. Caillet et al.**

General comments:

The submitted manuscript investigates the impact of internal climate (oceanic and atmospheric) variability in projections of Antarctica's sea-level contribution until year 2100. For this purpose the authors run standalone ice-sheet simulations applying output from a selection out of ensemble simulations of three CMIP6 climate models. Besides quantifying the relevance of the internal climate variability for sea-level projections, the authors also give recommendations for future ice-sheet projections. I deem the study a valuable contribution to the Earth System modeling community, especially the ice-sheet modeling community.

I find the manuscript clearly written, well structured and mostly understandible. The figures illustrate the findings well. The methodology seems consistent and the conclusions plausible. I would like to mention that my assessment is limited regarding some oceanic mechanisms described in the study (which I refer to in my specific comments) and details of the applied metrics in Sect. 4.3. They seem plausible but I didn't have the time to dig deeper into the details.

I would support the publication of the manuscripts after my few points below have been addressed.

Specific comments:

The last sentence of the abstract seems a bit detached from the rest of the abstract. Maybe a different introduction of the sentence would help for a smoother reading.

L28: Is there a typo: "of" instead of "on"?

L50: Why SPP2-4.5?

L56: drivers

L82: Which friction law is it? I don't see the necessity to write the law down here but a reference to the equation would be helpful.

L120: What's the sense of these abbreviations?

L164-166: I am not able to follow the causal chain. Which location do the authors mean by "there" at the end of the sentence (I guess the Eastern Ross Sea)? I am not an expert on such oceanic mechanisms/patterns and personally would be glad to get a clearer explaination.

L179: Again, as a non-specialist regarding the ocean: What does it mean when you state "Both IPSL-CM6A-LR and UKESM1-0-LL seem to be prone to convection"?

L224-225: "On average, the amplitude of SLC variability relative to the atmosphere is 3.4 times higher than that relative to the ocean." This sentence is not entirely clear to me and I would appreciate if the authors could briefly explain what is exactly meant. How is this finding deduced? Is it shown in a figure which could be referenced here?

L258-259: How is this finding deduced? If I am right it can be seen from Fig. 6a?

---

## Referee Comment (RC3)

**Review of manuscript egusphere-2024-128**
'*Uncertainty in the projected Antarctic contribution to sea level due to internal climate variability*' by J. Caillet et al.

**Summary**

The study by Caillet et al. quantifies the uncertainties in the projected Antarctic contribution to sea-level change by 2100 related to internal climate variability in a subset of CMIP6 models. Three CMIP6 models are selected based on a summary of previous evaluations (Purich and England, 2021; Beadling et al., 2020; Heuzé, 2021; Sect. 2.1). The Antarctic sea-level contribution is projected with the stand-alone ice-sheet model Elmer/Ice; the respective experimental setup is presented in Sect. 2.2 and Sect. 2.3.

First, internal climate variability in the selected CMIP6 models is explored (Sect. 3.1 and Sect. 3.2). Then, the Antarctic contribution to sea-level change projected by Elmer/Ice based on different ensembles members for the selected CMIP6 models is presented (Sect. 3.3).

The authors quantify the effect of internal climate variability on the Antarctic sea-level contribution by the end of this century with 45% to 93%, with a higher impact from atmospheric variability compared to ocean variability, and modulated by the CMIP6 model. Results are discussed in terms of the robust representation of internal climate variability in CMIP models (Sect. 4.1), the internal climate variability as a source of uncertainty in Antarctic sea-level projections (Sect. 4.2), and identifying best ensemble members as an alternative approach to account for internal climate variability in sea-level projections (Sect. 4.3). The authors conclude with general recommendations for future assessments of the Antarctic contribution to sea-level change (Sect. 5).

**General comments**

By bringing together internal climate variability and the future evolution of the Antarctic Ice Sheet, the paper addresses a relevant and scientific interesting question, that has rarely been explored in previous assessments of the future trajectory of the Antarctic Ice Sheet. The presented results and related discussion may be valuable for future assessments of the Antarctic contribution to sea-level rise.
The title clearly reflects the contents of the paper. The abstract provides a concise and complete summary. Overall, the study has a sound methodology and experimental setup. In some cases, the description of the methods could be more precise, and the clarity of language improved. While the results span a wide range from exploring the representation of internal climate variability in selected CMIP6 models to the modelled response of the Antarctic Ice Sheet, the manuscript may benefit from linking both in greater depth (e.g. explaining the climate model - dependence of the impact of climate variability on the projected sea-level contribution, ranging between 45% and 93%, if possible). Here, additional figures, e.g. in a Supplementary Material, may be helpful for the reader. In addition, the discussion on a possible selection of best ensemble members from CMIP models could be better integrated in the manuscript. Finally, some additional explanations may be needed to directly derive and support some of the recommendations given in the conclusion based on the results presented in this study.

I have included more specific comments, questions and suggestions below.

**Specific comments**

L12-14: In the abstract, the results of the sea-level projections are summarized before describing the internal climate variability in the CMIP6 models. Maybe it would be more intuitive to follow the same order as in the main text (that is, internal climate variability in CMIP6 models followed by sea-level projections)?

L12-14: Maybe a brief remark on the upper end of the amplitude of oceanic internal variability covered by different climate models could be added, in addition to the mentioned and explained weak mid-depth ocean variability?

L15: Please specify 'use of several members in the run and its initialisation'. I think I understand what is meant here after reading the manuscript but this phrase may be unclear for the reader when starting with the abstract.

L24: Maybe add 'estimates of' or 'projections of', e.g. 'Estimates of the AIS contribution to future sea level rise are currently based...'

L25: I think CMIP stands for Coupled Model Intercomparison Project. Please check.

L31-35: In this paragraph climate variability is introduced as consisting of two components (1) variability from natural and anthropogenic external forcings and (2) internal variability, and explanations for these components are given, after having referred to internal climate variability in the previous paragraph. Maybe some restructuring is possible to define internal climate variability with its first use?

L49/50: Maybe 'the *Antarctic* Sea Level Contribution'?

L50: Why did you chose the SSP2-4.5 emission pathway? Please add a short explanation either here or in Sect. 2.3.

L56: 'drivers' instead of 'driver'?

L58-64: Please specify the properties that are used to evaluate the CMIP6 models. Some of them are given in Figure 1 or its caption, but it may be helpful to also include them in the main text.

  - What properties of ASBW and CDW are evaluated? Please add this information also in the main text.
  - Many dynamical features for the Southern Ocean are listed in the legend of Figure 1b. It might be helpful for the reader to better link the legend and caption of Figure 1 to the description in the main text. This applies to e.g. the ocean properties that are evaluated in terms of their meridional gradients.
  - What bottom properties of the Southern Ocean are evaluated? Maybe add this information also in the main text.

L58-64: To facilitate readability, bold or italic fonts for some phrases in this list could be used (e.g. for the evaluated water masses). As an alternative, these properties could be given in a table rather than in a list.

L65: Is the assessment presented in Figure 1 based on one ensemble member of the respective CMIP6 models or an average over all available ensemble members? As different

ensemble members are used later in the manuscript, maybe add this information here (or in the figure caption) to avoid confusion.

L66: How is 'best' defined? Does UKESM-1-0-LL have one of the lowest RMSE in all three studies? Maybe state this more explicitly here.

L68: If I understand Figure 1 correctly, MPI-ESM1.2-HR was evaluated in two of three studies (red triangles in Figure 1a and c). Please check.

L70-73: This is a very general sentence, in particular for readers not familiar with the representation of ice-shelf melting in CMIP models. What is meant by '*some kind of* prescribed ice-shelf melting at depth'? Does this impact the assessment / ranking of CMIP6 models in Figure 1? I think it may be helpful to briefly discuss the link between the treatment of ice-shelf melting and the CMIP6 model assessment, if this information is mentioned here.

L74-77: Please add more detail on the assessment of atmospheric properties for the CMIP5/6 models in the manuscript, also given that Agosta et al. (2022) refers to a conference abstract. For example, which atmospheric properties have been evaluated and which method is used for the assessment?

Figure 1: Please consider marking the selected CMIP6 models in a different way, e.g. by colouring the model name or adding a box around the model name. The red triangles can be easily confused with the other markers (or appear within the legend, compare Figure 1b).

Figure 1: Please briefly introduce the abbreviations used in the legend, e.g. in panel b in the figure caption and / or in the main text (L58-64).

L82: Please add a reference for the friction law.

L84: What do you mean by '*preferentially* refined'? Please specify.

L85: What do you mean by 'high *curvatures*'? Please clarify.

L100-103: Are the ocean temperature corrections to match observed melt rates also based on Reese et al. (2023)? I assume that these may differ from the corrections presented in Reese et al. (2023) given the use of a different ocean climatology here. Please describe how the temperature corrections applied here are derived. It may also be helpful for the reader to briefly mention why temperature corrections are applied (instead of e.g. changing the PICO parameters to match present-day observed melt rates).

L106-107: Why is a *10 %* reduction of the inverted friction coefficients applied? Is this based on testing, a 'best fit' or some other methodology? Does the reduction of the friction coefficients change the modelled velocities (as this quantify has been the target of the inversion)?

L107-108: The ice-sheet model configuration slightly overestimates mass loss in West Antarctica (when compared to the uncertainty ranges of the observations) if I understand Table 1 correctly.

L108-109: Can you maybe add a brief remark (or a figure) on how large the trend bias in the ice-sheet model setup is?

L109-110: Please specify the reference that your results (in terms of the Antarctic sea-level contribution) are compared to. Are the projections in response to the CMIP6 climate models

analysed relative to each other? Do you substract from a control experiment to remove the drift? After reading the discussion, I think the trend is not removed.

L112-114: This formulation might be confusing for some readers.

Figure 2: It may be helpful to indicate the most relevant ice shelves in a map (as already done for the Antarctic basins in Figure 7).

L115-116: I got confused by the focus on the ocean here. Do you also run projections with ocean forcing only?

L116: I am not sure if I understand what is meant by 'constrained'. Maybe consider replacing by e.g. 'driven' or 'forced', if applicable.

L118-122: Please add more details on the selection of the CMIP6 ensembles members. I am not sure which section you are referring to for additional information on the selection, based on covering a wide spread in (1) possible ocean temperatures in the Amundsen Sea Embayment and (2) surface mass balance. Is the focus on the Amundsen Sea Embayment motivated by observed present-day mass loss in this region? Why is a different number of ensemble members chosen for each CMIP6 model? I appreciate the assessment of CMIP6 models in Figure 1, but, if I understand correctly, this evaluation justifies the choice of the CMIP6 model rather than the individual ensemble members for driving Elmer/Ice. It might be helpful for the reader to stress the link between the CMIP6 model evaluation, the assessment of the internal climate variability for these CMIP6 models and the selection of a subset of ensemble members for driving Elmer/Ice.

L120-122: I am not sure how familiar readers are with the CMIP variant labelling. While it may not be necessary to explain it in full detail, it may be helpful to briefly state that these lists describe different ensemble members for each of the CMIP6 models.

L129-135: This paragraph could be shortened. Maybe detailed information on the SMB in ISMIP6-Antarctic (L129-130) is not needed here.

L135: Maybe 'constrain' could be replaced by 'drive' or something similar, if applicable.

L136-144: I would like to mention that my comments are limited to this manuscript, and I have not assessed the approach for emulating MAR and thus for obtaining the estimates of SMB used in this study. From my point of view, no detailed evaluation is needed here and it is fine to refer to the approach described in Jourdain et al. (2024, in discussion) as done. Please make sure that respective inputs and outputs of this approach become clear (see some of the following comments / questions).

L137: '*surface* melting' instead of 'melting'?

L140-141: I am not sure if I understand correctly how the SMB for a given member is estimated. What is meant by '*perturbed* as a function of the annual temperature difference'?

L150: Maybe replace '*a* subset' by '*the* subset'?

L150: 'two first subsections' could be replaced by directly stating the subsections that you would like to refer to here to improve readability.

L153-155: It might be helpful for the reader to explicitly state the ocean properties that reflect the oceanic internal climate variability in the beginning of this section (that is, salinity and

temperature, as shown in Fig. 3, and as eventually described in the beginning of the following paragraph starting in L159).

L155: What is meant by '*typical*' standard deviation across model members? Does this mean that in most regions values are around 0.017 g kg-1 and 0.07°C for MPI-ESM1.1-HR? Or are these typical values for CMIP6 models?

L159: 'continental shelf' instead of 'shelf'?

L161: Maybe replace '*largest* variability' by '*large* variability' to avoid confusion? If I understand correctly, for example, the highest variability in mid-depth salinity for UKESM1-0-LL is found around Prydz Bay.

L171: Is 'deep ocean' considered as same ocean depth as 'mid-depth'?

L173: I would like to suggest to replace 'ice-sheet mass loss' by '*present-day* ice-sheet mass loss' or something similar.

L184-190: I think it may be helpful to add figures on the assessment of oceanic internal climate variability based on 60-year averages, at least in form of a Supplementary Material, given that the discussion and recommendations reflect on the time period of averaging.

L197: Please specify that the increased water vapour saturation in warmer air then results in enhanced precipitation.

L200: Is the SMB that you refer to here emulated or directly derived from the CMIP6 models? According to the caption of Figure 5 it is based on the MAR emulation. Maybe this could be specified again also in the main text.

L202: 'consistent' instead of 'consistently'?

L202 - 205: This section also refers to the absolute SMB. Since the atmosphere is suggested as an important factor for the (spread in the) projected Antarctic sea-level contribution in the following section, and the choice of the CMIP6 model at the same time also modulates the projected sea-level change, I would like to suggest to add a related figure of SMB and atmospheric temperatures (e.g. in the Supplementary Material) for interested readers.

L204: 'which is both due to' instead of 'which is due both'?

L205: MPI-ESM1.2-HR also shows a relatively high standard deviation in atmospheric temperature in the Siple Coast region, compared to the other selected CMIP6 models. Is this relevant for the projected future evolution of the Antarctic Ice Sheet? In Figure 7, the ice-sheet response in basin 9 (Siple Coast) to atmospheric changes in MPI-ESM1.2-HR (showing mass loss) differs from the other CMIP6 models (showing mass gain).

L205-208: What are the typical characteristics of the two Pacific-South American modes? Maybe add a short summary here for readers that are not familiar with Wang et al. (2022) and Marshall and Thompson (2016).

L210-227: Maybe the link between the analysis of internal climate variability in CMIP6 models and the projected Antarctic sea-level contribution could be stressed here, in particular, for explaining some of the key results related to the uncertainties in the projected contribution of the Antarctic Ice Sheet to sea-level change. This includes, for example, the results that (1) atmospheric internal climate variability has a larger effect on the spread in the projected sealevel change with Elmer/Ice than oceanic internal climate variability, (2) the impact of the choice of the CMIP6 model on the sea-level contribution from Antarctica, and (3) the similarity of atmospheric internal climate variability for the selected CMIP6 models.

L213: For MPI-ESM1.2-HR there is a mean (?) mass loss related to the atmosphere in West Antarctica (Fig. 6i).

L215-217: Do you meant to refer to 'Pine Island and Thwaites *ice shelves*' / 'Getz ice *shelf*' here or rather the respective basins?

L217-218: Is this drift of the unforced Elmer/Ice experiment removed or is the absolute Antarctic sea-level contribution given in the respective figures? I think I got confused by the statement in L109-110 (please also see my related previous comment). And can the influence of the drift on the trends in East Antarctica be quantified?

L217: I would like to suggest to replace 'contaminated' by 'influenced' (or something similar).

L218: What can be learned on the sensitivity of East Antarctica and the Antarctic Peninsula to internal climate variability based on the simulations presented here? Maybe you can make use of Figure 7 and add some details in this section.

L220-222: I would like to suggest to give the full name of the CMIP6 models throughout the whole manuscript (consistent with e.g. Sect. 3.1).

L223: Basin 5 (including Totten glacier) shows a relatively large spread in the dynamical sea-level contribution (Fig. 7b). Can this be related to the assessment of oceanic internal climate variability in Sect. 3.2?

L226-227: Please add more information on this finding. How is the number determined? Can it be seen in a figure (likely Figure 6)?

Figure 6: I assume that the number in brackets in the legend refers to the number of ensemble members for each CMIP6 model. Why do the numbers differ between panel a/b and panel c? Please check.

Figure 6: Does the solid line indicate the mean? Maybe I have missed this.

L232-233: Maybe specify which paleoclimate proxies are used in Parsons et al. (2020) (similar to stating that Casado et al. 2023 base their analysis on ice core reconstructions in the following paragraph), for readers that are not familiar with this study?

L232: 'global mean surface air temperature' or its variability?

L233: Maybe you could add the observational plausible range for the temperature variability for comparison with the values for the CMIP6 models?

L243: If possible, maybe a conclusion on the representation of atmospheric variability in CMIP models could be added, bringing together the results of this study (Sect. 3.1) with the previous literature?

L245: Maybe add a reference for these observations?

L258-259: As the choice of the CMIP6 model is suggested to have a similar impact on the Antarctic sea-level contribution as the internal climate variability, it may be helpful to add a short paragraph on this finding also in Sect. 3.3 (in addition to this statement in the discussion).

L265-266: This sentence can maybe be reformulated. As already indicated, given the limited impact of the emission pathway on the Antarctic sea-level contribution to 2100, SSP2.4.5 may not be the main explanation for the Elmer/Ice projections presented here being at the lower end of previous projections.

L269: I would like to suggest 'ocean-*induced* melting' (or something similar).

L270: I am not sure if I understand the meaning of 'high variability of 20-year means' correctly. Maybe it is possible to rephrase?

L278-283: This paragraph seems to contain much information that is also given in the beginning of the following Sect. 4.3. I would like to suggest to move L278-283 to Sect. 4.3 and merge with the first part of this section.

L284-354: This is an interesting analysis and discussion. If I understand correctly, it supports to include multiple CMIP ensemble members in Antarctic sea-level projections as done in the work presented here. At the same time, it seems slightly detached from the previous parts of the manuscript. I would like to suggest two options that may help to add focus to this section:

A) This section may be shortened, summarizing the main analysis and the conclusion. The major part of the analysis may be moved to the Supplementary Material.

B) Parts of this section (e.g., the metrics and justification for these metrics) may be included in the Methods, and the outcomes could be highlighted and discussed with the main results. If applicable, the Antarctic sea-level contribution for the 'best' ensemble members could be added separately to e.g. Figure 6.

L290: I am not sure if I understand this phrase. Maybe replace 'assess' by e.g. 'demonstrate'?

L368-384: This paragraph contains many valid and helpful recommendations for future assessments of the Antarctic contribution to sea-level change. However, some of these recommendations do not seem to be directly justified by the presented results, or a better link to and additional information in Sect. 3 may be needed. For example, a fully-coupled assessment would be ideal to include feedbacks of the ice sheet with the ocean and the atmosphere, but some additional discussion how this would e.g. improve the representation of or remove biases in the internal climate variability in the selected CMIP6 models presented in Sect. 3.1 and Sect. 3.2 may be needed to directly relate to this study.

L379-380 / L382: Do the 'various members' / 'multiple members' refer to the CMIP6 model ensemble members or to ice-sheet initial states?

---

## Author Comment (AC1)

We thank the Editor Claudia Timmreck and the three reviewers for their careful evaluation of our manuscript. We found the comments very useful and think that our manuscript will be greatly improved thanks to them. To ensure clarity, the reviewer's comments are written in black and our responses in light blue.

**Reviewer #2**

The submitted manuscript describes a model investigation of the uncertainties in medium-range projections of the Antarctica Ice Sheet's contribution to sea level. The study focuses on uncertainties related to internal climate variability of climate forcing, both ocean and atmospheric. First, the authors evaluate CMIP6 models, and choose a subset on which to conduct their analysis. They then present historical diagnostics on the available model ensemble for each climate model, extracting the temporal variability of various ocean and atmospheric related variables. Finally, they choose a subset of ensemble members of the SSP2-4.5 scenario for each climate model and use them to force an Elmer/Ice continental Antarctica, resulting in an ensemble of projections for Antarctic Ice Sheet sea-level contribution through year 2100. Results suggest that internal climate variability can affect sea-level contribution, ranging in magnitude from 45-93%, but most of that uncertainty is dominated by atmospheric forcing over ocean forcing. The authors conclude that internal climate variability varies among the climate models, especially for the ocean forcing; therefore, they suggest a strategy for choosing ensemble members that most realistically represent the dominant climate modes of the Antarctic region. They also make recommendations for how to best consider internal climate variability in ice sheet model projections. In general, the methods are well-described and the figures are adequately presented. The analyses and science results are of high quality, and the discussion and conclusion bring up intriguing and relevant points for the ice sheet modeling community.

Overall, I find that this is an interesting study, with important results comparing the effect of internal climate variability due to the ocean and the atmosphere on ice sheet model projections. While results presenting ice sheet modeling projections by themselves could constitute their own manuscript, the authors present much more analysis, including a list of metrics for choosing appropriate model ensemble members to capture internal climate variability. While interesting, these metrics are not the ones used for choosing members for the ensemble results presented. In addition, the authors do not show outcomes that illustrate/quantify the consequences resulting from an ice sheet model using all the suggested updates to their projection procedures. As a result, I find that the addition of these extra results leads to a manuscript that lacks focus. For instance, I think it would benefit the manuscript if some of the secondary analyses were moved to a supplement. In this way, the main manuscript could be dedicated to presenting results specifically on the quantification of the uncertainty in Antarctica's sea level contribution due to internal climate variability. If the authors feel as if the new metrics should be highlighted instead, then a new organization and general story built around those results would benefit the manuscript.

Due to the extensive modification needed, I suggest that major revisions be required before the manuscript is accepted. If the authors work on better organization of the results and on

improving the clarity of their language (per suggestions outlined below), I am confident that this manuscript could result in a valuable scientific contribution to the community. Please see my comments/suggestions/questions below with regards to my major and minor concerns with the current version of the paper.

**General comments/questions:**

Results show that the ocean internal variability has a minimal effect on the projections, as compared to atmospheric variability or choice of model. If this is the case, why do the designed ensemble metrics mostly focus on evaluating the ocean forcing of ensemble members? More specifically, why is it pertinent to choose members that capture ocean internal variability well if this variability is less important?

We agree and we will better balance and explain the respective roles of the oceanic and atmospheric contributions.

1. The model ranking for Antarctic atmospheric metrics proposed by Agosta et al., (2024) is now referenced (https://doi.org/10.5281/zenodo.11595213). We will therefore revise the paragraph on the choice of the CMIP6 model (§2.1) to give an equivalent weight to the atmosphere and the ocean (we agree that the paragraph currently focus more on oceanic properties).

2. Overall, we agree that internal climate variability mostly affect sea level projections through the surface mass balance (atmosphere), as evidenced by the shaded ranges in Fig. 6. At the basin scales, things can be different. For example, the oceanic internal variability has a stronger effect than the atmospheric variability in some sectors (e.g., basins 1, 5, 9, 10 for the IPSL-CM6A-LR in Fig. 7), and the effects are of comparable magnitude in West Antarctica for the IPSL-CM6A-LR model (Fig. 6h-i).

Do the authors suggest that the climate model ocean representation of internal climate variability lacks skill to the point that using an entire ensemble of forcing does not offer a realistic projection spread? It would improve the manuscript if these questions were considered in the text/discussion/overall story of your paper. It would be even more beneficial to the paper if the authors could support the answers with analysis or results, expanding upon the plots that are already included in the paper.

Assessing the amplitude of the internal climate variability in the ocean is complex. The oceanic internal variability varies greatly depending on the climate models (§3.1 and Fig 3) and is probably underestimated (§4.1), but it is difficult to show this clearly because of the lack of observational data in the ocean over long period. We plan to generalise the assessment of internal oceanic variability carried out in the §3.1 to 15 CMIP6 models, i.e., calculation of across member standard deviation of the 1995-2014 mean potential temperature over the whole continental shelf for the 200-700m depth, in order to better compare the variability of the selected models with other CMIP6 models. The new figure will be discussed in §4.1 and added in supplementary material. The same evaluation will also be carried out on the SMB. For few

regions like the Amundsen Sea, we also have multi-year observations that show a significant variability so that we can consider that a model producing very low variability like MPI-ESM1.2-HR is unrealistic. We nonetheless consider that the variability of IPSL-CM6A-LR may be realistic.

As discussed in your manuscript, climate model ensembles are typically used to represent the spread of model internal climate variability. Forcing the ice sheet model with a large subset of members allows for the propagation of uncertainty due to this variability into projections of sea-level contribution. Here, it is suggested that this might not be appropriate, and that filtering for members that exhibit more realistic variability ("in phase with observed") could be an adopted strategy. Do the authors anticipate that selecting for members would introduce bias into the interpretation of projection uncertainty due to internal climate variability? Is it possible to make runs, or use the runs already completed, to answer this question? (See further questions/comments on this below.)

We will make our recommendation clearer. First, ice sheet models need to be initialised and calibrated to match historical observations. Achieving this would in theory be easier with a forcing from the most realistic CMIP ensemble member, which is why we attempted a selection of the best member. Another reason is that ice sheet simulations can be computationally expensive, and running simulations forced by all members of several CMIP models may not be feasible. However, we agree with the reviewer that once the ice-sheet model is calibrated, the only way to properly assess the uncertainty related to internal climate variability is to force the ice sheet simulations with multiple members. Our study also gives typical relative errors that may be used as relative uncertainty in studies that can't afford to run full ensemble members.

In the conclusion paragraph, there are four listed recommendations for ice sheet model projections. While these are all pertinent discussion points, it is not clear to me why they are included as conclusions of the presented study. More specifically, these four statements - though they may be valid suggestions - are not directly justified by the results shown. It may be that the authors believe that they are, and in this case, please rework this section, so that is clear to the reader how the results map to each of these statements; or perhaps additional figures that better illustrate the connection can be included in the manuscript revision.

We will remove the recommendation on coupled ice-sheet/climate models as it is not clearly demonstrated in our study. Instead, the discussion on ice-sheet/climate coupling will be addressed in section 4.1 which deals with the issue of the robustness of internal variability in climate models. The recommendation regarding initialisation will also be removed, as the topic of initialisation is not directly addressed in the paper.

The other two recommendations stem from the results of our study and will therefore remain in the conclusion but the link with the present study will be clarified.

**Specific comments/questions:**

Lines 12-14: Please rephrase this sentence. It is awkward and unclear. Also please specify the type of convection you refer to.

We will change *« Conversely, the amplitude of oceanic internal climate variability around Antarctica strongly depends on the climate model as underestimated convection, due to either biases in the sea-ice behaviour or in the ocean stratification, leads to weak mid-depth ocean variability »* to *« Conversely, the amplitude of oceanic internal climate variability around Antarctica strongly depends on the climate model which underestimates convective mixing in the ocean. The latter is due to either biases in the sea-ice production and associated salt rejection or in the ocean stratification that modulates the depth of convective mixing. Such biases lead to weak mid-depth ocean variability »*

Line 15: Please rephrase to something like: "We recommend based on our results that ice sheet model projections consider …" or something similar.

Thanks for the suggestion. We will replace *« We then issue recommendations for future ice-sheet projections: use several members... »* with *« Based on these results, we recommend that ice sheet model projections consider several members... »*

Line 54: "than" -> "rather than"

Yes, the manuscript will be corrected accordingly.

Lines 58-64: It is difficult to read this list organized in the current configuration. Is there a way to simplify this so it would be easier to digest for a reader, like in a table for instance?

Figure 1 will be moved to Supplementary Material and a table summarising the variables and metrics used for the evaluation will be added.

In the main text, we propose to remove the list and to replace it with a sentence including both atmospheric and oceanic properties evaluation *« The selection of models was first based on the number of members available and on the availability of 6-hourly outputs that were needed to run regional climate projections. It was also based on the model ranking for Antarctic atmospheric metrics proposed by Agosta et al. (2024), and on the model ranking for Southern Ocean metrics provided by the review of three studies which evaluate water masses properties in the Southern Ocean and Antarctic seas (Purich et al., 2021), dynamical properties in the Southern Ocean (Beadling et al., 2020) and bottom properties in the Southern Ocean (Heuzé, 2021). »*

Line 66: Please clarify what is meant by "best" here? Can this be quantified?

For each study, we calculate the RMSE between the CMIP6 model and the observational dataset with which it is compared, for all the variables analysed. The CMIP6 models are then ranked by increasing RMSE.

We will extend the description of ocean properties analysis a bit further than the response to L58-64 by summarising the metric used, the reference dataset to which it is compared and the period over which they are evaluated.

Line 71: "have some kind of", this wording is very informal and difficult to understand. Does it mean that there is a tuning included for the historical? Please articulate this more clearly for the reader.

We will remove the word 'have some kind of' and replace with *« It is also interesting to note that both UKESM1-0-LL and IPSL-CM6A-LR have prescribed vertically distributed ice-shelf melting to ensure the conservation of the ice-sheet mass over the entire simulation, which is known to be important for coastal ocean properties around Antarctica (Mathiot et al., 2017; Donat-Magnin et al., 2021). »*

Line 74: Please expand upon this in your text, including a summary of the period and in what way they compare well to ERA5.

For further explanation on the evaluation of atmospheric properties, the reader may now refer to Agosta et al, (2024) which has just been referenced : https://doi.org/10.5281/zenodo.11595213.

The comparison of atmospheric properties with ERA5 has been expanded to rebalance the explanations of atmospheric and oceanic properties in the selection of the CMIP6 model. We will reshape the existing paragraph as follows *« Agosta et al (2024) evaluate 29 CMIP6 models around Antarctica by comparing their performance with the ERA5 reanalysis over the period 1980-2004 for 9 variables. The models are ranked based on two metrics, which are (i) the mean Root Mean Square Error (RMSE) over the 9 variables normalised by the multi-model RMSE and (ii) the second maximum implausible fraction, which corresponds to the fraction of the surface where the difference between CMIP6 models and ERA5 is greater than three times ERA5 standard deviation. »*

Line 76: Please summarize how these are the best, or add more quantitative language, i.e. the best with reference to what?

See response above for Line 74.

Figure 1: This figure might be better suited for a supplement, since contains more supportive information, based analysis of the climate model runs.

Figure 1 will be moved to Supplementary Material and a table summarising the variables and metrics used for the evaluation will be added.

Line 82: Please include a reference for the friction law.

As this comment appeared several times, we will add the law in the main text for clarity as follows *« The ice dynamics is computed by solving the Shallow Shelf Approximation (SSA) of the Stokes equations (MacAyeal, 1989), assuming an isotropic rheology following Glen's flow law (Glen, 1955) and a linear friction law (i.e., $\tau_b = C\, u_b$ where $\tau_b$ is the basal shear stress, $C$ the friction coefficient and $u_b$ the basal ice velocity). »*

Line 85: Please clarify in the text what is meant by curvatures here?

Curvature is the second derivative of the modelled fields (velocity and ice thickness here), i.e., the Hessian matrix. For more explanation, the reader can refer to §2.2 of Gillet-Chaulet et al. (2012):

Anisotropic mesh adaptation is now widely used in numerical simulations especially with finite elements, as it allows to refine the mesh where needed to capture the flow features within a certain accuracy without increasing the computational cost excessively. The method is generally based on an estimation of the interpolation error used to adjust the mesh size so that the discretisation error is equally distributed over the whole domain. It can be shown that an estimate of the interpolation error induced by the meshing is obtained from the Hessian matrix of the modelled field, allowing to define an anisotropic metric tensor at each node (Frey and Alauzet,

*In the main text, we will add a parenthesis « The mesh is preferentially refined both close to the grounding line and in areas where observed surface velocities and thickness show high curvatures (i.e., high second derivative of the modelled field, Gillet-Chaulet et al.,2012) ...»*

Line 104: Please explain in the text more about how this is done. Are there numerical techniques used (inverted?), or is there a procedure designed determine the right correction?

We assume that this comment is about Line 106. We minimise the RMSE between the modelled and the observed ice-sheet mass change for West Antarctica by applying reduction of the friction coefficient, to limit the model drift. A proper inversion is done to obtain the initial basal friction coefficients. Then, these coefficients are adjusted by trial and error to limit the model drift.

We will rephrase as: *« In contrast to (Hill et al., 2023), we do not correct the surface mass balance to maintain a steady state, but we apply a 10% uniform reduction of the inverted friction coefficients to reduce the model drift. For this, we minimise the RMSE between the modelled and the observed ice-sheet mass change (The IMBIE Team, 2018) for West Antarctica. Our configuration overestimates the mass loss trend in the West Antarctica by only 6% but still largely overestimates mass gain in East Antarctica and in the Peninsula (Tab.1). »*

Line 107: "correct" -> pleased rephrase this, as this term is not appropriate to describe model results, and if I am reading the rates right, the WAIS trend is still technically outside of the error bounds.

We will remove the word 'correct' and articulate the sentence more clearly *« Our configuration overestimates the mass loss trend in the West Antarctica by only 6% but still largely overestimates mass gain in East Antarctica and in the Peninsula (Tab.1). »*

Lines 109-119: Awkward – please rephrase this last sentence of the paragraph.

We will rephrase as *« However, this bias should not impact most of the analyses presented here, as the projections in response to the CMIP6 climate models are analysed relative to each other. »*

Lines 112-113: Please specify that this statement is for an Antarctic Ice Sheet model CMIP simulation.

Yes, the manuscript will be corrected accordingly.

Line 114: This phrasing is confusing for a reader because the sentence before already implies that you can add them together. I think the point is that we can attribute dynamic ice loss to ocean-forced changes, because the SMB driven dynamics is trivial. Please rephrase.

We will rephrase as *« Antarctic future mass change results from combined effects of surface mass balance and dynamical changes. In standalone ice-sheet simulations, variations in surface mass balance can be attributed to atmospheric-forced changes and dynamical mass loss to ocean-forced changes as SMB changes have little impact on the Antarctic sea-level dynamical contribution over a century (Seroussi et al., 2014; Seroussi et al., 2023). Thus, the effect of atmospheric and oceanic variations on Antarctic sea-level contribution can be analysed separately and then summed to reconstruct the combined effect (Bindschadler et al., 2013). »*

Line 120: Please give more specifics on how these ensembles were chosen. Even though "see next section" is included here as a reference, it is unclear where in the next section this information is included. If so, please note the specific section number for clarity.

We will add the specific section number for clarity.

Line 123: If this statement refers to both types of forcing (atmospheric and ocean), please specify that here, as the sentence is currently vague.

Yes, the manuscript will be corrected accordingly.

Lines 129-135: I do not think this detailed explanation is really needed here. A sentence explaining that MAR output was not available would likely suffice for justification.

We think that this is an important explanation for the ice-sheet community and we would prefer to keep this explanation.

Line 173: Awkward sentence, please rephrase.

We rephrase the sentence *« We now examine the Amundsen Sea more closely as the region is particularly important for the ice-sheet mass loss. »* like this: *« We now focus on the Amundsen Sea, as the region is currently experiencing the largest mass loss in Antarctica. »*

Line 184: "instead" -> something like "as well as" or "in addition to"

Yes, the manuscript will be corrected accordingly.

Line 187: Since one of the conclusions of this paper is related to this point, it would be beneficial to add the not shown plots to the manuscript, instead of just describing them (i.e. for 60 years). Perhaps in a supplement, so that the point can better be made that 60 would be an improvement over 20 years.

A figure similar to Figure 3 with a 60-year period instead of 20-year period will be added in the Supplementary Material to support our arguments.

Line 197: Please note here that this therefore infers more precipitation.

We will add 'resulting in more precipitation' in the sentence *« By 2100 and for the SSP2-4.5 medium scenario, runoff is supposed to remain limited (Kittel et al., 2021), so the SMB is projected to increase largely due to the increased water vapour saturation in warmer air, resulting in more precipitation. »*

Line 198: Please rephrase, e.g., "We therefore focus on variability in SMB components such as precipitation and air temperature."

Yes, thanks for the suggestion. As we also analyse SMB, we will rephrase as *« We therefore focus on variability in SMB and its main components such as precipitation and air temperature. »*

Line 202: consistently -> consistent

Yes, the manuscript will be corrected accordingly.

Line 203: Awkward, please rephrase this sentence.

We will rephrase as *« The largest SMB variability is simulated along the coast of the Amundsen and Bellingshausen seas, which results from the high internal climate variability of atmospheric circulation and air temperature in these regions (Fig. 4d-i). »*

Line 216: Please make these statements more specific, e.g., … explained by "grounding line change and dynamic response" of these glaciers.

We will rephrase as *« The West Antarctic positive SLC is mostly explained by the grounding line migration and the dynamical response of Pine Island and Thwaites ice shelves (~3 cm in Fig. 5c, basin 11) as well as Getz ice shelf (~1 cm, basin 10). »*

Lines 226-227: relative -> related?  (both instances in the sentence)

Yes, *related* is the right word. The manuscript will be corrected accordingly.

Line 252: although -> even though

Yes, the manuscript will be corrected accordingly.

Lines 265-267: The wording here is confusing, please rephrase.

We will remove the argument of GHG scenario in the sentence and focus mainly on the present-day drift and we will mention the sources of uncertainty described in Seroussi et al., (2023), i.e., uncertainties in the physics of the ice-sheet model, the choice of climate model and uncertainties associated with ice-climate interaction (melt parameterisation and calibration).

Line 269: It would be helpful to also note in the text that for the 20 years in question, the Antarctic Ice Sheet of reality appeared to be in a stable state, as opposed to its state under the random phase of a climate model you mention that would occur during these same "years" in a climate model.

*The ice sheet of reality was not necessarily in a stable state, e.g., it lost mass in the Amundsen Sea sector. But the random phase does change the mass evolution. We have not added anything on this point.*

Line 270: Please clarify here what is meant here by high variability of 20-year means.

*We will replace with « wide confidence interval on a 20-year mean ».*

Line 279: "applying" -> "to apply"

*Yes, the manuscript will be corrected accordingly.*

Line 290: The previous paragraphs justify why this would make sense to do, but please add some discussion about how one would go from that previous logic to each of the outlined decisions here. More specifically, based on this list of assessments, as a reader I can gather that one should choose the members that have the best representation of important modes of variability known to affect the ocean and atmosphere in/around Antarctica. Please walk the reader through why these key internal variability metrics are chosen, i.e. what it is in the results that can point to each of these as a justified criterion for ensemble members.

*The discussion section on the identification of the best member will be revised. To facilitate reading, Figure 8 and the list of associated metrics will be moved to the appendix and similar analysis will be performed with UKESM-0-LL to get more robust conclusions.*

*In the revised paragraph, we will specify that the metrics considered were chosen to:*

*– ensure a good representation of the mean atmospheric and oceanic states. We selected variables directly used to drive the ice sheet model, i.e., the SMB for the atmosphere and temperature for the ocean. We focused on the ocean temperature in the Amundsen sector as the region experiences the current main mass loss and CTD profile data are available for a relatively long period from 1994 to 2018 in this area.*

*– ensure a good representation of the amplitude of oceanic variability using the same observational data described in the previous paragraph. We did not evaluate the variability of SMB since it has been relatively stable in recent years and there is no observational data.*

*– ensure the best representation of important modes of variability known to affect the ocean and atmosphere in/around Antarctica. We focus our analyses on the SAM and TPI index.*

*– ensure a phasing of internal variability with observations, which could be important for future detection/attribution studies and for projected Antarctic sea-level contribution. We chose two variables, sea-ice concentration and the presence of warm periods on the continental shelf of the Amundsen Sea to provide insights on the phasing of internal variability.*

It should be noted that this part of our study remains exploratory and the choice of variables and metrics is indeed very subjective, which is part of the caveats that we discuss.

Line 344: Please remind the reader here that this is only for the IPSL members.

Following the response to the previous comment, the research for potential best members will be conducted only for both IPSL-CM6A-LR and UKESM1-0-LL.

Line 354: Would the results (projections and spread) change if one were to follow the newly created rankings and chose a different group of members for the analysis? I realize it might not be possible to do new runs to completely answer this question, but are there subsets of runs that are already completed that can be used (i.e. redo analysis of runs with top half vs. bottom half of members?). Or even still, in your discussion, is it possible to use the results presented and extrapolate the results to suggest how much it would matter to the projections to use the ranked members instead?

We believe that we do not have enough simulations to divide our set into two subsets (only two members in each subset for UKESM and IPSL). However, we can comment on whether the best-ranked members tend to have a strong/weak dynamical or SMB contribution.

Line 367: Please specify what is meant by convection here, for example where, and at what depth?

For clarity, we replace '*convection*' by '*ocean convective mixing on the continental shelf*'. The entire water column is affected.

Line 368: Please rephrase, i.e. changing -> "using the following practices to remove the effects of internal climate variability on projection results…"

Based on the comments of all the reviewers, we decided to only keep two recommendations in the conclusion, so this part has been reshaped.

Lines 369-372: Here, metrics to find ensemble members that are in phase with observed variability are described. Please comment on the consequences of filtering these members. In a way, this could defeat the purpose of using a large ensemble, because it discounts variability that is intrinsic within a climate model (even if it is a variability that is not realistic or wrong). Is doing a filtering justified because results suggest that the ocean models are so wrong (i.e. the various members are all over the place in terms of variability and not at all "realistic"), that they no longer offer trustworthy projections? Is it important to rank climate models themselves based on metrics of variability we can derive from their ensemble, in order to tell the community which ones might have completely unrealistic internal variability? Adding these concepts to the discussion would broaden and enhance the scope and impact of the presented work.

Thanks for the valuable suggestion. First of all, a justification for filtering members in the calculation of metrics is that ice sheets act as low-pass filters, they are not very sensitive to interannual forcing. We do not suggest that the variability is wrong in all ocean components, this conclusion only holds for MPI-ESM1.2-HR. Second, we agree that it is important to show

*where these 3 models stand in terms of internal climate variability, and we will add a comparison to a larger set of CMIP6 models in the revised manuscript.*

Line 374: Please add some text on why this is so, i.e., add a "because …". It would be helpful to clearly state your reasoning here. For example, 1) climate models show important modes longer than 20 years, and/or 2) the 20 years in an observational period that may seem appropriate in "reality" may not align with an appropriate period within a climate model.

*We plan to develop a more robust argumentation for this point in a discussion section.*

Line 377: "but not in the observations", following this, please add something like "and therefore introduce a different source of bias."

*Yes, the manuscript will be corrected accordingly.*

Line 382: "Ice-sheet models should" -> something like "To capture the full uncertainty due to internal climate variability, ice sheet models would ideally be …"

*Yes, the manuscript will be corrected accordingly.*

---

## Author Comment (AC2)

We thank the Editor Claudia Timmreck and the three reviewers for their careful evaluation of our manuscript. We found the comments very useful and think that our manuscript will be greatly improved thanks to them. To ensure clarity, the reviewer's comments are written in black and our responses in light blue.

**Reviewer #3**

**Summary**

The study by Caillet et al. quantifies the uncertainties in the projected Antarctic contribution to sea-level change by 2100 related to internal climate variability in a subset of CMIP6 models. Three CMIP6 models are selected based on a summary of previous evaluations (Purich and England, 2021; Beadling et al., 2020; Heuzé, 2021; Sect. 2.1). The Antarctic sea-level contribution is projected with the stand-alone ice-sheet model Elmer/Ice; the respective experimental setup is presented in Sect. 2.2 and Sect. 2.3. First, internal climate variability in the selected CMIP6 models is explored (Sect. 3.1 and Sect.3.2). Then, the Antarctic contribution to sea-level change projected by Elmer/Ice based on different ensembles members for the selected CMIP6 models is presented (Sect. 3.3). The authors quantify the effect of internal climate variability on the Antarctic sea-level contribution by the end of this century with 45% to 93%, with a higher impact from atmospheric variability compared to ocean variability, and modulated by the CMIP6 model. Results are discussed in terms of the robust representation of internal climate variability in CMIP models (Sect. 4.1), the internal climate variability as a source of uncertainty in Antarctic sea-level projections (Sect. 4.2), and identifying best ensemble members as an alternative approach to account for internal climate variability in sea-level projections (Sect. 4.3). The authors conclude with general recommendations for future assessments of the Antarctic contribution to sea-level change (Sect. 5).

**General comments**

By bringing together internal climate variability and the future evolution of the Antarctic Ice Sheet, the paper addresses a relevant and scientific interesting question, that has rarely been explored in previous assessments of the future trajectory of the Antarctic Ice Sheet. The presented results and related discussion may be valuable for future assessments of the Antarctic contribution to sea-level rise. The title clearly reflects the contents of the paper. The abstract provides a concise and complete summary. Overall, the study has a sound methodology and experimental setup. In some cases, the description of the methods could be more precise, and the clarity of language improved. While the results span a wide range from exploring the representation of internal climate variability in selected CMIP6 models to the modelled response of the Antarctic Ice Sheet, the manuscript may benefit from linking both in greater depth (e.g. explaining the climate model - dependence of the impact of climate variability on the projected sea-level contribution, ranging between 45% and 93%, if possible). Here, additional figures, e.g. in a Supplementary Material, may be helpful for the reader.

We will work on section 3.3 to better stress the links between the analysis of internal climate variability in the CMIP6 models and the projected Antarctic sea-level contribution.

In addition, the discussion on a possible selection of best ensemble members from CMIP models could be better integrated in the manuscript.

For a better readability and following suggestions by the other reviewers, the discussion section on the identification of the best member will be revised. Figure 8 and the list of associated metrics will be moved to the appendix and the same analysis will be perform for UKESM-0-LL to get more robust conclusions.

The reasons for choosing the best member will also be better explained. First, ice sheet models need to be initialised and calibrated to match historical observations. Achieving this would in theory be easier with a forcing from the most realistic CMIP ensemble member, which is why we attempted a selection of the best member. Another reason is that ice sheet simulations can be computationally expensive, and running simulations forced by all members of several CMIP models may not be feasible.

Finally, some additional explanations may be needed to directly derive and support some of the recommendations given in the conclusion based on the results presented in this study.

Based on the comments of all the reviewers, we decided to only keep the two recommendations that are the most motivated by our findings, and we will make sure that the link with our findings is explicit.

We will remove the recommendation on coupled ice-sheet/climate models as it is not clearly demonstrated in our study. Instead, the discussion on ice-sheet/climate coupling will be addressed in section 4.1 which deals with the issue of the robustness of internal variability in climate models. The recommendation regarding initialisation will also be removed, as the topic of initialisation is not directly addressed in the paper.

I have included more specific comments, questions and suggestions below.

**Specific comments**

L12-14: In the abstract, the results of the sea-level projections are summarized before describing the internal climate variability in the CMIP6 models. Maybe it would be more intuitive to follow the same order as in the main text (that is, internal climate variability in CMIP6 models followed by sea-level projections)?

We will rearrange the abstract to match the structure of the paper.

L12-14: Maybe a brief remark on the upper end of the amplitude of oceanic internal variability covered by different climate models could be added, in addition to the mentioned and explained weak mid-depth ocean variability?

We will add the range of the ratio between the amplitude of internal atmospheric variability and internal oceanic variability for the various models. This ratio varies from 1,8 for IPSL-CM6A-LR to 4,8 for MPI-ESM1.2-HR.

We will also temper our statement for West Antarctica, where the amplitude of oceanic and atmospheric variability may be similar depending on the CMIP model.

L15: Please specify 'use of several members in the run and its initialisation'. I think I understand what is meant here after reading the manuscript but this phrase may be unclear for the reader when starting with the abstract.

Based on the general comments described above, we have reformulated the recommendations by focusing only on those that are really motivated by our results, i.e., (i) the use of several members of the same model and several models since internal variability does not have the same amplitude depending on the models and (ii) the use of longer reference period. We suggest the following sentence « *Based on these results, we recommend that ice sheet model projections consider (i) several climate models and several members of a single climate model to include the impact of internal climate variability and (ii) 50-year averages for reference period from which anomalies are estimated to attenuate any unexpected shift triggered by internal variability.* »

L24: Maybe add 'estimates of' or 'projections of', e.g. 'Estimates of the AIS contribution to future sea level rise are currently based...'

Yes, the manuscript will be corrected accordingly.

L25: I think CMIP stands for Coupled Model Intercomparison Project. Please check.

Thanks, you are completely right. The manuscript will be corrected accordingly.

L31-35: In this paragraph climate variability is introduced as consisting of two components (1) variability from natural and anthropogenic external forcings and (2) internal variability, and explanations for these components are given, after having referred to internal climate variability in the previous paragraph. Maybe some restructuring is possible to define internal climate variability with its first use?

Internal climate variability is already defined in the abstract.

In the introduction, internal climate variability is used for the first time without having been introduced (L28-30), but its definition is provided in the following sentence (L31). We therefore prefer to keep these things unchanged.

L49/50: Maybe 'the Antarctic Sea Level Contribution'?

Yes, the manuscript will be corrected accordingly.

L50: Why did you chose the SSP2-4.5 emission pathway? Please add a short explanation either here or in Sect. 2.3.

We will clarify the choice of scenario in §2.3 rather than at the end of the introduction as follows « *We use the medium SSP2-4.5 scenario, which corresponds to a global warming of 1.4 to 3.0°C from 1995-2014 to 2081-2100 (90% confidence interval, Lee et al. 2021) and seems the most representative of current effort to tackle climate change (Riahi et al., 2017). As the choice*

*of greenhouse gas emission scenario has only a limited impact on the projected Antarctic contribution to sea level rise until 2100 (Seroussi et al., 2020), we have not repeated our calculations for other scenarios. »*

L56: 'drivers' instead of 'driver'?

Yes, the manuscript will be corrected accordingly.

L58-64: Please specify the properties that are used to evaluate the CMIP6 models. Some of them are given in Figure 1 or its caption, but it may be helpful to also include them in the main text.

- What properties of ASBW and CDW are evaluated? Please add this information also in the main text.

- Many dynamical features for the Southern Ocean are listed in the legend of Figure 1b. It might be helpful for the reader to better link the legend and caption of Figure 1 to the description in the main text. This applies to e.g. the ocean properties that are evaluated in terms of their meridional gradients.

- What bottom properties of the Southern Ocean are evaluated? Maybe add this information also in the main text.

Figure 1 will be moved to Supplementary Material and a table summarising the variables and metrics used for the evaluation will be added.

In the main text, we propose to remove the list and to replace it with a sentence including both atmospheric and oceanic properties evaluation « *The selection of models was first based on the number of members available and on the availability of 6-hourly outputs that were needed to run regional climate projections. It was also based on the model ranking for Antarctic atmospheric metrics proposed by Agosta et al. (2024), and on the model ranking for Southern Ocean metrics provided by the review of three studies which evaluate water masses properties in the Southern Ocean and Antarctic seas (Purich et al., 2021), dynamical properties in the Southern Ocean (Beadling et al., 2020) and bottom properties in the Southern Ocean (Heuzé, 2021). »*

L58-64: To facilitate readability, bold or italic fonts for some phrases in this list could be used (e.g. for the evaluated water masses). As an alternative, these properties could be given in a table rather than in a list.

See the response above.

L65: Is the assessment presented in Figure 1 based on one ensemble member of the respective CMIP6 models or an average over all available ensemble members? As different ensemble members are used later in the manuscript, maybe add this information here (or in the figure caption) to avoid confusion.

Thanks for the suggestion. The assessment presented in Figure 1 is only based on the first member of each CMIP6 model. We replace the sentence of L65 by « *A summary of this*

*assessment is provided in* Supplementary Material *based on the analysis of the first member of each climate model. »*

Figure 1 will be moved to Supplementary Material.

L66: How is 'best' defined? Does UKESM-1-0-LL have one of the lowest RMSE in all three studies? Maybe state this more explicitly here.

For each study, we calculate the RMSE between the CMIP6 model and the observational dataset with which it is compared, for all the variables analysed. The CMIP6 models are then ranked by increasing RMSE.

We will extend the description of ocean properties analysis a bit further than the response to L58-64 by summarising the metric used, the reference dataset to which it is compared and the period over which they are evaluated.

L68: If I understand Figure 1 correctly, MPI-ESM1.2-HR was evaluated in two of three studies (red triangles in Figure 1a and c). Please check.

Yes, MPI-ESM1.2-HR was evaluated in only two of the three studies.

L70-73: This is a very general sentence, in particular for readers not familiar with the representation of ice-shelf melting in CMIP models. What is meant by 'some kind of prescribed ice-shelf melting at depth'? Does this impact the assessment / ranking of CMIP6 models in Figure 1? I think it may be helpful to briefly discuss the link between the treatment of ice-shelf melting and the CMIP6 model assessment, if this information is mentioned here.

Prescribed meltwater flux at depth has an impact on the properties of the water masses and sea-ice. This inflow affects the stratification of the water column and can be more favourable to convective mixing by reducing the density at depth (Mathiot et al., 2017) and to the intrusion of circumpolar deep waters (Haigh et al., 2024). In contrast, models that prescribe meltwater flux only at the surface tend to increase the stratification of the water column and reduce exchanges between the surface and deeper waters, thereby preventing variability.

We will remove the word 'have some kind of' and replace with *« It is also interesting to note that both UKESM1-0-LL and IPSL-CM6A-LR have prescribed vertically distributed ice-shelf melting to ensure the conservation of the ice-sheet mass over the entire simulation, which is known to be important for coastal ocean properties around Antarctica (Mathiot et al., 2017; Donat-Magnin et al., 2021). Most of the CMIP models prescribes meltwater flux only at the surface, which tend to increase the stratification of the water column (Mathiot et al., 2017) and reduce exchanges between the surface and deeper waters, thereby preventing variability. »*

L74-77: Please add more detail on the assessment of atmospheric properties for the CMIP5/6 models in the manuscript, also given that Agosta et al. (2022) refers to a conference abstract. For example, which atmospheric properties have been evaluated and which method is used for the assessment?

For further explanation on the evaluation of atmospheric properties, the reader may now refer to Agosta et al, (2024) which has just been referenced : https://doi.org/10.5281/zenodo.11595213.

The comparison of atmospheric properties with ERA5 has been expanded to rebalance the explanations of atmospheric and oceanic properties in the selection of the CMIP6 model. We will reshape the existing paragraph as follows *«Agosta et al (2024) evaluate 29 CMIP6 models around Antarctica by comparing their performance with the ERA5 reanalysis over the period 1980-2004 for 9 variables. The models are ranked based on two metrics, which are (i) the mean Root Mean Square Error (RMSE) over the 9 variables normalised by the multi-model RMSE and (ii) the second maximum implausible fraction, which corresponds to the fraction of the surface where the difference between CMIP6 models and ERA5 is greater than three times ERA5 standard deviation. »*

Figure 1: Please consider marking the selected CMIP6 models in a different way, e.g. by colouring the model name or adding a box around the model name. The red triangles can be easily confused with the other markers (or appear within the legend, compare Figure 1b).

Yes, the manuscript will be corrected accordingly.

Figure 1: Please briefly introduce the abbreviations used in the legend, e.g. in panel b in the figure caption and / or in the main text (L58-64).

Yes, the manuscript will be corrected accordingly. We will add a table summarising the variables and metrics used for the evaluation. Figure 1 and the corresponding table will be moved to Supplementary Material.

L82: Please add a reference for the friction law.

As this comment appeared several times, we will add the law in the main text for clarity as follows *« The ice dynamics is computed by solving the Shallow Shelf Approximation (SSA) of the Stokes equations (MacAyeal, 1989), assuming an isotropic rheology following Glen's flow law (Glen, 1955) and a linear friction law (i.e., $\tau_b = C\, u_b$ where $\tau_b$ is the basal shear stress, $C$ the friction coefficient and $u_b$ the basal ice velocity). »*

L84: What do you mean by 'preferentially refined'? Please specify.

We will remove the word 'preferentially'.

L85: What do you mean by 'high curvatures'? Please clarify.

Curvature is the second derivative of the modelled fields (velocity and ice thickness here) i.e., the Hessian matrix. For more explanation, the reader can refer to §2.2 of Gillet-Chaulet et al. (2012):

Anisotropic mesh adaptation is now widely used in numerical simulations especially with finite elements, as it allows to refine the mesh where needed to capture the flow features within a certain accuracy without increasing the computational cost excessively. The method is generally based on an estimation of the interpolation error used to adjust the mesh size so that the discretisation error is equally distributed over the whole domain. It can be shown that an estimate of the interpolation error induced by the meshing is obtained from the Hessian matrix of the modelled field, allowing to define an anisotropic metric tensor at each node (Frey and Alauzet,

In the main text, we will add a parenthesis *« The mesh is preferentially refined both close to the grounding line and in areas where observed surface velocities and thickness show high curvatures (i.e., high second derivative of the modelled field, Gillet-Chaulet et al.,2012) ... »*

L100-103: Are the ocean temperature corrections to match observed melt rates also based on Reese et al. (2023)? I assume that these may differ from the corrections presented in Reese et al. (2023) given the use of a different ocean climatology here. Please describe how the temperature corrections applied here are derived. It may also be helpful for the reader to briefly mention why temperature corrections are applied (instead of e.g. changing the PICO parameters to match present-day observed melt rates).

We used the set of parameters defined in Reese et al., (2023) which is based on the sensitivity of melt rates to ocean temperature changes obtained from both observations and numerical ocean projections. Ocean temperature corrections are not based on Reese et al., (2023). We have carried out our own temperature correction because (i) the climatology is different (ISMIP6 climatology in our paper instead of climatology from Schmidiko et al. (2014) in Reese's paper) and (ii) the ice-sheet geometry in our Elmer/Ice configuration resulting from an inversion initialisation is quite different from the geometry of Reese's model resulting from a long spin-up initialisation. Our aim is to match the observational estimates from Adusumilli et al., (2020) for all ice shelves over the average period 1995-2014 (see Figure 2) by applying temperature corrections between -2°C and 2°C (step of 0,1°C) in each basin defined in Reese et al., (2018).

In the main text, we now indicate:

- that the set of parameters are based on observations and ocean models: *« Here, the parameters are those detailed in Reese et al. (2023), i.e., C = 2 Sv m3 kg−1 and $\gamma T$ = 5.5×10−5 m s−1, which are based on the observed and ocean-modelled sensitivity of melt rates to ocean temperature changes. »*

- the reasons that led us to carry out our own temperature correction: *« A correction of temperature, ranging from -1.8°C to 0.6°C with respect to the ocean climatology, is added to match the 1994-2018 melt rates estimates from Adusumilli et al. (2020) (see Fig. 2). This correction differs from Reese et al.,*

*(2023) as the current ice-sheet geometry and the oceanic climatology used in this study are different from the one considered in Reese et al. (2023). »*

L106-107: Why is a 10 % reduction of the inverted friction coefficients applied? Is this based on testing, a 'best fit' or some other methodology? Does the reduction of the friction coefficients change the modelled velocities (as this quantify has been the target of the inversion)?

We minimise the RMSE between the modelled and the observed ice-sheet mass change for West Antarctica by applying reduction of the friction coefficient, to limit the model drift. A proper inversion is done to obtain the initial basal friction coefficients. Then, these coefficients are adjusted by trial and error to limit the model drift.

We will rephrase as: *« In contrast to (Hill et al., 2023), we do not correct the surface mass balance to maintain a steady state, but we apply a 10% uniform reduction of the inverted friction coefficients to reduce the model drift. For this, we minimise the RMSE between the modelled and the observed ice-sheet mass change (The IMBIE Team, 2018) for West Antarctica. Our configuration overestimates the mass loss trend in the West Antarctica by only 6% but still largely overestimates mass gain in East Antarctica and in the Peninsula (Tab. 1). »*

The friction coefficient correction does not significantly impact the ability of the model to reproduce the observed velocities. The initial RMSE between modelled and observed velocities was around 40 m/yr and increased by around 10 m/yr due to friction coefficient correction.

L107-108: The ice-sheet model configuration slightly overestimates mass loss in West Antarctica (when compared to the uncertainty ranges of the observations) if I understand Table 1 correctly.

We will remove the word 'correct' and articulate the sentence more clearly *« Our configuration overestimates the mass loss trend in the West Antarctica by only 6% but still largely overestimates mass gain in East Antarctica and in the Peninsula (Tab.2). »*

L108-109: Can you maybe add a brief remark (or a figure) on how large the trend bias in the ice-sheet model setup is?

We will add *« In general, our configuration is currently gaining a little mass (+36 Gt/yr, Tab. 2), instead of losing mass (-109±56 Gt/yr, Tab. 1). »*

L109-110: Please specify the reference that your results (in terms of the Antarctic sea-level contribution) are compared to. Are the projections in response to the CMIP6 climate models analysed relative to each other? Do you substract from a control experiment to remove the drift? After reading the discussion, I think the trend is not removed.

The projections in response to the CMIP6 climate models are analysed relative to each other. We will replace *« This is why our results are primarily analysed in relative terms. »* with *« However, this bias should not impact most of the analyses presented here, as the projections in response to the CMIP6 climate models are analysed relative to each other. »*

L112-114: This formulation might be confusing for some readers.

*We will rephrase as « Antarctic future mass change results from combined effects of surface mass balance and dynamical changes. In standalone ice-sheet simulations, variations in surface mass balance can be attributed to atmospheric-forced changes and dynamical mass loss to ocean-forced changes as SMB changes have little impact on the Antarctic sea-level dynamical contribution over a century (Seroussi et al., 2014; Seroussi et al., 2023). Thus, the effect of atmospheric and oceanic variations on Antarctic sea-level contribution can be analysed separately and then summed to reconstruct the combined effect (Bindschadler et al., 2013). »*

Figure 2: It may be helpful to indicate the most relevant ice shelves in a map (as already done for the Antarctic basins in Figure 7).

Instead of a map, we suggest adding an indication of the main regions (Amundsen, Aurora and Bellingshausen) in red on the Figure 2, like this:

[Figure]

L115-116: I got confused by the focus on the ocean here. Do you also run projections with ocean forcing only?

Given that SMB change has a limited impact on the dynamical sea-level contribution, the Antarctic global sea-level contribution can be calculated as the sum of the dynamical contribution (modulated by oceanic variability) and the SMB contribution (modulated by the atmosphere variability).

On the one hand, the SMB seal-level contribution is directly deduced from the emulated SMB (cumulative sum of the SMB - initial SMB in 2015). On the other hand, to investigate the dynamical sea-level contribution, we run Elmer/Ice simulations driven by the SMB of the first member of selected CMIP6 models and the ocean of several members of the selected CMIP models from 2015 to 2100 and we then remove the SMB contribution of the first member.

We will add *« On the one hand, the SMB seal-level contribution is directly deduced from the emulated SMB (i.e., cumulative SMB – initial SMB). On the other hand, the dynamical sea-level contribution is investigated through Elmer/Ice simulations driven by the SMB of the first member and the ocean of several members of the selected CMIP models. We then remove the SMB contribution of the first member to deduced the dynamical contribution. »*

L116: I am not sure if I understand what is meant by 'constrained'. Maybe consider replacing by e.g. 'driven' or 'forced', if applicable.

'driven' will be used in the revised manuscript.

L118-122: Please add more details on the selection of the CMIP6 ensembles members. I am not sure which section you are referring to for additional information on the selection, based on covering a wide spread in (1) possible ocean temperatures in the Amundsen Sea Embayment and (2) surface mass balance. Is the focus on the Amundsen Sea Embayment motivated by observed present-day mass loss in this region? Why is a different number of ensemble members chosen for each CMIP6 model? I appreciate the assessment of CMIP6 models in Figure 1, but, if I understand correctly, this evaluation justifies the choice of the CMIP6 model rather than the individual ensemble members for driving Elmer/Ice. It might be helpful for the reader to stress the link between the CMIP6 model evaluation, the assessment of the internal climate variability for these CMIP6 models and the selection of a subset of ensemble members for driving Elmer/Ice.

We will replace the paragraph with *« Because of the numerical cost of our simulations, we select a limited number of members. The selection is made over the current period (1995-2014 means) to cover the widest range of values for the ocean temperature on the continental shelf in the Amundsen Sea. We focus on this region as (i) observed present-day ocean-induced mass loss and (ii) the amplitude of the across member standard deviation of the 1995-2014 mean potential temperature are among the highest around Antarctica (see Fig. 3d-f, j-l).*

*We selected 5 members for each CMIP6 model, i.e., the two members with the coldest temperatures and the two members with the warmest temperatures in the Amundsen Sea continental shelf (Fig. 3j-l), for which the SMB is available, as well as member 1, used by default in most of the studies. In total, we run 11 simulations, five with the IPSL-CM6A-LR model (r1i1p1f1, r3i1p1f1, r6i1p1f1, r11i1p1f1, r25i1p1f1, see https://goo.gl/v1drZl for CMIP6 convention name of ensemble members), four with the UKESM1-0-LL model as member 1 is already included in the temperature criteria (r1i1p1f2, r2i1p1f2, r4i1p1f2, r8i1p1f2) and two with the MPI-ESM1.2-HR model (r1i1p1f2, r2i1p1f2). As the oceanic variability is very low in MPI-ESM1.2-HR (Fig. 3f), we retained only member 1 and another member to verify the low impact of oceanic variability on the dynamic contribution. »*

We took into account all the members available for the SMB contribution.

L120-122: I am not sure how familiar readers are with the CMIP variant labelling. While it may not be necessary to explain it in full detail, it may be helpful to briefly state that these lists describe different ensemble members for each of the CMIP6 models.

These abbreviations are a CMIP convention and are a brief description of the experiment. rX corresponds to realization index (i.e., the member number), iX to initialisation index, pX to physics index and fX to the forcing index.

We will add, in the main text, a link to a URL (https://goo.gl/v1drZl) that describes the CMIP6 writing conventions for the attributes.

L129-135: This paragraph could be shortened. Maybe detailed information on the SMB in ISMIP6-Antarctic (L129-130) is not needed here.

We think that this is an important explanation for the ice-sheet community and we would prefer to keep this explanation.

L135: Maybe 'constrain' could be replaced by 'drive' or something similar, if applicable.

'drive' will be used in the revised manuscript.

L136-144: I would like to mention that my comments are limited to this manuscript, and I have not assessed the approach for emulating MAR and thus for obtaining the estimates of SMB used in this study. From my point of view, no detailed evaluation is needed here and it is fine to refer to the approach described in Jourdain et al. (2024, in discussion) as done. Please make sure that respective inputs and outputs of this approach become clear (see some of the following comments / questions).

We would like to keep this summary of the methodology so that readers can get the essence of this method without digging into Jourdain et al. (2024).

L137: 'surface melting' instead of 'melting'?

Yes, the manuscript will be corrected accordingly.

L140-141: I am not sure if I understand correctly how the SMB for a given member is estimated. What is meant by 'perturbed as a function of the annual temperature difference'?

Yes, the exponential dependencies are based on temperature differences between the members.

L150: Maybe replace 'a subset' by 'the subset'?

Yes, the manuscript will be corrected accordingly.

L150: 'two first subsections' could be replaced by directly stating the subsections that you would like to refer to here to improve readability.

Yes, the manuscript will be corrected accordingly.

L153-155: It might be helpful for the reader to explicitly state the ocean properties that reflect the oceanic internal climate variability in the beginning of this section (that is, salinity and temperature, as shown in Fig. 3, and as eventually described in the beginning of the following paragraph starting in L159).

*At the beginning of the paragraph, we will add « Ocean internal climate variability is investigated through salinity and temperature variability. »*

L155: What is meant by 'typical' standard deviation across model members? Does this mean that in most regions values are around 0.017 g kg-1 and 0.07°C for MPI-ESM1.1-HR? Or are these typical values for CMIP6 models?

We will remove the word 'typical' which has indeed no relevance here.

L159: 'continental shelf' instead of 'shelf'?

Yes, the manuscript will be corrected accordingly.

L161: Maybe replace 'largest variability' by 'large variability' to avoid confusion? If I understand correctly, for example, the highest variability in mid-depth salinity for UKESM1-0-LL is found around Prydz Bay.

Yes, the manuscript will be corrected accordingly.

L171: Is 'deep ocean' considered as same ocean depth as 'mid-depth'?

'deep ocean' is used here to characterise offshore waters, outside the region of the continental shelf, but Figure 3 shows the mean salinity and temperature standard deviation between 200 and 700 m over the entire area represented. We can replace 'deep ocean' by 'beyond the continental shelf'.

L173: I would like to suggest to replace 'ice-sheet mass loss' by 'present-day ice-sheet mass loss' or something similar.

Yes, the manuscript will be corrected accordingly.

L184-190: I think it may be helpful to add figures on the assessment of oceanic internal climate variability based on 60-year averages, at least in form of a Supplementary Material, given that the discussion and recommendations reflect on the time period of averaging.

We agree and we will add the figure for 60 years in the supplementary material to support our arguments.

L197: Please specify that the increased water vapour saturation in warmer air then results in enhanced precipitation.

We will add 'resulting in more precipitation' in the sentence. *« By 2100 and for the SSP2-4.5 medium scenario, run-off is supposed to remain limited (Kittel et al., 2021), so the SMB is projected to increase largely due to the increased water vapour saturation in warmer air, resulting in more precipitation. »*

L200: Is the SMB that you refer to here emulated or directly derived from the CMIP6 models? According to the caption of Figure 5 it is based on the MAR emulation. Maybe this could be specified again also in the main text.

The SMB is an emulation of MAR simulations. We will specify 'emulated SMB' in the main text.

L202: 'consistent' instead of 'consistently'?

Yes, the manuscript will be corrected accordingly.

L202 - 205: This section also refers to the absolute SMB. Since the atmosphere is suggested as an important factor for the (spread in the) projected Antarctic sea-level contribution in the following section, and the choice of the CMIP6 model at the same time also modulates the projected sea-level change, I would like to suggest to add a related figure of SMB and atmospheric temperatures (e.g. in the Supplementary Material) for interested readers.

Thanks for the suggestion. We will add a figure of absolute SMB, surface temperature and precipitation in the Supplementary Material.

L204: 'which is both due to' instead of 'which is due both'?

We will replace 'which is both due to' with 'which results from'.

L205: MPI-ESM1.2-HR also shows a relatively high standard deviation in atmospheric temperature in the Siple Coast region, compared to the other selected CMIP6 models. Is this relevant for the projected future evolution of the Antarctic Ice Sheet? In Figure 7, the ice-sheet response in basin 9 (Siple Coast) to atmospheric changes in MPI-ESM1.2-HR (showing mass loss) differs from the other CMIP6 models (showing mass gain).

The temperature variability appears to be linked to the variability in the ASL position between members (see Figure 5g), which could explain why large anomalies are found in all the basins influenced by the ASL (basins 9,10,11 in Fig. 7).

L205-208: What are the typical characteristics of the two Pacific-South American modes? Maybe add a short summary here for readers that are not familiar with Wang et al. (2022) and Marshall and Thompson (2016).

We consider that this would be going too far away from the main focus and our sentence already gives a lot of information « *As previously reported by Marshall and Thompson(2016), the internal climate variability of sea level pressure and air temperature have the typical characteristics of the two Pacific-South American modes (usually referred to as PSA1 and PSA2), which are associated with wave trains originating in the tropical Pacific and possibly modulated by feedbacks with clouds and sea-ice (Wang et al., 2022)* ».

L210-227: Maybe the link between the analysis of internal climate variability in CMIP6 models and the projected Antarctic sea-level contribution could be stressed here, in particular, for explaining some of the key results related to the uncertainties in the projected contribution of the Antarctic Ice Sheet to sea-level change. This includes, for example, the results that (1) atmospheric internal climate variability has a larger effect on the spread in the projected sea-level change with Elmer/Ice than oceanic internal climate variability, (2) the impact of the

choice of the CMIP6 model on the sea-level contribution from Antarctica, and (3) the similarity of atmospheric internal climate variability for the selected CMIP6 models.

For (1), this was already specified in L223-227.

For the other two points, we will rephrase this section to stress the link between the analysis of internal climate variability in the CMIP6 models and the projected Antarctic Sea level contribution.

L213: For MPI-ESM1.2-HR there is a mean (?) mass loss related to the atmosphere in West Antarctica (Fig. 6i).

We used only 2 ensemble members for MPI-ESM1.2-HR. One of them projects a positive SMB sea-level contribution and the other one a negative contribution (Fig. 6i). Indeed, the average of the two ensemble members indicates a slightly positive contribution to sea levels (Fig. 6i). We will add « *except in West Antarctica for MPI-ESM1.2-HR (Fig. 6i).* »

L215-217: Do you meant to refer to 'Pine Island and Thwaites ice shelves' / 'Getz ice shelf' here or rather the respective basins?

We refer to 'Pine Island and Thwaites ice shelves' / 'Getz ice shelf'. For clarity, we add « *The West Antarctic positive SLC is mostly explained by the grounding line migration and the dynamical response of Pine Island and Thwaites ice shelves (~3 cm in Fig. 5c, basin 11) as well as Getz ice shelf (~1 cm, basin 10).* »

L217-218: Is this drift of the unforced Elmer/Ice experiment removed or is the absolute Antarctic sea-level contribution given in the respective figures? I think I got confused by the statement in L109-110 (please also see my related previous comment). And can the influence of the drift on the trends in East Antarctica be quantified?

The drift of the control Elmer/ice experiment (i.e., Elmer/ice configuration driven by current atmospheric and oceanic forcing) is not removed, which is why we had to correct the friction coefficient to reduce the drift.

All the figures display the absolute Antarctic sea-level contribution.

Based on Table 1, the mass change rate simulated in East Antarctica is equal to +107 Gt/yr whereas the observed mass change is +5±46 Gt/yr, meaning an overestimation between +56 Gt/yr and +148 Gt/yr.

L217: I would like to suggest to replace 'contaminated' by 'influenced' (or something similar).

Yes, the manuscript will be corrected accordingly.

L218: What can be learned on the sensitivity of East Antarctica and the Antarctic Peninsula to internal climate variability based on the simulations presented here? Maybe you can make use of Figure 7 and add some details in this section.

We can complete the analysis for the three regions (East Antarctica and Antarctic Peninsula in addition to West Antarctica) on the same model as the analysis carried out for West Antarctica.

L220-222: I would like to suggest to give the full name of the CMIP6 models throughout the whole manuscript (consistent with e.g. Sect. 3.1).

For sure, the manuscript will be corrected accordingly.

L223: Basin 5 (including Totten glacier) shows a relatively large spread in the dynamical sea-level contribution (Fig. 7b). Can this be related to the assessment of oceanic internal climate variability in Sect. 3.2?

Yes, the ocean variability in Fig. 3e-f is particularly strong near Totten, which explains why ocean has a larger effect on internal variability there. This will be mentioned in the revised manuscript.

L226-227: Please add more information on this finding. How is the number determined? Can it be seen in a figure (likely Figure 6)?

For each of the three CMIP6 models, we compare the amplitude of sea-level contribution (SLC) variability induced by ice flow dynamics, which is largely modulated by ocean-induced basal melting (Figure 6b), with the amplitude of the SLC variability due to the surface mass balance (Figure 6c), which is largely driven by the atmosphere.

When we talk about amplitude, we mean the difference between the SLC value in 2100 of the member giving the smallest contribution and the member giving the largest contribution.

We will add a reference to Figure 6 in the main text for a clearer explanation « *On average, by the end of the century, the amplitude of SLC variability due to the atmosphere (Fig.6c) is 3.4 times higher than the variability due to the ocean (Fig.6b).* »

Figure 6: I assume that the number in brackets in the legend refers to the number of ensemble members for each CMIP6 model. Why do the numbers differ between panel a/b and panel c? Please check.

You are completely right, the number in brackets in the legend refers to the number of ensemble members for each CMIP6 model. We will add it to the legend.

For panel b): « *Because of the numerical cost of our simulations, we select a limited number of members. The selection is made over the current period (1995-2014 means) to cover the widest range of values for the ocean temperature on the continental shelf in the Amundsen Sea. We focus on this region as (i) observed present-day ocean-induced mass loss and (ii) the amplitude of the across member standard deviation of the 1995-2014 mean potential temperature are among the highest around Antarctica (see Fig. 3d-f, j-l).*

*We selected 5 members for each CMIP6 model, i.e., the two members with the coldest temperatures and the two members with the warmest temperatures in the Amundsen Sea continental shelf (Fig. 3j-l), for which the SMB is available, as well as member 1, used by*

*default in most of the studies. In total, we run 11 simulations, five with the IPSL-CM6A-LR model (r1i1p1f1, r3i1p1f1, r6i1p1f1, r11i1p1f1, r25i1p1f1, see https://goo.gl/v1drZl for CMIP6 convention name of ensemble members), four with the UKESM1-0-LL model as member 1 is already included in the temperature criteria (r1i1p1f2, r2i1p1f2, r4i1p1f2, r8i1p1f2) and two with the MPI-ESM1.2-HR model (r1i1p1f2, r2i1p1f2). As the oceanic variability is very low in MPI-ESM1.2-HR (Fig. 3f), we retained only member 1 and another member to verify the low impact of oceanic variability on the dynamic contribution. »*

For panel c): we take into account all the members available.

For panel a): the total contribution is a combination of dynamical contribution and SMB contribution, so the number of ensemble member for the total contribution depends on the limited ensemble members of the dynamical contribution and thus have the same number of ensemble members.

Figure 6: Does the solid line indicate the mean? Maybe I have missed this.

We forgot to include this information in the legend of Figure 6. The manuscript will be corrected accordingly.

L232-233: Maybe specify which paleoclimate proxies are used in Parsons et al. (2020) (similar to stating that Casado et al. 2023 base their analysis on ice core reconstructions in the following paragraph), for readers that are not familiar with this study?

Yes, the manuscript will be corrected accordingly.

L232: 'global mean surface air temperature' or its variability?

Thank you for the careful reading, the word 'standard deviation' is missing.

*We will rephrase as « Parsons et al. (2020) compared the distribution of standard deviation of global mean surface air temperature of CMIP piControl simulations to paleoclimate proxies representative of the 1450-1849 period. »*

L233: Maybe you could add the observational plausible range for the temperature variability for comparison with the values for the CMIP6 models?

Yes, the manuscript will be corrected accordingly.

L243: If possible, maybe a conclusion on the representation of atmospheric variability in CMIP models could be added, bringing together the results of this study (Sect. 3.1) with the previous literature?

We will generalise the assessment of internal oceanic and atmospheric variability carried out in §3.1 to 15 CMIP6 models:

- for ocean: calculation of across member standard deviation of the 1995-2014 mean potential temperature over the whole continental shelf for the 200-700m depth,

 - for atmosphere: calculation of across member standard deviation of the 1995-2014 mean SMB over the whole Antarctica.

This will allow us to see how the three selected models stand in comparison to a larger set of CMIP6 models. The new figure will be discussed in §4.1 and added in supplementary material.

L245: Maybe add a reference for these observations?

We will add a reference to CTD profiles measured in the Amundsen Sea and described in Dutrieux et al., 2014 and Jenkins et al., 2018.

L258-259: As the choice of the CMIP6 model is suggested to have a similar impact on the Antarctic sea-level contribution as the internal climate variability, it may be helpful to add a short paragraph on this finding also in Sect. 3.3 (in addition to this statement in the discussion).

Yes, the manuscript will be corrected accordingly.

L265-266: This sentence can maybe be reformulated. As already indicated, given the limited impact of the emission pathway on the Antarctic sea-level contribution to 2100, SSP2.4.5 may not be the main explanation for the Elmer/Ice projections presented here being at the lower end of previous projections.

We will remove the argument of GHG scenario in the sentence and focus mainly on the present-day drift and we will mention the sources of uncertainty described in Seroussi et al., (2023), i.e., uncertainties in the physics of the ice-sheet model, the choice of climate model and uncertainties associated with ice-climate interaction (melt parameterisation and calibration).

L269: I would like to suggest 'ocean-induced melting' (or something similar).

Yes, the manuscript will be corrected accordingly.

L270: I am not sure if I understand the meaning of 'high variability of 20-year means' correctly. Maybe it is possible to rephrase?

We will replace with « *wide confidence interval on a 20-year mean* ».

L278-283: This paragraph seems to contain much information that is also given in the beginning of the following Sect. 4.3. I would like to suggest to move L278-283 to Sect. 4.3 and merge with the first part of this section.

Yes, the manuscript will be corrected accordingly.

L284-354: This is an interesting analysis and discussion. If I understand correctly, it supports to include multiple CMIP ensemble members in Antarctic sea-level projections as done in the work presented here. At the same time, it seems slightly detached from the previous parts of the manuscript. I would like to suggest two options that may help to add focus to this section:

A) This section may be shortened, summarizing the main analysis and the conclusion. The major part of the analysis may be moved to the Supplementary Material.

B) Parts of this section (e.g., the metrics and justification for these metrics) may be included in the Methods, and the outcomes could be highlighted and discussed with the main results. If applicable, the Antarctic sea-level contribution for the 'best' ensemble members could be added separately to e.g. Figure 6.

Thanks for the suggestion. We choose option A).

For a better readability and following suggestions by the other reviewers, the discussion section on the identification of the best member will be revised. Figure 8 and the list of associated metrics will be moved to the appendix and the same analysis will be perform for UKESM-0-LL to get more robust conclusions.

L290: I am not sure if I understand this phrase. Maybe replace 'assess' by e.g. 'demonstrate'?

In the revised paragraph, we will specify that the metrics considered were chosen to:

– ensure a good representation of the mean atmospheric and oceanic states. We selected variables directly used to drive the ice sheet model, i.e., the SMB for the atmosphere and temperature for the ocean. We focused on the ocean temperature in the Amundsen sector as the region experiences the current main mass loss and CTD profile data are available for a relatively long period from 1994 to 2018 in this area.

– ensure a good representation of the amplitude of oceanic variability using the same observational data described in the previous paragraph. We did not evaluate the variability of SMB since it has been relatively stable in recent years and there is no observational data.

– ensure the best representation of important modes of variability known to affect the ocean and atmosphere in/around Antarctica. We focus our analyses on the SAM and TPI index.

– ensure a phasing of internal variability with observations, which could be important for future detection/attribution studies and for projected Antarctic sea-level contribution. We chose two variables, sea-ice concentration and the presence of warm periods on the continental shelf of the Amundsen Sea to provide insights on the phasing of internal variability.

It should be noted that this part of our study remains exploratory and the choice of variables and metrics is indeed very subjective, which is part of the caveats that we discuss.

L368-384: This paragraph contains many valid and helpful recommendations for future assessments of the Antarctic contribution to sea-level change. However, some of these recommendations do not seem to be directly justified by the presented results, or a better link to and additional information in Sect. 3 may be needed. For example, a fully-coupled assessment would be ideal to include feedbacks of the ice sheet with the ocean and the atmosphere, but some additional discussion how this would e.g. improve the representation of or remove biases in the internal climate variability in the selected CMIP6 models presented in Sect. 3.1 and Sect. 3.2 may be needed to directly relate to this study.

Following the comments of all the reviewers, we have only kept the two recommendations that are directly motivated by our findings, and we will make the links with our findings more explicit.

L379-380 / L382: Do the 'various members' / 'multiple members' refer to the CMIP6 model ensemble members or to ice-sheet initial states?

This sentence will be removed from the revised manuscript.

---

## Author Comment (AC3)

We thank the Editor Claudia Timmreck and the three reviewers for their careful evaluation of our manuscript. We found the comments very useful and think that our manuscript will be greatly improved thanks to them. To ensure clarity, the reviewer's comments are written in black and our responses in light blue.

**Reviewer #1**

**General comments:**

The submitted manuscript investigates the impact of internal climate (oceanic and atmospheric) variability in projections of Antarctica's sea-level contribution until year 2100. For this purpose the authors run standalone ice-sheet simulations applying output from a selection out of ensemble simulations of three CMIP6 climate models. Besides quantifying the relevance of the internal climate variability for sea-level projections, the authors also give recommendations for future ice-sheet projections. I deem the study a valuable contribution to the Earth System modeling community, especially the ice-sheet modeling community. I find the manuscript clearly written, well structured and mostly understandible. The figures illustrate the findings well. The methodology seems consistent and the conclusions plausible. I would like to mention that my assessment is limited regarding some oceanic mechanisms described in the study (which I refer to in my specific comments) and details of the applied metrics in Sect. 4.3. They seem plausible but I didn't have the time to dig deeper into the details.

I would support the publication of the manuscripts after my few points below have been addressed.

**Specific comments:**

The last sentence of the abstract seems a bit detached from the rest of the abstract. Maybe a different introduction of the sentence would help for a smoother reading.

Indeed, the transition to the last sentence is a bit abrupt. We will replace « *We then issue recommendations for future ice-sheet projections: use several members... »* with « *Based on these results, we recommend that ice sheet model projections consider several... »*

L28: Is there a typo: "of" instead of "on"?

Yes, the manuscript will be corrected accordingly.

L50: Why SPP2-4.5?

We will clarify the choice of scenario in §2.3 rather than at the end of the introduction as follows « *We use the medium SSP2-4.5 scenario, which corresponds to a global warming of 1.4 to 3.0°C from 1995-2014 to 2081-2100 (90% confidence interval, Lee et al. 2021) and seems the most representative of current effort to tackle climate change (Riahi et al., 2017). As the choice of greenhouse gas emission scenario has only a limited impact on the projected Antarctic contribution to sea level rise until 2100 (Seroussi et al., 2020), we have not repeated our calculations for other scenarios. »*

L56: drivers

Yes, the manuscript will be corrected accordingly.

L82: Which friction law is it? I don't see the necessity to write the law down here but a reference to the equation would be helpful.

Since this comment appeared several times, we will add the law in the main text for clarity as follows « *The ice dynamics is computed by solving the Shallow Shelf Approximation (SSA) of the Stokes equations (MacAyeal, 1989), assuming an isotropic rheology following Glen's flow law (Glen, 1955) and a linear friction law (i.e., $\tau_b = C\, u_b$ where $\tau_b$ is the basal shear stress, $C$ the friction coefficient and $u_b$ the basal ice velocity).* »

L120: What's the sense of these abbreviations?

If we are correct, you are referring to the abbreviations on lines L121-122 for the names of CMIP6 experiments rXiXpXfX. These abbreviations are a CMIP convention and are a brief description of the experiment. rX corresponds to realization index (i.e., the member number), iX to initialisation index, pX to physics index and fX to the forcing index.

We will add, in the main text, a link to a URL ([https://goo.gl/v1drZl](https://goo.gl/v1drZl)) that describes the CMIP6 writing conventions for the attributes.

L164-166: I am not able to follow the causal chain. Which location do the authors mean by "there" at the end of the sentence (I guess the Eastern Ross Sea)? I am not an expert on such oceanic mechanisms/patterns and personally would be glad to get a clearer explaination.

We will replace « *As described by Mathiot and Jourdain (2023) and Siahaan et al. (2022), lower rates of HSSW formation in the eastern Ross Sea can favor the intrusion of Circumpolar Deep Water (CDW) in the western part, which may explain why mid-depth temperature variability is highest there (Fig. 3e)* » with « *The deepest part of the Ross Sea is occupied by the densest water mass, so that there is a competition between intrusions of relatively warm and salty Circumpolar Deep Water (CDW) advected from offshore and the production of cold dense water (HSSW) through sea ice formation and associated convection (Mathiot and Jourdain, 2023 ; Siahaan et al. 2022). The variation between the occupation of these two water masses may explain the high mid-depth temperature variability in the Ross Sea (Fig. 3e).* »

L179: Again, as a non-specialist regarding the ocean: What does it mean when you state "Both IPSL-CM6A-LR and UKESM1-0-LL seem to be prone to convection"?

We will replace the current sentence with « *The weaker stratification of IPSL-CM6A-LR and UKESM1-0-LL than in MPI-ESM1.2-HR indicates the presence of more convective mixing, as convection mixes cold and salty water produced by sea ice formation with warmer water at depth.* »

L224-225: "On average, the amplitude of SLC variability relative to the atmosphere is 3.4 times higher than that relative to the ocean." This sentence is not entirely clear to me and I would

appreciate if the authors could briefly explain what is exactly meant. How is this finding deduced? Is it shown in a figure which could be referenced here?

For each of the three CMIP6 models, we compare the amplitude of sea-level contribution (SLC) variability induced by ice flow dynamics, which is largely modulated by ocean-induced basal melting (Figure 6b), with the amplitude of the SLC variability due to the surface mass balance (Figure 6c), which is largely driven by the atmosphere. When we talk about amplitude, we mean the difference between the SLC value in 2100 of the member giving the smallest contribution and the member giving the largest contribution.

We will add a reference to Figure 6 in the main text for a clearer explanation as follows *« On average, by the end of the century, the amplitude of SLC variability due to the atmosphere (Fig.6c) is 3.4 times higher than the variability due to the ocean (Fig.6b). »*

L258-259: How is this finding deduced? If I am right it can be seen from Fig.6a?

You are entirely right, this statement is directly based on Figure 6a. We compared the amplitude due to internal variability (i.e., amplitude of the difference between the SLC value in 2100 of the member giving the smallest contribution and the member giving the largest contribution) with the amplitude related to the choice of climate model (i.e., amplitude of the difference between the multi-member mean SLC value in 2100 of the climate model giving the smallest contribution and the climate giving the largest contribution).

We suggest to replace *« Our simulations, involving three climate models, suggest that the choice of climate model and internal climate variability both have a similar impact on Antarctic SLC, although there are disparities on a finer scale »* with *« The comparison of the amplitude of SLC in 2100 due to internal variability (shaded area in Figure 6a) with the one due to the choice of climate model (difference between extreme thick lines in Fig. 6a) shows that the choice of climate model and internal climate variability both have a similar impact on Antarctic SLC. »*

---

## Author Response (AR1)

We thank the Editor Claudia Timmreck and the three reviewers for their careful evaluation of our manuscript. We found the comments very useful and think that our manuscript will be greatly improved thanks to them. To ensure clarity, the reviewer's comments are written in black and our responses in light blue.

**Reviewer #1**

**General comments:**

The submitted manuscript investigates the impact of internal climate (oceanic and atmospheric) variability in projections of Antarctica's sea-level contribution until year 2100. For this purpose the authors run standalone ice-sheet simulations applying output from a selection out of ensemble simulations of three CMIP6 climate models. Besides quantifying the relevance of the internal climate variability for sea-level projections, the authors also give recommendations for future ice-sheet projections. I deem the study a valuable contribution to the Earth System modeling community, especially the ice-sheet modeling community. I find the manuscript clearly written, well structured and mostly understandible. The figures illustrate the findings well. The methodology seems consistent and the conclusions plausible. I would like to mention that my assessment is limited regarding some oceanic mechanisms described in the study (which I refer to in my specific comments) and details of the applied metrics in Sect. 4.3. They seem plausible but I didn't have the time to dig deeper into the details.

I would support the publication of the manuscripts after my few points below have been addressed.

**Specific comments:**

The last sentence of the abstract seems a bit detached from the rest of the abstract. Maybe a different introduction of the sentence would help for a smoother reading.

Indeed, the transition to the last sentence is a bit abrupt. We have replaced « *We then issue recommendations for future ice-sheet projections: use several members use several members in the run and in its initialisation, favor 50-year averages to correct or weight simulations over the present-day period, and couple ice-sheet and climate models.* » with « *Based on these results, we recommend that ice-sheet model projections consider (i) several climate models and several members of a single climate model to account for the impact of internal climate variability and (ii) longer temporal period when correcting historical climate forcing to match present-day observations.* » (L16-19).

L28: Is there a typo: "of" instead of "on"?

Yes, the manuscript has been corrected accordingly (L31).

L50: Why SPP2-4.5?

We have clarified the choice of SPP scenario in §2.3 (L125-128) rather than at the end of the introduction as follows « *We use the medium SSP2-4.5 scenario, which corresponds to a global warming of 1.4 to 3.0°C from 1995-2014 to 2081-2100 (90% confidence interval, Lee et al., 2021) and seems the most representative of current efforts to tackle climate change (Riahi et al., 2017). As the choice of greenhouse gas emission scenario has only a limited impact on the projected Antarctic contribution to sea-level rise until 2100 (Seroussi et al., 2020), we have not repeated our calculations for other scenarios.* »

L56: drivers

The sentence including 'driver' has been replaced and the comment no longer applies.

L82: Which friction law is it? I don't see the necessity to write the law down here but a reference to the equation would be helpful.

Since this comment appeared several times, we added the law in the main text for clarity (L78-80) as follows *« The ice dynamics is computed by solving the Shallow Shelf Approximation (SSA) of the Stokes equations (MacAyeal, 1989), assuming an isotropic rheology following Glen's flow law (Glen, 1955) and a linear friction law (i.e., $\tau_b = C u_b$ where $\tau_b$ is the basal shear stress, $C$ the friction coefficient and $u_b$ the basal velocity). »*

L120: What's the sense of these abbreviations?

If we're correct, you are referring to the abbreviations on lines L121-122 of the initial manuscript for the names of CMIP6 experiments rXiXpXfX. These abbreviations are a CMIP convention and are a brief description of the experiment. rX corresponds to realization index (i.e., the member number), iX to initialisation index, pX to physics index and fX to the forcing index.

We have added, in the main text (L134), a link to a URL (https://goo.gl/v1drZl) that describes the CMIP6 writing conventions for the attributes.

L164-166: I am not able to follow the causal chain. Which location do the authors mean by "there" at the end of the sentence (I guess the Eastern Ross Sea)? I am not an expert on such oceanic mechanisms/patterns and personally would be glad to get a clearer explaination.

We have replaced *« As described by Mathiot and Jourdain (2023) and Siahaan et al. (2022), lower rates of HSSW formation in the eastern Ross Sea can favor the intrusion of Circumpolar Deep Water (CDW) in the western part, which may explain why mid-depth temperature variability is highest there (Fig. 3e) »* with *« The deepest part of the Ross Sea is occupied by the densest water mass, so that there is a competition between intrusions of relatively warm and salty Circumpolar Deep Water (CDW) advected from offshore and the production of cold dense water (HSSW) through sea-ice formation and associated convection (Siahaan et al., 2022 ; Mathiot and Jourdain, 2023). The variation between the occupation of these two water masses may explain the high mid-depth temperature variability in the Ross Sea (Fig. 2e). »* (L178-182).

L179: Again, as a non-specialist regarding the ocean: What does it mean when you state "Both IPSL-CM6A-LR and UKESM1-0-LL seem to be prone to convection"?

We have replaced the current sentence with *« The weaker stratification in IPSL-CM6A-LR and UKESM1-0-LL than in MPI-ESM1.2-HR indicates the presence of more convective mixing, as convection mixes cold and salty water produced by sea-ice formation with warmer water at depth. Consequently, both IPSL-CM6A-LR and UKESM1-0-LL exhibit more realistic temperature profiles than MPI-ESM1.2-HR in the Amundsen Sea. »* (L195-198)

L224-225: "On average, the amplitude of SLC variability relative to the atmosphere is 3.4 times higher than that relative to the ocean." This sentence is not entirely clear to me and I would appreciate if the authors could briefly explain what is exactly meant. How is this finding deduced? Is it shown in a figure which could be referenced here?

For each of the three CMIP6 models, we compared the amplitude of sea-level contribution (SLC) variability induced by ice flow dynamics, which is largely modulated by ocean-induced

basal melting (Figure 5b), with the amplitude of the SLC variability due to the surface mass balance (Figure 5c), which is largely driven by the atmosphere. When we talk about amplitude, we mean the difference between the SLC value in 2100 of the member giving the smallest contribution and the member giving the largest contribution.

We have added a reference to Figure 5 in the main text for a clearer explanation as follows *« On average, by the end of the century, the amplitude of SLC variability related to the atmosphere (Fig. 5c) is 3.4 times higher than that related to the ocean (Fig. 5b). »* (L245-246).

L258-259: How is this finding deduced? If I am right it can be seen from Fig.6a?

You're entirely right, this statement is directly based on Figure 6a (now Figure 5a in the new version of the manuscript). We compared the amplitude due to internal variability (i.e., amplitude of the difference between the SLC value in 2100 of the member giving the smallest contribution and the member giving the largest contribution) with the amplitude related to the choice of climate model (i.e., amplitude of the difference between the multi-member mean SLC value in 2100 of the climate model giving the smallest contribution and the climate giving the largest contribution).

We have replaced *« Our simulations, involving three climate models, suggest that the choice of climate model and internal climate variability both have a similar impact on Antarctic SLC, although there are disparities on a finer scale »* with *« The comparison of the amplitude of SLC in 2100 due to internal variability (shaded area in Fig. 5a) with the one due to the choice of climate model (difference between extreme thick lines, Fig. 5a) shows that the choice of climate model and internal climate variability both have a similar impact on Antarctic SLC. »* (L296-298).

The submitted manuscript describes a model investigation of the uncertainties in medium-range projections of the Antarctica Ice Sheet's contribution to sea level. The study focuses on uncertainties related to internal climate variability of climate forcing, both ocean and atmospheric. First, the authors evaluate CMIP6 models, and choose a subset on which to conduct their analysis. They then present historical diagnostics on the available model ensemble for each climate model, extracting the temporal variability of various ocean and atmospheric related variables. Finally, they choose a subset of ensemble members of the SSP2-4.5 scenario for each climate model and use them to force an Elmer/Ice continental Antarctica, resulting in an ensemble of projections for Antarctic Ice Sheet sea-level contribution through year 2100. Results suggest that internal climate variability can affect sea-level contribution, ranging in magnitude from 45-93%, but most of that uncertainty is dominated by atmospheric forcing over ocean forcing. The authors conclude that internal climate variability varies among the climate models, especially for the ocean forcing; therefore, they suggest a strategy for choosing ensemble members that most realistically represent the dominant climate modes of the Antarctic region. They also make recommendations for how to best consider internal climate variability in ice sheet model projections. In general, the methods are well-described and the figures are adequately presented. The analyses and science results are of high quality, and the discussion and conclusion bring up intriguing and relevant points for the ice sheet modeling community.

Overall, I find that this is an interesting study, with important results comparing the effect of internal climate variability due to the ocean and the atmosphere on ice sheet model projections. While results presenting ice sheet modeling projections by themselves could constitute their own manuscript, the authors present much more analysis, including a list of metrics for choosing appropriate model ensemble members to capture internal climate variability. While interesting, these metrics are not the ones used for choosing members for the ensemble results presented. In addition, the authors do not show outcomes that illustrate/quantify the consequences resulting from an ice sheet model using all the suggested updates to their projection procedures. As a result, I find that the addition of these extra results leads to a manuscript that lacks focus. For instance, I think it would benefit the manuscript if some of the secondary analyses were moved to a supplement. In this way, the main manuscript could be dedicated to presenting results specifically on the quantification of the uncertainty in Antarctica's sea level contribution due to internal climate variability. If the authors feel as if the new metrics should be highlighted instead, then a new organization and general story built around those results would benefit the manuscript.

Due to the extensive modification needed, I suggest that major revisions be required before the manuscript is accepted. If the authors work on better organization of the results and on improving the clarity of their language (per suggestions outlined below), I am confident that this manuscript could result in a valuable scientific contribution to the community. Please see my comments/suggestions/questions below with regards to my major and minor concerns with the current version of the paper.

**General comments/questions:**

Results show that the ocean internal variability has a minimal effect on the projections, as compared to atmospheric variability or choice of model. If this is the case, why do the designed ensemble metrics mostly focus on evaluating the ocean forcing of ensemble members? More specifically, why is it pertinent to choose members that capture ocean internal variability well if this variability is less important?

We agree and we have thus tried to better balance and explain the respective roles of oceanic and atmospheric contributions:

1. The model ranking for Antarctic atmospheric metrics proposed by Agosta et al., (2024) is now referenced (https://doi.org/10.5281/zenodo.11595213). We have therefore revised the paragraph on the choice of the CMIP6 model (§2.1) to give an equivalent weight to the atmosphere and the ocean (we agree that the paragraph focused more on oceanic properties in the first version).

2. Overall, we agree that internal climate variability mostly affect sea-level projections through the surface mass balance (atmosphere), as evidenced by the shaded ranges in Fig. 5. At the basin scales, things can be different. For example, the oceanic internal climate variability has a stronger effect than the atmospheric variability in some sectors (e.g., basins 1, 5, 9, 10 for the IPSL-CM6A-LR in Fig. 6), and the effects are of comparable magnitude in West Antarctica for the IPSL-CM6A-LR model (Fig. 5h-i).

Do the authors suggest that the climate model ocean representation of internal climate variability lacks skill to the point that using an entire ensemble of forcing does not offer a realistic projection spread? It would improve the manuscript if these questions were considered in the text/discussion/overall story of your paper. It would be even more beneficial to the paper if the authors could support the answers with analysis or results, expanding upon the plots that are already included in the paper.

Assessing the amplitude of the internal climate variability in the ocean is complex. The oceanic internal variability varies greatly depending on the climate models (§3.1 and Fig 2) and is probably underestimated (§4.1), but it is difficult to show this clearly because of the lack of observational data in the ocean over long period. We have generalised the assessment of internal oceanic variability carried out in the §3.1 to 15 CMIP6 models, i.e., calculation of across member standard deviation of the 1995-2014 mean potential temperature over the whole continental shelf for the 200-700m depth, in order to better compare the variability of the selected models with other CMIP6 models. The new figure is discussed in §4.1 and added in Appendix (Appendix B).

We showed (see Figure below) that the multi-member standard deviation of the ocean temperature averaged between 200 and 700 m over the continental shelf varies between 0.02°C and 0.12°C. The MPI-ESM1.2-HR model is one of the models with the lowest ocean variability (0.02°C), UKESM1-0-LL is close to the median (0.04°C) and IPSL-CM6A-LR is one of the models with the highest variability (0.07°C).

[Figure]

**Figure 1: Assessment of 1995-2014 multi-member mean and standard deviation of Antarctic air temperature at 2 m (left) and circum-Antarctic ocean temperature between 200 and 700 m depth on the continental shelf (right) in 15 CMIP6 models. The number of members for each model is in brackets. When two numbers are indicated, they correspond to the available members for the atmosphere and ocean, respectively. The black dashed lines represent the observational means, from the ERA5 atmospheric reanalysis (Hersbach et al., 2020) and from the ISMIP6 observational ocean climatology (Jourdain et al., 2020).**

For few regions like the Amundsen Sea, we also have multi-year observations that show a significant variability so that we can consider that a model producing very low variability like MPI-ESM1.2-HR is unrealistic. We nonetheless consider that the variability of IPSL-CM6A-LR may be realistic.

As discussed in your manuscript, climate model ensembles are typically used to represent the spread of model internal climate variability. Forcing the ice sheet model with a large subset of members allows for the propagation of uncertainty due to this variability into projections of sea-level contribution. Here, it is suggested that this might not be appropriate, and that filtering for members that exhibit more realistic variability ("in phase with observed") could be an adopted strategy. Do the authors anticipate that selecting for members would introduce bias into the interpretation of projection uncertainty due to internal climate variability? Is it possible to make runs, or use the runs already completed, to answer this question? (See further questions/comments on this below.)

We made our recommendation clearer. First, ice-sheet models need to be initialised and calibrated to match historical observations. Achieving this would, in theory, be easier with a forcing from the most realistic CMIP ensemble member, which is why we attempted a selection of the best member. Another reason is that ice-sheet simulations can be computationally expensive, and running simulations forced by all members of several CMIP models may not be feasible. However, we agree with the reviewer that once the ice-sheet model is calibrated, the only way to properly assess the uncertainty related to internal climate variability is to force the ice-sheet simulations with multiple members. Our study also gives typical relative errors that may be used as relative uncertainty in studies that can't afford to run full ensemble members.

In the conclusion paragraph, there are four listed recommendations for ice sheet model projections. While these are all pertinent discussion points, it is not clear to me why they are included as conclusions of the presented study. More specifically, these four statements - though they may be valid suggestions - are not directly justified by the results shown. It may

be that the authors believe that they are, and in this case, please rework this section, so that is clear to the reader how the results map to each of these statements; or perhaps additional figures that better illustrate the connection can be included in the manuscript revision.

We have removed the recommendation on coupled ice-sheet/climate models as it is not clearly demonstrated in our study. Instead, the discussion on ice-sheet/climate coupling has been addressed in section 4.1 which deals with the issue of the robustness of internal variability in climate models. The recommendation regarding initialisation has also been removed, as the topic of initialisation is not directly addressed in the paper.

The other two recommendations stem from the results of our study and therefore remain in the conclusion but the link with the present study has been clarified.

**Specific comments/questions:**

Lines 12-14: Please rephrase this sentence. It is awkward and unclear. Also please specify the type of convection you refer to.

We have replaced « *Conversely, the amplitude of oceanic internal climate variability around Antarctica strongly depends on the climate model as underestimated convection, due to either biases in the sea-ice behaviour or in the ocean stratification, leads to weak mid-depth ocean variability* » with « *In contrast, the amplitude of the oceanic component strongly depends on the climate model and its representation of convective mixing in the ocean. A low bias in sea-ice production and an overly stratified ocean lead to a lack of deep convective mixing which results in weak ocean variability near the entrance of ice-shelf cavities.* » (L8-11).

Line 15: Please rephrase to something like: "We recommend based on our results that ice sheet model projections consider …" or something similar.

Thanks for the suggestion. We have replaced « *We then issue recommendations for future ice-sheet projections: use several members...* » with « *Based on these results, we recommend that ice-sheet model projections consider...* » (L16-17).

Line 54: "than" -> "rather than"

We have reformulated the sentence as follows « *We choose to analyse three CMIP6 models to get a more general picture of the internal climate variability than we would get using a single model.* » (L57-58).

Lines 58-64: It is difficult to read this list organized in the current configuration. Is there a way to simplify this so it would be easier to digest for a reader, like in a table for instance?

Figure 1 has been moved to Appendix (Appendix A) and a table summarising the variables and metrics used for the evaluation has been added.

In the main text, we removed the list and replaced it with a sentence including both atmospheric and oceanic properties evaluation « *The selected models are UKESM1-0-LL (19 members, Sellar et al., 2020), MPI-ESM1.2-HR (10 members, Müller et al., 2018) and IPSL-CM6A-LR model (33 members, Boucher et al., 2020). This choice was made based on (i) the size of their ensemble (at least 10 members), (ii) the availability of 6-hourly outputs that were needed to run regional climate projections, and (iii) their representation of the present-day oceanic and atmospheric properties. For the third point, the three selected models are in the best half of the CMIP6 ensemble according to Agosta (2024) who ranked 45 models based on several atmospheric*

*variables relevant for precipitation over Antarctica. These three models also have a high fidelity in the representation of the mean ocean properties, as detailed in Appendix A. »* (L59-65).

Line 66: Please clarify what is meant by "best" here? Can this be quantified?

For each study, the CMIP6 models are ranked by increasing RMSE. In Appendix A, we extended the description of ocean properties analysis a bit further than the response to L58-64 by summarising the metric used, the reference dataset to which it is compared and the period over which they are evaluated.

Line 71: "have some kind of", this wording is very informal and difficult to understand. Does it mean that there is a tuning included for the historical? Please articulate this more clearly for the reader.

We have removed the word 'have some kind of' and replace with *« It is interesting to note that both UKESM1-0-LL and IPSL-CM6A-LR have prescribed ice-shelf melting that is vertically*

*distributed to mimic the presence of unresolved ice-shelf cavities (Mathiot et al., 2017), which is known to be important for coastal ocean properties around Antarctica (Mathiot et al., 2017; Donat-Magnin et al., 2021). »* (L71-73).

Line 74: Please expand upon this in your text, including a summary of the period and in what way they compare well to ERA5.

For further explanation on the evaluation of atmospheric properties, the reader may now refer to Agosta (2024) which has just been referenced : https://doi.org/10.5281/zenodo.11595213.

Agosta (2024) evaluated 45 CMIP models around Antarctica by comparing their performance with the ERA5 reanalysis over the period 1980-2004 for 9 variables. The models are ranked based on two metrics, which are (i) the mean Root Mean Square Error (RMSE) over the 9 variables normalised by the multi-model RMSE and (ii) the second maximum implausible fraction, which corresponds to the fraction of the surface where the difference between CMIP models and ERA5 is greater than three times ERA5 standard deviation.

We have kept paragraph 2.1 concise and refer the reader to Agosta (2024) for the ranking of CMIP6 models according to atmospheric properties and to Appendix A for the evaluation of oceanic properties.

Line 76: Please summarize how these are the best, or add more quantitative language, i.e. the best with reference to what?

see response above for Line 74.

Figure 1: This figure might be better suited for a supplement, since contains more supportive information, based analysis of the climate model runs.

Figure 1 has been moved to Appendix A and a table summarising the variables and metrics used for the evaluation has been added.

Line 82: Please include a reference for the friction law.

As this comment appeared several times, we have added the law in the main text for clarity as follows *« The ice dynamics is computed by solving the Shallow Shelf Approximation (SSA) of the Stokes equations (MacAyeal, 1989), assuming an isotropic rheology following Glen's flow*

*law (Glen, 1955) and a linear friction law (i.e., $\tau_b = C\, u_b$ where $\tau_b$ is the basal shear stress, $C$ the friction coefficient and $u_b$ the basal velocity). »* (L78-80).

Line 85: Please clarify in the text what is meant by curvatures here?

Curvature is the second derivative of the modelled fields (velocity and ice thickness here), i.e., the Hessian matrix. For more explanation, the reader can refer to §2.2 of Gillet-Chaulet et al. (2012):

Anisotropic mesh adaptation is now widely used in numerical simulations especially with finite elements, as it allows to refine the mesh where needed to capture the flow features within a certain accuracy without increasing the computational cost excessively. The method is generally based on an estimation of the interpolation error used to adjust the mesh size so that the discretisation error is equally distributed over the whole domain. It can be shown that an estimate of the interpolation error induced by the meshing is obtained from the Hessian matrix of the modelled field, allowing to define an anisotropic metric tensor at each node (Frey and Alauzet,

In the main text, we replaced the sentence with *« The mesh is refined both close to the grounding line and in areas where observed surface velocities and thickness show high curvatures (i.e., high second derivative of the modelled field, Gillet-Chaulet et al., 2012). »* (L83-84).

Line 104: Please explain in the text more about how this is done. Are there numerical techniques used (inverted?), or is there a procedure designed determine the right correction?

We assumed that this comment is about Line 106. We minimised the RMSE between the modelled and the observed ice-sheet mass change for West Antarctica by applying reduction of the friction coefficient. A proper inversion was done to obtain the initial basal friction coefficients. Then, these coefficients were adjusted by trial and error to limit the model drift.

We have rephrased as: *« In contrast to Hill et al. (2023), we do not correct the surface mass balance to maintain a steady state, but we uniformly lower the inverted friction coefficients by 10% to reduce the model drift. For this, we minimise the RMSE between the modelled and the observed ice-sheet mass change for West Antarctica. »* (L106-109).

Line 107: "correct" -> pleased rephrase this, as this term is not appropriate to describe model results, and if I am reading the rates right, the WAIS trend is still technically outside of the error bounds.

We have removed the word 'correct' and articulated the sentence more clearly *« The resulting model configuration overestimates the mass loss trend in the West Antarctica by only 6% but still largely overestimates mass gain in East Antarctica and in the Peninsula (Tab. 1). As a consequence, the simulated Antarctic Ice Sheet is currently gaining a little mass (+36 Gt yr−1, Tab. 1), instead of losing mass as observed (-109±56 Gt yr−1, Tab. 1). »* (L109-111).

Lines 109-119:  Awkward – please rephrase this last sentence of the paragraph.

We have rephrased « *This mass change trend bias is quite common in ice-sheet models (Seroussi et al., 2020; Aschwanden et al., 2021). This is why our results are primarily analysed in relative terms.* » as « *This growing bias is quite common in ice-sheet models (Seroussi et al., 2020; Aschwanden et al., 2021). However, this bias should not impact most of the analyses presented here, as the projections in response to the CMIP6 climate models are analysed relatively to each other.* » (L112-114).

Lines 112-113: Please specify that this statement is for an Antarctic Ice Sheet model CMIP simulation.

see response for Line 114. We have added '*In standalone ice-sheet simulations*' (L117).

Line 114: This phrasing is confusing for a reader because the sentence before already implies that you can add them together. I think the point is that we can attribute dynamic ice loss to ocean-forced changes, because the SMB driven dynamics is trivial. Please rephrase.

We have rephrased as « *The future mass imbalance of Antarctica results from combined effects of changes in surface mass balance (SMB) and ice dynamics. In standalone ice-sheet simulations, variations in surface mass balance can be attributed to the atmosphere and dynamical mass loss can be attributed to the ocean as SMB changes have little impact on the Antarctic dynamical contribution to sea level over a century (Seroussi et al., 2014, 2023). Thus, the effect of atmospheric and oceanic variations on Antarctic contribution to sea-level change can be analysed separately and then summed to reconstruct the combined effect (Bindschadler120 et al., 2013).* » (L116-121).

Line 120: Please give more specifics on how these ensembles were chosen. Even though "see next section" is included here as a reference, it is unclear where in the next section this information is included. If so, please note the specific section number for clarity.

We have replaced the current sentence with « *Because of the numerical cost of our simulations, we select a limited number of members. In addition to the first member, the selection is made over the current period (1995-2014 means) to cover the widest range of values for the ocean temperature on the continental shelf in the Amundsen Sea. We focus on this region as (i) the largest mass loss is observed there and has been attributed to the ocean, and (ii) the amplitude of the standard deviation of the 1995-2014 mean potential temperature across all members is particularly high in this region (see section 3.1). In total, we run 11 simulations, five with the IPSL-CM6A-LR model (r1i1p1f1, r3i1p1f1, r6i1p1f1, r11i1p1f1, r25i1p1f1, see the CMIP6 naming convention in https://goo.gl/v1drZl), four with the UKESM1-0-LL model (r1i1p1f2, r2i1p1f2, r4i1p1f2, r8i1p1f2), and only two with the MPI-ESM1.2-HR model (r1i1p1f2, r2i1p1f2) given that its oceanic variability is very low (see section 3.1).* » (L129-136).

Line 123: If this statement refers to both types of forcing (atmospheric and ocean), please specify that here, as the sentence is currently vague.

We have replaced « *All the Elmer/Ice simulations start from the same state, corresponding to 2014, and yearly anomalies are added to the present-day forcing to drive future projections as previously done in ISMIP6 (Nowicki et al., 2020).* » with « *All the Elmer/Ice simulations start from the same state, corresponding to 2014, and yearly atmospheric and oceanic anomalies are added to the present-day atmospheric and oceanic forcing to drive future projections as previously done in ISMIP6 (Nowicki et al., 2020).* » (L137-139).

Lines 129-135: I do not think this detailed explanation is really needed here. A sentence explaining that MAR output was not available would likely suffice for justification.

We think that this is an important explanation for the ice-sheet community and we would prefer to keep this explanation.

Line 173: Awkward sentence, please rephrase.

We have rephrased the sentence « *We now examine the Amundsen Sea more closely as the region is particularly important for the ice-sheet mass loss.* » like this: « *We now focus on the Amundsen Sea, as the region is currently experiencing the largest mass loss in Antarctica.* » (L189).

Line 184: "instead" -> something like "as well as" or "in addition to"

We replaced « *These conclusions remain valid for 60-year averages instead of 20-year averages, albeit with attenuated internal climate variability.* » with « *These conclusions remain valid for 60-year averages as well as 20-year averages, albeit with attenuated internal climate variability.* » (L202-203).

Line 187: Since one of the conclusions of this paper is related to this point, it would be beneficial to add the not shown plots to the manuscript, instead of just describing them (i.e. for 60 years). Perhaps in a supplement, so that the point can better be made that 60 would be an improvement over 20 years.

We agree. A figure similar to Figure 2 with a 60-year period instead of 20-year period has been added in Appendix C to support our arguments.

Line 197: Please note here that this therefore infers more precipitation.

We added 'resulting in more precipitation' in the sentence « *By 2100 and for the SSP2-4.5 medium scenario, runoff is supposed to remain limited (Kittel et al. , 2021), so the SMB is projected to increase largely due to the increased water vapour saturation in warmer air, resulting in more precipitation (e.g. Krinner et al., 2008; Agosta et al.,2013).* » (L213-216).

Line 198: Please rephrase, e.g., "We therefore focus on variability in SMB components such as precipitation and air temperature."

Yes, thanks for the suggestion. As we have also analysed SMB, we rephrased as « *We therefore focus on variability in emulated SMB and its main components such as precipitation and air temperature.* » (L216).

Line 202: consistently -> consistent

Yes, the manuscript has been corrected accordingly (L220).

Line 203: Awkward, please rephrase this sentence.

We rephrased as « *The largest SMB variability is simulated along the coast of the Amundsen and Bellingshausen seas, which results from the high internal climate variability of atmospheric circulation (e.g., Amundsen Sea Low position) and air temperature in these regions (Fig. 4d-i).* » (L222-224).

Line 216: Please make these statements more specific, e.g., … explained by "grounding line change and dynamic response" of these glaciers.

We rephrased as *« The West Antarctic positive SLC is mostly explained by the dynamical response of Pine Island and Thwaites ice shelves (∼3 cm in Fig. 6c, basin 11) as well as Getz ice shelf (∼1 cm, basin 10). »* (L235-236).

The dynamic response includes both the migration of the grounding line but also the change in flux at the grounding line.

Lines 226-227: relative -> related?  (both instances in the sentence)

Yes, *related* is the right word. The manuscript has been corrected accordingly (L245-246).

Line 252: although -> even though

Yes, the manuscript has been corrected accordingly (L288).

Lines 265-267: The wording here is confusing, please rephrase.

We rephrased like this *« In contrast, our simulations are at the very low end of the ensemble of other ice-sheet projections (-8.5 to -1.3 cm, Fig. 5a). This is partly due to the present-day drift in East Antarctica and Peninsula that we did not remove from our projected trends as opposed to the aforementioned other models. »* (L303-306).

Line 269: It would be helpful to also note in the text that for the 20 years in question, the Antarctic Ice Sheet of reality appeared to be in a stable state, as opposed to its state under the random phase of a climate model you mention that would occur during these same "years" in a climate model.

The ice sheet of reality was not necessarily in a stable state, e.g., it lost mass in the Amundsen Sea sector. But the random phase does change the mass evolution. We have not added anything on this point.

Line 270: Please clarify here what is meant here by high variability of 20-year means.

We have replaced *« However, given the high variability of 20-year means, correcting a random phase of the historical CMIP simulations towards the actual 1995-2014 period may significantly shift the projections. »* with « *However, given the wide confidence interval on a 20-year means ([0.06°C;0.24°C] for air temperature and [0.02°C;0.12°C] for oceanic temperature, see Fig. B1), correcting a random phase of the historical CMIP simulations towards the actual 1995-2014 period may significantly shift the projections.* » (L308-311).

Line 279: "applying" -> "to apply"

Yes, the manuscript has been corrected accordingly (L319).

Line 290: The previous paragraphs justify why this would make sense to do, but please add some discussion about how one would go from that previous logic to each of the outlined decisions here.  More specifically, based on this list of assessments, as a reader I can gather that one should choose the members that have the best representation of important modes of variability known to affect the ocean and atmosphere in/around Antarctica.  Please walk the reader through why these key internal variability metrics are chosen, i.e. what it is in the results that can point to each of these as a justified criterion for ensemble members.

The discussion section on the identification of the best member has been revised. To facilitate reading, Figure 8 and the list of associated metrics has been moved to Appendix E and similar analysis has been performed with UKESM-0-LL to get more robust conclusions.

In the revised paragraph (L330-344), we specified that the metrics considered were chosen to ensure:

- a good representation of the mean atmospheric and oceanic states. We selected variables directly used to drive the ice-sheet model, such as SMB for the atmosphere and temperature for the ocean. For the ocean, we focused our analyses on the Amundsen sector as the region experiences the current main mass loss and CTD profile data are available for a relatively long period from 1994 to 2018.

- a good representation of the amplitude of oceanic variability using the same observational data described in the previous paragraph. We did not evaluate the variability of SMB since it has been relatively stable in recent years.

- a good representation of important modes of variability known to affect the ocean and atmosphere in/around Antarctica. We focus our analyses on the indices representative of the Southern Annular Model and the Interdecadal Pacific Oscillation.

- a good phasing of internal variability with observations, which could be important for future detection/attribution studies and for projected Antarctic contribution to sea-level rise. We chose two variables, sea-ice concentration and the presence of warm periods on the continental shelf of the Amundsen Sea to provide insights on the phasing of internal variability.

It should be noted that this part of our study remains exploratory and the choice of variables and metrics is indeed very subjective, which is part of the caveats that we discuss.

Line 344: Please remind the reader here that this is only for the IPSL members.

Following the response to the previous comment, the research for potential best members was conducted for both IPSL-CM6A-LR and UKESM1-0-LL.

Line 354: Would the results (projections and spread) change if one were to follow the newly created rankings and chose a different group of members for the analysis? I realize it might not be possible to do new runs to completely answer this question, but are there subsets of runs that are already completed that can be used (i.e. redo analysis of runs with top half vs. bottom half of members?). Or even still, in your discussion, is it possible to use the results presented and extrapolate the results to suggest how much it would matter to the projections to use the ranked members instead?

We believe that we do not have enough simulations to divide our set into two subsets (only two members in each subset for UKESM and IPSL and one for MPI).

Line 367: Please specify what is meant by convection here, for example where, and at what depth?

For clarity, we replaced 'convection' by 'ocean convective mixing on the continental shelf' (L373). The entire water column is affected.

Line 368: Please rephrase, i.e. changing -> "using the following practices to remove the effects of internal climate variability on projection results…"

Based on the comments of all the reviewers, we decided to only keep two recommendations in the conclusion, so this part has been reshaped.

Lines 369-372: Here, metrics to find ensemble members that are in phase with observed variability are described. Please comment on the consequences of filtering these members. In a way, this could defeat the purpose of using a large ensemble, because it discounts variability that is intrinsic within a climate model (even if it is a variability that is not realistic or wrong). Is doing a filtering justified because results suggest that the ocean models are so wrong (i.e. the various members are all over the place in terms of variability and not at all "realistic"), that they no longer offer trustworthy projections? Is it important to rank climate models themselves based on metrics of variability we can derive from their ensemble, in order to tell the community which ones might have completely unrealistic internal variability? Adding these concepts to the discussion would broaden and enhance the scope and impact of the presented work.

Thanks for the valuable suggestion. First of all, a justification for filtering members in the calculation of metrics is that ice sheets act as low-pass filters, they are not very sensitive to interannual forcing. We do not suggest that the variability is wrong in all ocean components, this conclusion only holds for MPI-ESM1.2-HR. Second, we agree that it is important to show where these 3 models stand in terms of internal climate variability, and we have added a comparison to a larger set of CMIP6 models in the revised manuscript (see Appendix B).

Line 374: Please add some text on why this is so, i.e., add a "because …". It would be helpful to clearly state your reasoning here. For example, 1) climate models show important modes longer than 20 years, and/or 2) the 20 years in an observational period that may seem appropriate in "reality" may not align with an appropriate period within a climate model.

We have rephrased the recommendation like this *« The use of longer reference period for the calculation of anomalies than that usually used (e.g., 20 years in ISMIP Nowicki et al., 2020) as climate models show important modes of variability longer than 20 years. Casado et al. (2023) recommend averaging over 50 years to be long enough to weaken internal climate variability and short enough not to dilute forced trends. Few observations were available 50 years ago in Antarctica, so the observational climatologies will likely remain representative of 20-30 years. This nonetheless likely remains a preferable approach than using the last 20 years. »* (L381-386).

Line 377: "but not in the observations", following this, please add something like "and therefore introduce a different source of bias."

See response to Line 374.

Line 382: "Ice-sheet models should" -> something like "To capture the full uncertainty due to internal climate variability, ice sheet models would ideally be …"

This recommendation has been removed.

**Reviewer #3**

**Summary**

The study by Caillet et al. quantifies the uncertainties in the projected Antarctic contribution to sea-level change by 2100 related to internal climate variability in a subset of CMIP6 models. Three CMIP6 models are selected based on a summary of previous evaluations (Purich and England, 2021; Beadling et al., 2020; Heuzé, 2021; Sect. 2.1). The Antarctic sea-level contribution is projected with the stand-alone ice-sheet model Elmer/Ice; the respective experimental setup is presented in Sect. 2.2 and Sect. 2.3. First, internal climate variability in the selected CMIP6 models is explored (Sect. 3.1 and Sect.3.2). Then, the Antarctic contribution to sea-level change projected by Elmer/Ice based on different ensembles members for the selected CMIP6 models is presented (Sect. 3.3). The authors quantify the effect of internal climate variability on the Antarctic sea-level contribution by the end of this century with 45% to 93%, with a higher impact from atmospheric variability compared to ocean variability, and modulated by the CMIP6 model. Results are discussed in terms of the robust representation of internal climate variability in CMIP models (Sect. 4.1), the internal climate variability as a source of uncertainty in Antarctic sea-level projections (Sect. 4.2), and identifying best ensemble members as an alternative approach to account for internal climate variability in sea-level projections (Sect. 4.3). The authors conclude with general recommendations for future assessments of the Antarctic contribution to sea-level change (Sect. 5).

**General comments**

By bringing together internal climate variability and the future evolution of the Antarctic Ice Sheet, the paper addresses a relevant and scientific interesting question, that has rarely been explored in previous assessments of the future trajectory of the Antarctic Ice Sheet. The presented results and related discussion may be valuable for future assessments of the Antarctic contribution to sea-level rise. The title clearly reflects the contents of the paper. The abstract provides a concise and complete summary. Overall, the study has a sound methodology and experimental setup. In some cases, the description of the methods could be more precise, and the clarity of language improved. While the results span a wide range from exploring the representation of internal climate variability in selected CMIP6 models to the modelled response of the Antarctic Ice Sheet, the manuscript may benefit from linking both in greater depth (e.g. explaining the climate model - dependence of the impact of climate variability on the projected sea-level contribution, ranging between 45% and 93%, if possible). Here, additional figures, e.g. in a Supplementary Material, may be helpful for the reader.

We worked on section 3.3 to better stress the links between the analysis of internal climate variability in the CMIP6 models and the projected Antarctic sea-level contribution.

In addition, the discussion on a possible selection of best ensemble members from CMIP models could be better integrated in the manuscript.

For a better readability and following suggestions by the other reviewers, the discussion section on the identification of the best member has been revised. Figure 8 and the list of associated metrics have been moved to the appendix and the same analysis has been perform for UKESM-0-LL to get more robust conclusions.

The reasons for choosing the best member have also been better explained. First, ice sheet models need to be initialised and calibrated to match historical observations. Achieving this

would in theory be easier with a forcing from the most realistic CMIP ensemble member, which is why we attempted a selection of the best member. Another reason is that ice-sheet simulations can be computationally expensive, and running simulations forced by all members of several CMIP models may not be feasible.

Finally, some additional explanations may be needed to directly derive and support some of the recommendations given in the conclusion based on the results presented in this study.

Based on the comments of all the reviewers, we decided to only keep the two recommendations that are the most motivated by our findings, and we made sure that the link with our findings is explicit.

We have removed the recommendation on coupled ice-sheet/climate models as it is not clearly demonstrated in our study. Instead, the discussion on ice-sheet/climate coupling has been addressed in section 4.1 which deals with the issue of the robustness of internal variability in climate models. The recommendation regarding initialisation was also removed, as the topic of initialisation is not directly addressed in the paper.

I have included more specific comments, questions and suggestions below.

**Specific comments**

L12-14: In the abstract, the results of the sea-level projections are summarized before describing the internal climate variability in the CMIP6 models. Maybe it would be more intuitive to follow the same order as in the main text (that is, internal climate variability in CMIP6 models followed by sea-level projections)?

We have rearranged the abstract to match the structure of the paper.

L12-14: Maybe a brief remark on the upper end of the amplitude of oceanic internal variability covered by different climate models could be added, in addition to the mentioned and explained weak mid-depth ocean variability?

We added the range of the ratio between the amplitude of internal atmospheric variability and internal oceanic variability for the various models. This ratio varies from 2 for IPSL-CM6A-LR to 5 for MPI-ESM1.2-HR (L15).

We have also tempered our statement for Dronning Maud area and Amundsen, Getz and Aurora basins, where the amplitude of oceanic and atmospheric variability may be similar depending on the CMIP model (L15-16).

L15: Please specify 'use of several members in the run and its initialisation'. I think I understand what is meant here after reading the manuscript but this phrase may be unclear for the reader when starting with the abstract.

Based on the general comments described above, we have reformulated the recommendations by focusing only on those that are really motivated by our results. We rephrased like this « *Based on these results, we recommend that ice-sheet model projections consider (i) several climate models and several members of a single climate model to account for the impact of internal climate variability and (ii) longer temporal period when correcting historical climate forcing to match present-day observations.* » (L16-19).

L24: Maybe add 'estimates of' or 'projections of', e.g. 'Estimates of the AIS contribution to future sea level rise are currently based...'

We replaced *« The AIS contribution to future sea level rise is currently mostly based on... »* with *« Estimates of the AIS contribution to future sea-level rise are currently mostly based on... »* (L27).

L25: I think CMIP stands for Coupled Model Intercomparison Project. Please check.

Thanks, you're completely right. The manuscript has been corrected accordingly.

L31-35: In this paragraph climate variability is introduced as consisting of two components (1) variability from natural and anthropogenic external forcings and (2) internal variability, and explanations for these components are given, after having referred to internal climate variability in the previous paragraph. Maybe some restructuring is possible to define internal climate variability with its first use?

Internal climate variability is already defined in the abstract (L3).

In the introduction, internal climate variability is used for the first time without having been introduced (L31-33), but its definition is provided in the following sentence (L34). We therefore prefer to keep these things unchanged.

L49/50: Maybe 'the Antarctic Sea Level Contribution'?

Yes, the manuscript has been corrected accordingly (L52/53).

L50: Why did you chose the SSP2-4.5 emission pathway? Please add a short explanation either here or in Sect. 2.3.

We have clarified the choice of SPP scenario in §2.3 (L125-128) rather than at the end of the introduction as follows *« We use the medium SSP2-4.5 scenario, which corresponds to a global warming of 1.4 to 3.0°C from 1995-2014 to 2081-2100 (90% confidence interval, Lee et al., 2021) and seems the most representative of current efforts to tackle climate change (Riahi et al., 2017). As the choice of greenhouse gas emission scenario has only a limited impact on the projected Antarctic contribution to sea-level rise until 2100 (Seroussi et al., 2020), we have not repeated our calculations for other scenarios. »*

L56: 'drivers' instead of 'driver'?

The sentence including 'driver' has been replaced and the comment no longer applies.

L58-64: Please specify the properties that are used to evaluate the CMIP6 models. Some of them are given in Figure 1 or its caption, but it may be helpful to also include them in the main text.

- What properties of ASBW and CDW are evaluated? Please add this information also in the main text.

- Many dynamical features for the Southern Ocean are listed in the legend of Figure 1b. It might be helpful for the reader to better link the legend and caption of Figure 1 to the description in the main text. This applies to e.g. the ocean properties that are evaluated in terms of their meridional gradients.

- What bottom properties of the Southern Ocean are evaluated? Maybe add this information also in the main text.

Figure 1 has been moved to Appendix (Appendix A) and a table summarising the variables and metrics used for the evaluation has been added.

In the main text, we removed the list and replaced it with a sentence including both atmospheric and oceanic properties evaluation « *The selected models are UKESM1-0-LL (19 members, Sellar et al., 2020), MPI-ESM1.2-HR (10 members, Müller et al., 2018) and IPSL-CM6A-LR model (33 members, Boucher et al., 2020). This choice was made based on (i) the size of their ensemble (at least 10 members), (ii) the availability of 6-hourly outputs that were needed to run regional climate projections, and (iii) their representation of the present-day oceanic and atmospheric properties. For the third point, the three selected models are in the best half of the CMIP6 ensemble according to Agosta (2024) who ranked 45 models based on several atmospheric variables relevant for precipitation over Antarctica. These three models also have a high fidelity in the representation of the mean ocean properties, as detailed in Appendix A.* » (L59-65).

L58-64: To facilitate readability, bold or italic fonts for some phrases in this list could be used (e.g. for the evaluated water masses). As an alternative, these properties could be given in a table rather than in a list.

See the response above.

L65: Is the assessment presented in Figure 1 based on one ensemble member of the respective CMIP6 models or an average over all available ensemble members? As different ensemble members are used later in the manuscript, maybe add this information here (or in the figure caption) to avoid confusion.

Thanks for the suggestion. The assessment presented in Figure 1 is only based on the first member of each CMIP6 model. We added in Appendix A « *The analysis is done here for the first available member of each model.* » (L395).

L66: How is 'best' defined? Does UKESM-1-0-LL have one of the lowest RMSE in all three studies? Maybe state this more explicitly here.

For each study, the CMIP6 models are ranked by increasing RMSE. In Appendix A, we extended the description of ocean properties analysis a bit further than the response to L58-64 by summarising the metric used, the reference dataset to which it is compared and the period over which they are evaluated.

L68: If I understand Figure 1 correctly, MPI-ESM1.2-HR was evaluated in two of three studies (red triangles in Figure 1a and c). Please check.

Yes, MPI-ESM1.2-HR was evaluated in only two of the three studies.

L70-73: This is a very general sentence, in particular for readers not familiar with the representation of ice-shelf melting in CMIP models. What is meant by 'some kind of prescribed ice-shelf melting at depth'? Does this impact the assessment / ranking of CMIP6 models in Figure 1? I think it may be helpful to briefly discuss the link between the treatment of ice-shelf melting and the CMIP6 model assessment, if this information is mentioned here.

Prescribed meltwater flux at depth has an impact on the properties of the water masses and sea-ice. This inflow affects the stratification of the water column and can be more favourable to convective mixing by reducing the density at depth (Mathiot et al., 2017) and to the intrusion of circumpolar deep waters (Haigh et al., 2024). In contrast, models that prescribe meltwater flux only at the surface tend to increase the stratification of the water column and reduce exchanges between the surface and deeper waters, thereby preventing variability.

We have removed the word 'have some kind of' and replace with *« It is interesting to note that both UKESM1-0-LL and IPSL-CM6A-LR have prescribed ice-shelf melting that is vertically distributed to mimic the presence of unresolved ice-shelf cavities (Mathiot et al., 2017), which is known to be important for coastal ocean properties around Antarctica (Mathiot et al., 2017; Donat-Magnin et al., 2021). Most CMIP models prescribe meltwater fluxes at the surface, which tends to increase the ocean stratification (Mathiot et al., 2017) and reduce exchanges between the surface and deeper waters, thereby limiting variability at depth. »* (L71-75).

L74-77: Please add more detail on the assessment of atmospheric properties for the CMIP5/6 models in the manuscript, also given that Agosta et al. (2022) refers to a conference abstract. For example, which atmospheric properties have been evaluated and which method is used for the assessment?

For further explanation on the evaluation of atmospheric properties, the reader may now refer to Agosta (2024) which has just been referenced : https://doi.org/10.5281/zenodo.11595213.

Agosta (2024) evaluated 45 CMIP models around Antarctica by comparing their performance with the ERA5 reanalysis over the period 1980-2004 for 9 variables. The models are ranked based on two metrics, which are (i) the mean Root Mean Square Error (RMSE) over the 9 variables normalised by the multi-model RMSE and (ii) the second maximum implausible fraction, which corresponds to the fraction of the surface where the difference between CMIP models and ERA5 is greater than three times ERA5 standard deviation.

We have kept paragraph 2.1 concise and refer the reader to Agosta (2024) for the ranking of CMIP6 models according to atmospheric properties and to Appendix A for the evaluation of oceanic properties.

Figure 1: Please consider marking the selected CMIP6 models in a different way, e.g. by colouring the model name or adding a box around the model name. The red triangles can be easily confused with the other markers (or appear within the legend, compare Figure 1b).

We have added a star in front of the selected model on the x-axis (see Appendix A).

Figure 1: Please briefly introduce the abbreviations used in the legend, e.g. in panel b in the figure caption and / or in the main text (L58-64).

We added a table summarising the variables and metrics used for the evaluation. Figure 1 and the corresponding table have been moved to Appendix A.

L82: Please add a reference for the friction law.

As this comment appeared several times, we have added the law in the main text for clarity as follows *« The ice dynamics is computed by solving the Shallow Shelf Approximation (SSA) of the Stokes equations (MacAyeal, 1989), assuming an isotropic rheology following Glen's flow law (Glen, 1955) and a linear friction law (i.e., $\tau_b = C\, u_b$ where $\tau_b$ is the basal shear stress, $C$ the friction coefficient and $u_b$ the basal velocity). »* (L78-80).

L84: What do you mean by 'preferentially refined'? Please specify.

We removed the word 'preferentially'.

L85: What do you mean by 'high curvatures'? Please clarify.

Curvature is the second derivative of the modelled fields (velocity and ice thickness here), i.e., the Hessian matrix. For more explanation, the reader can refer to §2.2 of Gillet-Chaulet et al. (2012):

Anisotropic mesh adaptation is now widely used in numerical simulations especially with finite elements, as it allows to refine the mesh where needed to capture the flow features within a certain accuracy without increasing the computational cost excessively. The method is generally based on an estimation of the interpolation error used to adjust the mesh size so that the discretisation error is equally distributed over the whole domain. It can be shown that an estimate of the interpolation error induced by the meshing is obtained from the Hessian matrix of the modelled field, allowing to define an anisotropic metric tensor at each node (Frey and Alauzet,

In the main text, we replaced the sentence with *« The mesh is refined both close to the grounding line and in areas where observed surface velocities and thickness show high curvatures (i.e., high second derivative of the modelled field, Gillet-Chaulet et al., 2012). »* (L83-84).

L100-103: Are the ocean temperature corrections to match observed melt rates also based on Reese et al. (2023)? I assume that these may differ from the corrections presented in Reese et al. (2023) given the use of a different ocean climatology here. Please describe how the temperature corrections applied here are derived. It may also be helpful for the reader to briefly mention why temperature corrections are applied (instead of e.g. changing the PICO parameters to match present-day observed melt rates).

We used the set of parameters defined in Reese et al., (2023) which is based on the sensitivity of melt rates to ocean temperature changes obtained from both observations and numerical ocean projections. Ocean temperature corrections are not based on Reese et al., (2023). We have carried out our own temperature correction because (i) the climatology is different (ISMIP6 climatology in our paper instead of climatology from Schmidiko et al. (2014) in Reese's paper) and (ii) the ice-sheet geometry in our Elmer/Ice configuration resulting from an inversion initialisation is quite different from the geometry of Reese's model resulting from a long spin-up initialisation. Our aim is to match the observational estimates from Adusumilli et al., (2020) for all ice shelves over the average period 1995-2014 (see Figure 2) by applying temperature corrections between -2°C and 2°C (step of 0,1°C) in each basin defined in Reese et al., (2018).

In the main text, we now indicate:

- that the set of parameters are based on observations and ocean models: *« Here, the parameters are those detailed in Reese et al. (2023), i.e., C = 2 Sv m³ kg⁻¹ and $\gamma_T$ = 5.5×10⁻⁵ m s⁻¹, which are based on the observed or ocean-modelled sensitivity of melt rates to ocean temperature changes. »* (L98-99).

- the reasons that led us to carry out our own temperature correction: *« A correction of temperature, ranging from -1.8°C to 0.6°C with respect to the ocean climatology, is added to match the 1994-2018 melt rates estimates from Adusumilli et al. (2020) (see Fig. 1). This correction differs from Reese et al.*

*(2023) as the current ice-sheet geometry and the oceanic climatology used in this study are different from the one considered in Reese et al. (2023). » (L102-104).*

L106-107: Why is a 10 % reduction of the inverted friction coefficients applied? Is this based on testing, a 'best fit' or some other methodology? Does the reduction of the friction coefficients change the modelled velocities (as this quantify has been the target of the inversion)?

We minimised the RMSE between the modelled and the observed ice-sheet mass change for West Antarctica by applying reduction of the friction coefficient. A proper inversion was done to obtain the initial basal friction coefficients. Then, these coefficients were adjusted by trial and error to limit the model drift.

We have rephrased as: *« In contrast to Hill et al. (2023), we do not correct the surface mass balance to maintain a steady state, but we uniformly lower the inverted friction coefficients by 10% to reduce the model drift. For this, we minimise the RMSE between the modelled and the observed ice-sheet mass change for West Antarctica. » (L106-109).*

The friction coefficient correction does not significantly impact the ability of the model to reproduce the observed velocities. The initial RMSE between modelled and observed velocities was around 40 m/yr and increased by around 10 m/yr due to friction coefficient correction.

L107-108: The ice-sheet model configuration slightly overestimates mass loss in West Antarctica (when compared to the uncertainty ranges of the observations) if I understand Table 1 correctly.

We have removed the word 'correct' and articulated the sentence more clearly *« The resulting model configuration overestimates the mass loss trend in the West Antarctica by only 6% but still largely overestimates mass gain in East Antarctica and in the Peninsula (Tab. 1). As a consequence, the simulated Antarctic Ice Sheet is currently gaining a little mass (+36 Gt yr$^{-1}$, Tab. 1), instead of losing mass as observed (-109±56 Gt yr$^{-1}$, Tab. 1). » (L109-111).*

L108-109: Can you maybe add a brief remark (or a figure) on how large the trend bias in the ice-sheet model setup is?

See response to L107-108.

L109-110: Please specify the reference that your results (in terms of the Antarctic sea-level contribution) are compared to. Are the projections in response to the CMIP6 climate models analysed relative to each other? Do you substract from a control experiment to remove the drift? After reading the discussion, I think the trend is not removed.

The projections in response to the CMIP6 climate models are analysed relative to each other. We have replaced *« This is why our results are primarily analysed in relative terms. »* with *« However, this bias should not impact most of the analyses presented here, as the projections in response to the CMIP6 climate models are analysed relatively to each other. » (L112-114).*

L112-114: This formulation might be confusing for some readers.

We have rephrased as *« The future mass imbalance of Antarctica results from combined effects of changes in surface mass balance (SMB) and ice dynamics. In standalone ice-sheet simulations, variations in surface mass balance can be attributed to the atmosphere and dynamical mass loss can be attributed to the ocean as SMB changes have little impact on the Antarctic dynamical contribution to sea level over a century (Seroussi et al., 2014, 2023). Thus,*

*the effect of atmospheric and oceanic variations on Antarctic contribution to sea-level change can be analysed separately and then summed to reconstruct the combined effect (Bindschadler120 et al., 2013). »* (L116-121).

Figure 2: It may be helpful to indicate the most relevant ice shelves in a map (as already done for the Antarctic basins in Figure 7).

In this study, we have only made very limited reference to ice shelves and have mainly focused on the scale of basins. For this reason, we have only added indications of the regions to which the ice shelf belongs (basin number between 1 and 18) as in Figure 6.

[Figure]

L115-116: I got confused by the focus on the ocean here. Do you also run projections with ocean forcing only?

Given that SMB change has a limited impact on the dynamical contribution to sea level, the Antarctic total sea-level contribution can be calculated as the sum of the dynamical contribution (modulated by oceanic variability) and the SMB contribution (modulated by the atmosphere variability).

On the one hand, the SMB contribution is directly deduced from the emulated SMB (cumulative sum of the SMB - initial SMB in 2015). On the other hand, to investigate the dynamical contribution, we run Elmer/Ice simulations driven by the SMB of the first member of selected CMIP6 models and the ocean of several members of the selected CMIP models from 2015 to 2100 and we then remove the SMB contribution of the first member.

We added *« In our study, the SMB contribution to sea level is directly deduced from the emulated SMB anomalies (i.e., cumulative SMB – initial SMB). The contribution of ice dynamics to sea level is estimated through Elmer/Ice simulations driven by the SMB of the first member and the ocean of several members of the selected CMIP models. We then remove the SMB contribution of the first member to deduced the dynamical contribution. »* (L121-124).

L116: I am not sure if I understand what is meant by 'constrained'. Maybe consider replacing by e.g. 'driven' or 'forced', if applicable.

'driven' is used in the revised manuscript (L123).

L118-122: Please add more details on the selection of the CMIP6 ensembles members. I am not sure which section you are referring to for additional information on the selection, based on covering a wide spread in (1) possible ocean temperatures in the Amundsen Sea Embayment and (2) surface mass balance. Is the focus on the Amundsen Sea Embayment motivated by observed present-day mass loss in this region? Why is a different number of ensemble members chosen for each CMIP6 model? I appreciate the assessment of CMIP6 models in Figure 1, but, if I understand correctly, this evaluation justifies the choice of the CMIP6 model rather than the individual ensemble members for driving Elmer/Ice. It might be helpful for the reader to stress the link between the CMIP6 model evaluation, the assessment of the internal climate variability for these CMIP6 models and the selection of a subset of ensemble members for driving Elmer/Ice.

For ocean, we selected 5 members for each CMIP6 model, i.e., the two members with the coldest temperatures and the two members with the warmest temperatures in the Amundsen Sea continental shelf (Fig. 3j-l) as well as member 1, used by default in most of the studies. In total, we run 11 simulations, five with the IPSL-CM6A-LR model (r1i1p1f1, r3i1p1f1, r6i1p1f1, r11i1p1f1, r25i1p1f1, see https://goo.gl/v1drZl for CMIP6 convention name of ensemble members), four with the UKESM1-0-LL model as member 1 is already included in the temperature criteria (r1i1p1f2, r2i1p1f2, r4i1p1f2, r8i1p1f2) and two with the MPI-ESM1.2-HR model (r1i1p1f2, r2i1p1f2). As the oceanic variability is very low in MPI-ESM1.2-HR (Fig. 3f), we retained only member 1 and another member to verify the low impact of oceanic variability on the dynamic contribution.

We have replaced the current paragraph with *« Because of the numerical cost of our simulations, we select a limited number of members. In addition to the first member, the selection is made over the current period (1995-2014 means) to cover the widest range of values for the ocean temperature on the continental shelf in the Amundsen Sea. We focus on this region as (i) the largest mass loss is observed there and has been attributed to the ocean, and (ii) the amplitude of the standard deviation of the 1995-2014 mean potential temperature across all members is particularly high in this region (see section 3.1). In total, we run 11 simulations, five with the IPSL-CM6A-LR model (r1i1p1f1, r3i1p1f1, r6i1p1f1, r11i1p1f1, r25i1p1f1, see the CMIP6 naming convention in https://goo.gl/v1drZl), four with the UKESM1-0-LL model (r1i1p1f2, r2i1p1f2, r4i1p1f2, r8i1p1f2), and only two with the MPI-ESM1.2-HR model (r1i1p1f2, r2i1p1f2) given that its oceanic variability is very low (see section 3.1). »* (L129-136).

We took into account all the members available for the SMB contribution.

L120-122: I am not sure how familiar readers are with the CMIP variant labelling. While it may not be necessary to explain it in full detail, it may be helpful to briefly state that these lists describe different ensemble members for each of the CMIP6 models.

These abbreviations are a CMIP convention and are a brief description of the experiment. rX corresponds to realization index (i.e., the member number), iX to initialisation index, pX to physics index and fX to the forcing index.

We have added, in the main text, a link to a URL (https://goo.gl/v1drZl) that describes the CMIP6 writing conventions for the attributes (L134).

L129-135: This paragraph could be shortened. Maybe detailed information on the SMB in ISMIP6-Antarctic (L129-130) is not needed here.

We think that this is an important explanation for the ice-sheet community and we would prefer to keep this explanation.

L135: Maybe 'constrain' could be replaced by 'drive' or something similar, if applicable.

'drive' is used in the revised manuscript (L149).

L136-144: I would like to mention that my comments are limited to this manuscript, and I have not assessed the approach for emulating MAR and thus for obtaining the estimates of SMB used in this study. From my point of view, no detailed evaluation is needed here and it is fine to refer to the approach described in Jourdain et al. (2024, in discussion) as done. Please make sure that respective inputs and outputs of this approach become clear (see some of the following comments / questions).

We have shortened the paragraph a little to keep only the main features. As no evaluation is presented in detail, reader may refer to Jourdain et al. (2024, in discussion) if more details are needed (L150-156).

L137: 'surface melting' instead of 'melting'?

Yes, the manuscript has been corrected accordingly (L151).

L140-141: I am not sure if I understand correctly how the SMB for a given member is estimated. What is meant by 'perturbed as a function of the annual temperature difference'?

This sentence has been removed. The reader may refer to Jourdain et al. (2024, in discussion) for more details.

L150: Maybe replace 'a subset' by 'the subset'?

Yes, the manuscript has been corrected accordingly (L162).

L150: 'two first subsections' could be replaced by directly stating the subsections that you would like to refer to here to improve readability.

Yes, the manuscript has been corrected accordingly (L163).

L153-155: It might be helpful for the reader to explicitly state the ocean properties that reflect the oceanic internal climate variability in the beginning of this section (that is, salinity and temperature, as shown in Fig. 3, and as eventually described in the beginning of the following paragraph starting in L159).

At the beginning of the paragraph, we added *« Oceanic internal climate variability is investigated through salinity and temperature variability. »* (L165).

L155: What is meant by 'typical' standard deviation across model members? Does this mean that in most regions values are around 0.017 g kg-1 and 0.07°C for MPI-ESM1.1-HR? Or are these typical values for CMIP6 models?

We removed the word 'typical' which has indeed no relevance here (L167).

L159: 'continental shelf' instead of 'shelf'?

Yes, the manuscript has been corrected accordingly (L172).

L161: Maybe replace 'largest variability' by 'large variability' to avoid confusion? If I understand correctly, for example, the highest variability in mid-depth salinity for UKESM1-0-LL is found around Prydz Bay.

Yes, the manuscript has been corrected accordingly (L174).

L171: Is 'deep ocean' considered as same ocean depth as 'mid-depth'?

'deep ocean' is used here to characterise offshore waters, outside the region of the continental shelf, but Figure 2 shows the mean salinity and temperature standard deviation between 200 and 700 m over the entire area represented. We replaced 'deep ocean' by 'beyond the continental shelf' (L187).

L173: I would like to suggest to replace 'ice-sheet mass loss' by 'present-day ice-sheet mass loss' or something similar.

We replaced the sentence *« We now examine the Amundsen Sea more closely as the region is particularly important for the ice-sheet mass loss. »* with *« We now focus on the Amundsen Sea, as the region is currently experiencing the largest mass loss in Antarctica. »* (L189).

L184-190: I think it may be helpful to add figures on the assessment of oceanic internal climate variability based on 60-year averages, at least in form of a Supplementary Material, given that the discussion and recommendations reflect on the time period of averaging.

We agree. A figure similar to Figure 2 with a 60-year period instead of 20-year period has been added in Appendix C to support our arguments.

L197: Please specify that the increased water vapour saturation in warmer air then results in enhanced precipitation.

We added 'resulting in more precipitation' in the sentence *« By 2100 and for the SSP2-4.5 medium scenario, runoff is supposed to remain limited (Kittel et al. , 2021), so the SMB is projected to increase largely due to the increased water vapour saturation in warmer air, resulting in more precipitation (e.g. Krinner et al., 2008; Agosta et al.,2013). »* (L213-216).

L200: Is the SMB that you refer to here emulated or directly derived from the CMIP6 models? According to the caption of Figure 5 it is based on the MAR emulation. Maybe this could be specified again also in the main text.

The SMB is an emulation of MAR simulations. We specified 'emulated SMB' in the main text (L216).

L202: 'consistent' instead of 'consistently'?

Yes, the manuscript has been corrected accordingly (L220).

L202 - 205: This section also refers to the absolute SMB. Since the atmosphere is suggested as an important factor for the (spread in the) projected Antarctic sea-level contribution in the following section, and the choice of the CMIP6 model at the same time also modulates the projected sea-level change, I would like to suggest to add a related figure of SMB and atmospheric temperatures (e.g. in the Supplementary Material) for interested readers.

Thanks for the suggestion. We added a figure of absolute SMB, surface temperature and precipitation in Appendix D.

L204: 'which is both due to' instead of 'which is due both'?

*We replaced « The largest SMB internal climate variability is simulated along the coast of the Amundsen and Bellingshausen seas, which is due both to the particularly high present-day mean SMB and to the high internal climate variability of atmospheric circulation and air temperature in these regions (Fig. 5d-i). » with « The largest SMB variability is simulated along the coast of the Amundsen and Bellingshausen seas, which results from the high internal climate variability of atmospheric circulation (e.g., Amundsen Sea Low position) and air temperature in these regions (Fig. 4d-i). » (L222-224).*

L205: MPI-ESM1.2-HR also shows a relatively high standard deviation in atmospheric temperature in the Siple Coast region, compared to the other selected CMIP6 models. Is this relevant for the projected future evolution of the Antarctic Ice Sheet? In Figure 7, the ice-sheet response in basin 9 (Siple Coast) to atmospheric changes in MPI-ESM1.2-HR (showing mass loss) differs from the other CMIP6 models (showing mass gain).

*The temperature variability appears to be linked to the variability in the ASL position between members (see Figure 4g), which could explain why large anomalies are found in all the basins influenced by the ASL (basins 9,10,11 in Fig. 6).*

L205-208: What are the typical characteristics of the two Pacific-South American modes? Maybe add a short summary here for readers that are not familiar with Wang et al. (2022) and Marshall and Thompson (2016).

*We consider that this would be going too far away from the main focus and our sentence already gives a lot of information « As previously reported by Marshall and Thompson(2016), the internal climate variability of sea level pressure and air temperature have the typical characteristics of the twoPacific-South American modes (usually referred to as PSA1 and PSA2), which are associated with wave trains originating in the tropical Pacific and possibly modulated by feedbacks with clouds and sea-ice (Wang et al., 2022) ».*

L210-227: Maybe the link between the analysis of internal climate variability in CMIP6 models and the projected Antarctic sea-level contribution could be stressed here, in particular, for explaining some of the key results related to the uncertainties in the projected contribution of the Antarctic Ice Sheet to sea-level change. This includes, for example, the results that (1) atmospheric internal climate variability has a larger effect on the spread in the projected sea-level change with Elmer/Ice than oceanic internal climate variability, (2) the impact of the choice of the CMIP6 model on the sea-level contribution from Antarctica, and (3) the similarity of atmospheric internal climate variability for the selected CMIP6 models.

*We added new paragraphs in this section to stress the link between the analysis of internal climate variability in the CMIP6 models and the projected Antarctic sea level contribution: « The West Ross, Getz and Amundsen basins (n°9,10,11 in Fig. 6) show the most significant atmospheric and oceanic variability in the WAIS region. For the IPSL-CM6A-LR model, internal oceanic variability even exceeds atmospheric variability in these basins (Fig. 6b-c). As described in the previous paragraphs, this variability results from competition of CDW intrusions and convective mixing on the continental shelf (subsect. 3.1), and from the atmospheric circulation, especially the varying position of the Amundsen Sea Low depending on the members (subsect. 3.2). It should be noted that the MPI-ESM1.2-HR model does not show any internal oceanic variability, as expected from the analyses carried out in subsect. 3.1.*

*In East Antarctica, the Totten basin, which is currently experiencing the highest melt rates in East Antarctica (Rignot et al., 2019), and the Dronning Maud basin (No. 5 and 1 in Fig. 6) show strong internal oceanic variability reaching or exceeding the internal atmospheric variability for the three CMIP6 models. The other basins, like those of the Peninsula, show low basal melting and are largely dominated by internal atmospheric variability, induced primarily by interconnections with the tropical Pacific (see subsect. 3.2). »* (L248-258).

L213: For MPI-ESM1.2-HR there is a mean (?) mass loss related to the atmosphere in West Antarctica (Fig. 6i).

We used only 2 ensemble members for MPI-ESM1.2-HR. One of them projects a positive SMB sea-level contribution and the other one a negative contribution (Fig. 5i). Indeed, the average of the two ensemble members indicates a slightly positive contribution to sea level (Fig. 5i). We reformulated as « *(ii) increasing SMB (Fig. 5c), occurring in all regions for almost all members (Fig. 5f,i,l).* » (L232).

L215-217: Do you meant to refer to 'Pine Island and Thwaites ice shelves' / 'Getz ice shelf' here or rather the respective basins?

We refer to 'Pine Island and Thwaites ice shelves' / 'Getz ice shelf'. For clarity, we added « *The West Antarctic positive SLC is mostly explained by the dynamical response of Pine Island and Thwaites ice shelves (~3 cm in Fig. 5c, basin 11) as well as Getz ice shelf (~1 cm, basin 10). »*

L217-218: Is this drift of the unforced Elmer/Ice experiment removed or is the absolute Antarctic sea-level contribution given in the respective figures? I think I got confused by the statement in L109-110 (please also see my related previous comment). And can the influence of the drift on the trends in East Antarctica be quantified?

The drift of the control Elmer/Ice experiment (i.e., Elmer/ice configuration driven by current atmospheric and oceanic forcing) is not removed, which is why we had to correct the friction coefficient to reduce the drift.

All the figures display the absolute Antarctic sea-level contribution.

Based on Table 1, the mass change rate simulated in East Antarctica is equal to +107 Gt/yr whereas the observed mass change is +5±46 Gt/yr, meaning an overestimation between +56 Gt/yr and +148 Gt/yr.

L217: I would like to suggest to replace 'contaminated' by 'influenced' (or something similar).

Yes, the manuscript has been corrected accordingly (L236).

L218: What can be learned on the sensitivity of East Antarctica and the Antarctic Peninsula to internal climate variability based on the simulations presented here? Maybe you can make use of Figure 7 and add some details in this section.

See response to L210-227.

L220-222: I would like to suggest to give the full name of the CMIP6 models throughout the whole manuscript (consistent with e.g. Sect. 3.1).

For sure, the manuscript has been corrected accordingly.

L223: Basin 5 (including Totten glacier) shows a relatively large spread in the dynamical sea-level contribution (Fig. 7b). Can this be related to the assessment of oceanic internal climate variability in Sect. 3.2?

Yes, the ocean variability in Fig. 2e-f is particularly strong near Totten, which explains why ocean has a larger effect on internal variability there. This is mentioned in the revised manuscript *« In East Antarctica, the Totten basin, which is currently experiencing the highest melt rates in East Antarctica (Rignot et al., 2019), and the Dronning Maud basin (No. 5 and 1 in Fig. 6) show strong internal oceanic variability reaching or exceeding the internal atmospheric variability for the three CMIP6 models. »* (L254-256).

L226-227: Please add more information on this finding. How is the number determined? Can it be seen in a figure (likely Figure 6)?

For each of the three CMIP6 models, we compare the amplitude of sea-level contribution (SLC) variability induced by ice dynamics, which is largely modulated by ocean-induced basal melting (Figure 5b), with the amplitude of the SLC variability due to the surface mass balance (Figure 5c), which is largely driven by the atmosphere.

When we talk about amplitude, we mean the difference between the SLC value in 2100 of the member giving the smallest contribution and the member giving the largest contribution.

We added a reference to Figure 5 in the main text for a clearer explanation *« On average, by the end of the century, the amplitude of SLC variability related to the atmosphere (Fig. 5c) is 3.4 times higher than that related to the ocean (Fig. 5b). »* (L245-246).

Figure 6: I assume that the number in brackets in the legend refers to the number of ensemble members for each CMIP6 model. Why do the numbers differ between panel a/b and panel c? Please check.

Figure 6 is now Figure 5. You are completely right, the number in brackets in the legend refers to the number of ensemble members for each CMIP6 model. We added this information to the legend *« The number in bracket refers to the number of selected members for each CMIP6 model. »*

For panel b): For ocean, we selected 5 members for each CMIP6 model, i.e., the two members with the coldest temperatures and the two members with the warmest temperatures in the Amundsen Sea continental shelf (Fig. 3j-l) as well as member 1, used by default in most of the studies. In total, we run 11 simulations, five with the IPSL-CM6A-LR model (r1i1p1f1, r3i1p1f1, r6i1p1f1, r11i1p1f1, r25i1p1f1, see https://goo.gl/v1drZl for CMIP6 convention name of ensemble members), four with the UKESM1-0-LL model as member 1 is already included in the temperature criteria (r1i1p1f2, r2i1p1f2, r4i1p1f2, r8i1p1f2) and two with the MPI-ESM1.2-HR model (r1i1p1f2, r2i1p1f2). As the oceanic variability is very low in MPI-ESM1.2-HR (Fig. 3f), we retained only member 1 and another member to verify the low impact of oceanic variability on the dynamic contribution.

We have replaced the current paragraph with *« Because of the numerical cost of our simulations, we select a limited number of members. In addition to the first member, the selection is made over the current period (1995-2014 means) to cover the widest range of values for the ocean temperature on the continental shelf in the Amundsen Sea. We focus on this region as (i) the largest mass loss is observed there and has been attributed to the ocean, and (ii) the amplitude of the standard deviation of the 1995-2014 mean potential temperature across all members is particularly high in this region (see section 3.1). In total, we run 11 simulations,*

*five with the IPSL-CM6A-LR model (r1i1p1f1, r3i1p1f1, r6i1p1f1, r11i1p1f1, r25i1p1f1, see the CMIP6 naming convention in https://goo.gl/v1drZl), four with the UKESM1-0-LL model (r1i1p1f2, r2i1p1f2, r4i1p1f2, r8i1p1f2), and only two with the MPI-ESM1.2-HR model (r1i1p1f2, r2i1p1f2) given that its oceanic variability is very low (see section 3.1). »* (L129-136).

For panel c): we took into account all the members available.

For panel a): the total contribution is a combination of dynamical contribution and SMB contribution, so the number of ensemble member for the total contribution depends on the limited ensemble members of the dynamical contribution and thus have the same number of ensemble members than panel b).

Figure 6: Does the solid line indicate the mean? Maybe I have missed this.

We forgot to include this information in the legend of Figure 5. The manuscript has been corrected like this *« The solid line represents the multi-member mean, while the shaded area represents the range of values covered by the ensemble members. ».*

L232-233: Maybe specify which paleoclimate proxies are used in Parsons et al. (2020) (similar to stating that Casado et al. 2023 base their analysis on ice core reconstructions in the following paragraph), for readers that are not familiar with this study?

We added a reference to PAGES2k, (2019) for interested reader (L264).

L232: 'global mean surface air temperature' or its variability?

Thank you for the careful reading, the word 'standard deviation' is missing.

*We rephrased as « Parsons et al. (2020) compared the distribution of standard deviation of global mean surface air temperature of CMIP piControl simulations to paleoclimate proxies representative of the 1450-1849 period (PAGES2k, 2019). »* (L263-264).

L233: Maybe you could add the observational plausible range for the temperature variability for comparison with the values for the CMIP6 models?

Yes, the manuscript has been corrected accordingly (L266).

L243: If possible, maybe a conclusion on the representation of atmospheric variability in CMIP models could be added, bringing together the results of this study (Sect. 3.1) with the previous literature?

We have generalised the assessment of internal oceanic and atmospheric variability carried out in §3.1 to 15 CMIP6 models (see Appendix B) and have added in the discussion:

For atmosphere: *« Both IPSL-CM6A and MPI-ESM1.2-HR have an internal variability of their 20-year mean surface air temperature close to the CMIP6 multi-model median (Appendix B), so their atmospheric multi-decadal variability is possibly underestimated given the results of Casado et al. (2023). Nevertheless, this variability is significantly stronger in UKESM1-0-LL, which suggests that our study may cover realistic atmosphere internal variability. »* (L275-278).

For ocean: « When compared with 12 other CMIP6 models (Appendix B), the three selected models cover the whole range of oceanic multi-decadal variability in the CMIP6 ensemble,

with one of the lowest values (MPI-ESM1.2-HR), one close to the multi-model median (UKESM1-0-LL) and one of the highest values (IPSL-CM6A-LR) » (L280-282).

L245: Maybe add a reference for these observations?

We added a reference to CTD profiles measured in the Amundsen Sea and described in Dutrieux et al., 2014 and Jenkins et al., 2018: « *The low variability of the MPI-ESM1.2-HR model is inconsistent with the temperature and salinity profiles observed in the Amundsen Sea (Dutrieux et al., 2014; Jenkins et al., 2018)... »* (L282-284).

L258-259: As the choice of the CMIP6 model is suggested to have a similar impact on the Antarctic sea-level contribution as the internal climate variability, it may be helpful to add a short paragraph on this finding also in Sect. 3.3 (in addition to this statement in the discussion).

Yes, the manuscript has been corrected accordingly (L239-243).

L265-266: This sentence can maybe be reformulated. As already indicated, given the limited impact of the emission pathway on the Antarctic sea-level contribution to 2100, SSP2.4.5 may not be the main explanation for the Elmer/Ice projections presented here being at the lower end of previous projections.

We removed the argument of GHG scenario in the sentence and we rephrased like this « *In contrast, our simulations are at the very low end of the ensemble of other ice-sheet projections (-8.5 to -1.3 cm, Fig. 5a). This is partly due to the present-day drift in East Antarctica and Peninsula that we did not remove from our projected trends as opposed to the aforementioned other models. »* (L303-306).

L269: I would like to suggest 'ocean-induced melting' (or something similar).

Yes, the manuscript has been corrected accordingly (L308).

L270: I am not sure if I understand the meaning of 'high variability of 20-year means' correctly. Maybe it is possible to rephrase?

We have replaced « *However, given the high variability of 20-year means, correcting a random phase of the historical CMIP simulations towards the actual 1995-2014 period may significantly shift the projections. »* with « *However, given the wide confidence interval on a 20-year means ([0.06°C;0.24°C] for air temperature and [0.02°C;0.12°C] for oceanic temperature, see Fig. B1), correcting a random phase of the historical CMIP simulations towards the actual 1995-2014 period may significantly shift the projections. »* (L308-311).

L278-283: This paragraph seems to contain much information that is also given in the beginning of the following Sect. 4.3. I would like to suggest to move L278-283 to Sect. 4.3 and merge with the first part of this section.

The information is clearly distinguishable and the paragraph has not been substantially modified.

L284-354: This is an interesting analysis and discussion. If I understand correctly, it supports to include multiple CMIP ensemble members in Antarctic sea-level projections as done in the work presented here. At the same time, it seems slightly detached from the previous parts of the manuscript. I would like to suggest two options that may help to add focus to this section:

A) This section may be shortened, summarizing the main analysis and the conclusion. The major part of the analysis may be moved to the Supplementary Material.

B) Parts of this section (e.g., the metrics and justification for these metrics) may be included in the Methods, and the outcomes could be highlighted and discussed with the main results. If applicable, the Antarctic sea-level contribution for the 'best' ensemble members could be added separately to e.g. Figure 6.

Thanks for the suggestion. We chose option A).

The discussion section on the identification of the best member has been revised. To facilitate reading, Figure 8 and the list of associated metrics has been moved to Appendix E and similar analysis has been performed with UKESM-0-LL to get more robust conclusions.

In the revised paragraph (L330-344), we specified that the metrics considered were chosen to ensure:

- a good representation of the mean atmospheric and oceanic states. We selected variables directly used to drive the ice-sheet model, such as SMB for the atmosphere and temperature for the ocean. For the ocean, we focused our analyses on the Amundsen sector as the region experiences the current main mass loss and CTD profile data are available for a relatively long period from 1994 to 2018.

- a good representation of the amplitude of oceanic variability using the same observational data described in the previous paragraph. We did not evaluate the variability of SMB since it has been relatively stable in recent years.

- a good representation of important modes of variability known to affect the ocean and atmosphere in/around Antarctica. We focus our analyses on the indices representative of the Southern Annular Model and the Interdecadal Pacific Oscillation.

- a good phasing of internal variability with observations, which could be important for future detection/attribution studies and for projected Antarctic contribution to sea-level rise. We chose two variables, sea-ice concentration and the presence of warm periods on the continental shelf of the Amundsen Sea to provide insights on the phasing of internal variability.

It should be noted that this part of our study remains exploratory and the choice of variables and metrics is indeed very subjective, which is part of the caveats that we discuss.

L290: I am not sure if I understand this phrase. Maybe replace 'assess' by e.g. 'demonstrate'?

The paragraph has changed. See response to L284-354.

L368-384: This paragraph contains many valid and helpful recommendations for future assessments of the Antarctic contribution to sea-level change. However, some of these recommendations do not seem to be directly justified by the presented results, or a better link to and additional information in Sect. 3 may be needed. For example, a fully-coupled assessment would be ideal to include feedbacks of the ice sheet with the ocean and the atmosphere, but some additional discussion how this would e.g. improve the representation of or remove biases in the internal climate variability in the selected CMIP6 models presented in Sect. 3.1 and Sect. 3.2 may be needed to directly relate to this study.

Based on the comments of all the reviewers, we decided to only keep the two recommendations that are the most motivated by our findings, and we made sure that the link with our findings is explicit.

We have removed the recommendation on coupled ice-sheet/climate models as it is not clearly demonstrated in our study. Instead, the discussion on ice-sheet/climate coupling has been addressed in section 4.1 which deals with the issue of the robustness of internal variability in climate models. The recommendation regarding initialisation was also removed, as the topic of initialisation is not directly addressed in the paper.

L379-380 / L382: Do the 'various members' / 'multiple members' refer to the CMIP6 model ensemble members or to ice-sheet initial states?

This sentence has been removed from the revised manuscript.

---

## Author Response (AR2)

We thank the Editor Claudia Timmreck and the two reviewers for their second attentive evaluation of the manuscript. We hope that the clarifications we provide in this second review will fully address the comments and questions raised by the reviewers. To ensure clarity, the reviewer's comments are written in black and our responses in light blue.

**Reviewer #1**

**General comments:**

This manuscript is a revision of an initial submission, in response to comments from three reviewers. The authors describe research based on modeling experiments, with the goal of characterizing projection uncertainty related to internal climate variability of the atmosphere and the ocean. First, they choose a set of three CMIP6 model ensembles, forced by the SSP2-4.5 scenario. Then they characterize the internal climate variability for each. Finally, the authors force a continental model of the Antarctic Ice Sheet with a representative subset of ensemble members in order to determine how internal climate variability in the atmosphere and the ocean affects Antarctica sea-level contribution over the coming century. The authors conclude that internal climate variability affects sea-level contribution by 45-93%, and this spread is dictated most strongly by the atmospheric forcing in most regions. Based on their results, CMIP models significantly vary in their representation of internal climate variability. Therefore, they suggest that that ice sheet model projection efforts should consider how well a CMIP model represents decadal-scale variability when choosing a future atmosphere and ocean forcing. They also recommend a longer reference period be used (greater than the standard 20 years) to calculate climate anomalies. Finally, they propose that multiple ensemble members be run to better capture internal climate variability in ice sheet model projections.

In response to suggestions by the reviewers, the authors have made a significant number of revisions to the manuscript, including a valuable reorganization effort. I find that they have improved the presentation of their results and the overall clarity of the manuscript's story. In addition, the authors have included a comprehensive response to all the reviewer comments and have adequately explained their modification/lack of modification stemming from each comment. Consequentially, I believe this work represents a novel contribution to the field, and I support the acceptance of this revised manuscript. I only have a few additional questions/comments about phrasing that I think may confuse the reader. I note these below.

**Specific comments:**

Line 102: Please specify here the spatial fidelity of this correction (i.e. per glacier I think). Temperature correction is performed by region (the 19 regions defined in Reese et al. 2018 mentioned in the previous sentence in the main text). Nevertheless, we have verified that this basin-scale correction results in melting in agreement with Adusumilli's observational estimates on a local ice-shelf scale (Fig. 1).

We have replaced *"A correction of temperature, ranging from -1.8°C to 0.6°C with respect to the ocean climatology, is added to match the 1994-2018 melt rates estimates from Adusumilli et al. (2020) (see Fig. 1)."* by *"A correction of temperature, ranging from -1.8°C to 0.6°C with*

*respect to the ocean climatology, is added to match the 1994-2018 observational melt estimates from Adusumilli et al. (2020) for each of the 19 regions. This regional correction resulted in improved estimates of local ice-shelf melting, except for Totten and Thwaites ice shelves (see Fig. 1)." (L101-104)*

Line 108: Please include more details about the steps of the friction tuning process. For instance, what years are considered for the mass change? You minimize RMSE of total ice sheet mass change in all of West Antarctica during that period, using which observations? If I understand correctly, this results in an estimation of 10%, and then you apply this reduction in friction to the entire ice sheet?

For clarity, we have added more details about the friction tuning process. We have also realized that bias is more appropriate than RMSE as there is a single value. We have therefore replaced *"For this, we minimise the RMSE between the modelled and the observed ice-sheet mass change for West Antarctica."* by *"For this, we minimise the model bias in West Antarctic grounded ice mass loss with respect to the 1995-2014 observational estimate of the IMBIE Team (2018). West Antarctica is chosen to tune the basal friction coefficients as the ice dynamics is known to strongly explain mass loss in this sector. We then apply the resulting 10% correction to the friction coefficients of the entire ice sheet." (L110-113)*

To avoid any confusion about the period over which the calibration is performed, we have also indicated IMBIE values for the period 1995-2014 (instead of 1992-2017) in Table 1.

Line 122: Please indicate that the SMB is cumulative "over time".

We have reorganised the paragraph following the comment hereafter and the term *cumulative* no longer appears.

Lines 122-124: These sentences are awkward and a bit confusing. Please clearly rephrase them to describe what was done. For instance, it sounds like only 3 different SMBs were used for forcing (one per CMIP model), but all the ocean forcing was run. It is clear that the dynamics and SMB can reconstruct the mass balance in this way, but can you explain why only the first member SMB is used? Is there a computational constraint that prompted this decision?

Due to the number of comments received on section 2.3, we have reorganised the section to improve clarity. Section 2.3 is now divided into 4 paragraphs:

- paragraph 1: a general paragraph explaining that Antarctic contribution to sea-level rise can be seen as the sum of the SMB contribution and the dynamical contribution. We have also added a general sentence that explains how we calculate each of these contributions in our study, i.e., through emulation of a regional climate model driven by the atmosphere of the selected CMIP6 models for SMB contribution and through simulation of ice-sheet model driven by the ocean of the selected CMIP6 models for dynamical contribution.

- paragraph 2: a paragraph explaining the choice of scenario used for the projections (scenario SSP2-4.5).

- paragraph 3: a paragraph explaining in greater detail the method used for the SMB contribution.

- paragraph 4: a paragraph explaining in greater detail the method used to estimate the dynamical contribution.

We hope that the addition of the general sentence in the first paragraph helps the reader to understand the method more clearly: *"In this study, variations in SMB are evaluated through the emulation of a regional climate model driven by the atmosphere of the selected CMIP6 models, while the dynamical mass losses are calculated using the ice-sheet model Elmer/Ice driven by the ocean of the selected CMIP6 models." (L125-127)*

Lastly, we explained the lack of impact of the choice of member 1 for the SMB within the Elmer/Ice simulations at the beginning of paragraph 4: *"For each selected CMIP6 model, the contribution of ice dynamics to sea level is estimated through Elmer/Ice simulations driven by the SMB of the first member (as SMB changes have little impact on the Antarctic dynamical contribution to sea level over a century, the choice of SMB member does not matter) and the ocean of several members. We then remove the SMB contribution of the first member to isolate the dynamical contribution of each member." (L150-153)*

Small SMB variations between each member have little impact on the dynamics (Seroussi et al, 2023). However, the total absence of SMB during the simulation could have impacts on the dynamics, which is why we kept atmospheric forcing for the Elmer/Ice simulation. Note that we could also have forced the Elmer/Ice model by the same member for the atmosphere and the ocean, rather than only member 1 for the atmosphere, even though, taking the same member for the atmosphere theoretically gives a better estimate of the internal variability purely related to the ocean.

Line 126: This statement is not clear. Please rephrase and specify what is meant by word "seems". Is it that this scenario is the best match to present-day warming in Antarctica?

'seems' is not appropriate. We wanted to indicate that we chose the SSP2-4.5 scenario because it is considered as one of the most plausible scenarios (at global scale) based on the current effort and policies (Hausfather and Peters, 2020) whereas SSP5-8.5 is considered unlikely (Huard et al, 2022). We have not verified whether this scenario is the one that most closely matches current warming in Antarctica, as the ice-sheet simulations carried out for ISMIP6-Antarctica show that the choice of scenario for total mass change projections to 2100 has little impact, and as both SMB and ocean-induced melt rates are influenced by the scenario only after ~2050 (Jourdain et al., 2024; Naughten et al., 2023).

We have removed the word *'seems'* and the sentence is now *"We use the medium SSP2-4.5 scenario, which corresponds to a global warming of 1.4 to 3.0°C from 1995-2014 to 2081-2100*

*(90% confidence interval, Lee et al., 2021) and considered plausible in view of current efforts to tackle climate change (Hausfather and Peters, 2020)." (L128-130)*

Line 129: To prevent confusion, maybe you could explicitly list somewhere the various subsets of runs and forcing used for your analysis: i.e., those used for characterization of variability, those for deducing dynamics, and those used for 2100 ice sheet model simulations. This could be in a diagram or table of some sort, or maybe a few sentences in the methods where the number of simulations and the forcings for each analysis are described.

We think that this is a very good idea, so we have added a table in the appendix describing the number of members used and the method used for each analysis (see Appendix C).

For the projections, we have also added for each of the contributions (dynamical or SMB) the number of members used for each of the three selected CMIP6 models in the main text, as well as a reference to the table in Appendix C.

Line 130: For clarity, please specify in these sentences that this is for the selection of CMIP model ensemble members used for the ice sheet model runs.

See response to comment on Line 129.

Line 214: supposed => "expected" ?

Yes, the manuscript has been corrected accordingly. (L222)

Line 229: Please clarify that this is for all the members chosen, i.e. a subset of members (instead of using "all")

We have added the word *'selected'* in the sentence *"In our ice-sheet projections, Antarctica gains mass over the century for all selected members of the three CMIP models…" (L237)*

Figure 4, caption: Please clarify what is meant by the "MAR-based" reconstructions here.

We have replaced *"standard deviation of the 1995-2014 mean SMB across the ensemble relative to the multi-member mean, from the MAR-based reconstructions"* by *"standard deviation of the 1995-2014 mean SMB across the ensemble relative to the multi-member mean. The SMB shown is not a direct CMIP6 output but is derived from emulated behaviour of the regional climate model MAR driven by selected CMIP6 models."*

Figure 5, caption: Just a note that the caption goes beyond the page, so it was not possible to read it all.

We have reduced the size of the figure to ensure that the last line of the caption appears.

Line 367: Please specify that this result pertains to your chosen medium-range scenario over ~ 1 century time scale.

We have added *'for medium-range scenario'* in the sentence *"In this study, we show that internal climate variability affects the Antarctic contribution to changes in sea level until 2100, for medium-range scenario, by 45%-93%." (L369-370)*

Line 398: seat => "sit"?

Yes, the manuscript has been corrected accordingly. (L406)

Line 457: Extra ")" included at the end of the sentence.

Yes, the manuscript has been corrected accordingly. (L478)

**References:**

Huard, D., Fyke, J., Capellán-Pérez, I., Matthews, H. D., & Partanen, A. I. (2022). Estimating the likelihood of GHG concentration scenarios from probabilistic Integrated Assessment Model simulations. *Earth's Future*, *10*(10), e2022EF002715.

Naughten, K. A., Holland, P. R., & De Rydt, J. (2023). Unavoidable future increase in West Antarctic ice-shelf melting over the twenty-first century. *Nature Climate Change*, *13*(11), 1222-1228.